# Lower Bounds on Metropolized Sampling Methods for Well-Conditioned Distributions

**Yin Tat Lee**
University of Washington and Microsoft Research
yintat@uw.edu

**Ruoqi Shen**
University of Washington
shenr3@cs.washington.edu

**Kevin Tian**
Stanford University
kjtian@stanford.edu

## Abstract

We give lower bounds on the performance of two of the most popular sampling methods in practice, the Metropolis-adjusted Langevin algorithm (MALA) and multi-step Hamiltonian Monte Carlo (HMC) with a leapfrog integrator, when applied to well-conditioned distributions. Our main result is a nearly-tight lower bound of $\widetilde{\Omega}(\kappa d)$ on the mixing time of MALA from an exponentially warm start, matching a line of algorithmic results [DCWY18, CDWY19, LST20a] up to logarithmic factors and answering an open question of [CLA$^+$20]. We also show that a polynomial dependence on dimension is necessary for the relaxation time of HMC under any number of leapfrog steps, and bound the gains achievable by changing the step count. Our HMC analysis draws upon a novel connection between leapfrog integration and Chebyshev polynomials, which may be of independent interest.

## 1 Introduction

Sampling from a continuous distribution in high dimensions is a fundamental problem in algorithm design. As sampling serves as a key subroutine in a variety of tasks in machine learning [AdFDJ03], statistical methods [RC99], and scientific computing [Liu01], it is an important undertaking to understand the complexity of sampling from families of distributions arising in applications.

The more restricted problem of sampling from a particular distribution family we call "well-conditioned" has garnered a substantial amount of recent research effort from the algorithmic learning and statistics communities. This specific family is interesting for a number of reasons. First, it is *practically relevant*: Bayesian methods have found increasing use in machine learning applications [Bar12], and many distributions in these settings are well-conditioned, such as multivariate Gaussians, mixture models with small separation, and densities arising from Bayesian logistic regression with a Gaussian prior [DCWY18]. Moreover, for several widely-used sampler implementations in popular packages [Aba16, CGH$^+$17], such as the Metropolis-adjusted Langevin algorithm (MALA) and Hamiltonian Monte Carlo (HMC), the target having a small condition number is in some sense a *minimal assumption for known provable guarantees* (discussed more thoroughly in Section 1.2).

Finally, the highly-documented success of first-order (gradient) methods in optimization [Bec17], which are particularly favorable in the well-conditioned setting, has driven interest in connections between optimization and sampling. Exploring this connection has been highly fruitful: since seminal work of [JKO98], demonstrating that the Langevin dynamics which MALA and HMC discretize has an interpretation as gradient descent on density space, a flurry of work [Dal17, CCBJ18, DCWY18, DR18, DM19, DMM19, CDWY19, CV19, SL19, MMW$^+$19, LST20a, LST20b, CLA$^+$20] has ob-

35th Conference on Neural Information Processing Systems (NeurIPS 2021).

tained improved upper bounds for the mixing of various Langevin discretizations for well-conditioned sampling. Many of these works draw inspiration from first-order optimization.

On the other hand, demonstrating *lower bounds* on the complexity of sampling tasks (in the well-conditioned regime or otherwise) has proven remarkably challenging. To our knowledge, there are few unconditional lower bounds for sampling tasks. This is in stark contrast to the theory of optimization, where there are matching upper and lower bounds for a variety of problems and query models, such as convex optimization under first-order oracle access [Nes03]. This gap in the algorithmic theory of sampling is our primary motivation: we aim to answer the following more restricted question.

*What is the complexity of the popular sampling methods, MALA and HMC,*
*for sampling well-conditioned distributions?*

The problem we study is still less general than *unconditional query lower bounds* for sampling, in that our lower bounds are *algorithm-specific*; we characterize the performance of particular sampling algorithms on a distribution family. However, we believe asking this question, and developing an understanding of it, is an important first step towards a theory of complexity for sampling. On the one hand, algorithm-specific lower bounds highlight their weaknesses, pinpointing bottlenecks in attaining faster rates; this is useful for algorithm designers, as it clarifies what key barriers must be overcome. On the other hand, the hard instances which arise in designing lower bounds may have important structural properties, paving the way to stronger and more general bounds.

For these reasons, we focus on characterizing the complexity of MALA and HMC (see Section 2 for algorithm definitions), which are often the samplers of choice in practice, by lower bounding their performance when they are used to sample from densities proportional to $\exp(-f(x))$, where $f : \mathbb{R}^d \to \mathbb{R}$ has a finite condition number $\kappa < \infty$.[1] We call such a density "well-conditioned." Finally, we explicitly assume throughout that $\kappa = O(d^4)$, as otherwise in light of our lower bounds the general-purpose logconcave samplers of [LV07, JLLV20, Che21] are preferable.

## 1.1 Our results

Our primary contribution is a nearly-tight characterization of the performance of MALA for sampling from two high-dimensional distribution families without a warm start assumption: well-conditioned Gaussians, and the more general family of well-conditioned densities. We prove the following two lower bounds on MALA's complexity, which is a one-parameter algorithm (for a given target distribution) depending only on step size. We fix a scale $[1, \kappa]$ on the problem, as otherwise non-scale-invariance can be exploited to give more trivial lower bounds (cf. Appendix B, supplement).

**Theorem 1.** *For every step size, there is a target Gaussian on $\mathbb{R}^d$ whose negative log-density always has Hessian eigenvalues in $[1, \kappa]$, such that the relaxation time of MALA is $\Omega(\frac{\kappa\sqrt{d}}{\sqrt{\log d}})$.*

**Theorem 2.** *For every step size, there is a target density on $\mathbb{R}^d$ whose negative log-density always has Hessian eigenvalues in $[1, \kappa]$, such that the relaxation time of MALA is $\Omega(\frac{\kappa d}{\log d})$.*

To give more context, MALA is a Metropolis-adjusted Markov chain, which performs updates preserving the stationary distribution. It can be derived by applying a filter on the Euler discretization of the Langevin dynamics, a stochastic differential equation with stationary density $\propto \exp(-f(x))$:

$$dx_t = -\nabla f(x_t)dt + \sqrt{2}dW_t,$$

where $W_t$ is Brownian motion. Such *Metropolis-adjusted* methods typically provide total variation distance guarantees, and attain logarithmic dependence on the target accuracy.[2] The mixing of such chains is governed by their relaxation time, also known as the inverse *spectral gap*.

However, in continuous space, it is not always clear how to relate the relaxation time and *mixing time*, the time it takes to reach total variation $\frac{1}{e}$ to the stationary distribution from a given start (we

---

[1]$f$ has condition number $\kappa$ if it is $L$-smooth and $\mu$-strongly convex (has all second derivatives in the range $[\mu, L]$), where $\kappa = \frac{L}{\mu}$; we overload this terminology and say the density itself has condition number $\kappa$.

[2]This contrasts with a different family of *unadjusted* discretizations, which are analyzed by coupling them with the stochastic differential equation they simulate (see e.g. [Dal17, CCBJ18] for examples), at the expense of a polynomial dependence on the target accuracy; we focus on Metropolis-adjusted discretizations in this work.

choose $\frac{1}{e}$ to match the literature, but any constant smaller than 1 will do). There is extensive research on when these two quantities are relateable [BGL14], but typically these arguments are tailored to chain-specific properties, causing relaxation time bounds to not be entirely satisfactory in some cases. We complement Theorems 1 and 2 with a *mixing time* lower bound from an exponentially warm start.

**Theorem 3.** *For every step size, there is a target density on $\mathbb{R}^d$ whose negative log-density always has Hessian eigenvalues in $[1, \kappa]$, such that MALA initialized at an $\exp(d)$-warm start requires $\Omega(\frac{\kappa d}{\log^2 d})$ iterations to reach $e^{-1}$ total variation distance to the stationary distribution.*

We remark that Theorem 3 is the first *mixing time* lower bound for discretizations of the Langevin dynamics we are aware of: other related bounds have primarily been on relaxation times [CV19, LST20a, CLA+20]. It is unknown how to obtain a starting distribution for a general $\kappa$-conditioned distribution with warmness better than $\kappa^d$ (obtained by $\mathcal{N}(x^*, \frac{1}{L}\mathbf{I})$ where $L$ is the smoothness and $x^*$ is the mode).[3] A line of work [DCWY18, CDWY19, LST20a] analyzed the performance of MALA under this start, culminating in a mixing time of $\widetilde{O}(\kappa d)$, where $\widetilde{O}$ hides logarithmic factors in $\kappa$, $d$, and the target accuracy. On the other hand, a recent work [CLA+20] demonstrated that MALA obtains a mixing time scaling as $\widetilde{O}(\text{poly}(\kappa)\sqrt{d})$, initialized at a *polynomially* warm start,[4] and further showed that such a mixing time is tight (in its dependence on $d$). They posed as an open question whether it was possible to obtain $\widetilde{O}(\text{poly}(\kappa)d^{1-\Omega(1)})$ mixing from an explicit start.

We address this question via Theorem 3, showing the $\widetilde{O}(\kappa d)$ rate of [LST20a] for MALA applied to a $\kappa$-conditioned density is nearly tight. To prove Theorems 1-3, in each case we exhibit an $\exp(-d)$-sized set according to the stationary measure where either the chain cannot move in $\text{poly}(d)$ steps, or must choose a very small step size. Beyond exhibiting a mixing bound, this demonstrates the subexponential warmness assumption in [CLA+20] is necessary for their improved bound. To our knowledge, this is the first *nearly-tight* characterization of a specific sampler's performance, and improves bounds of [CLA+20, LST20a]. It also implies going beyond $\widetilde{O}(\kappa d)$ mixing for general well-conditioned densities requires subexponential warmness.

The lower bound statement of Theorem 3 is warmness-sensitive, and is of the following (somewhat non-standard) form: for $\beta = \exp(d)$, we provide a lower bound on the quantity

$$\inf_{\substack{\text{algorithm parameters}}} \sup_{\substack{\text{starts of warmness} \leq \beta \\ \text{densities in target family}}} \text{mixing time of algorithm.}$$

In other words, we are allowed to choose both the hard density and starting distribution adaptively based on the algorithm parameters (in the case of MALA, our choices are in response to the step size). We note that this type of lower bound is compatible with standard conductance-based upper bound analyses, which typically only depend on the starting distribution through the warmness parameter.

We further study the multi-step generalization of MALA, known as Hamiltonian Monte Carlo with a leapfrog integrator (which we refer to as HMC). In addition to a step size $\eta$, HMC is parameterized by a number of steps per iteration $K$, making $K$ gradient queries in every step to perform a discretization of the Langevin dynamics, before applying a Metropolis filter. Multi-step HMC has the potential of attaining overall runtime gains by effectively taking longer steps. It was recently shown [CDWY19] that under higher derivative bounds, balancing $\eta$ and $K$ more carefully depending on problem parameters could break the apparent $\kappa d$ barrier of MALA, even from an exponentially warm start.

It is natural to ask if there is a stopping point for improving HMC. We demonstrate that HMC cannot obtain a better relaxation time than $\widetilde{O}(\kappa\sqrt{d}K^{-1})$ for any $K$, even when the target is a Gaussian. Since every HMC step requires $K$ gradients, this suggests $\widetilde{\Omega}(\kappa\sqrt{d})$ queries are necessary.

**Theorem 4.** *For every step size and count, there is a target Gaussian on $\mathbb{R}^d$ whose negative log-density always has Hessian eigenvalues in $[1, \kappa]$, such that the relaxation time of HMC is $\Omega(\frac{\kappa\sqrt{d}}{K\sqrt{\log d}})$.*

In Appendix B of the supplement, we also lower bound how much increasing $K$ can improve HMC in the in-between range $\kappa\sqrt{d}$ to $\kappa d$. In particular, we demonstrate that if $K \leq d^c$ for some constant $c$, then $K$-step HMC can only improve the relaxation time of Theorem 4 by roughly $K^2$, showing that

---

[3]The warmness of a distribution is the worst-case ratio between measures it and the stationary assign to a set.

[4]As discussed, it is currently unknown how to obtain such a warm start generically.

to truly go beyond a $\kappa d$ relaxation time by a $d^{\Omega(1)}$ factor, the step size must scale polynomially with the dimension. Our mixing lower bound technique in Theorem 3 does not directly extend to give a complementary lower bound for Theorem 4 for all $K$, but we defer this to interesting future work.

## 1.2 Prior work

Sampling from well-conditioned distributions (as well those with stronger higher derivative bounds) is an extremely active and rich research area, so for brevity we focus on discussing two types of related work in this section: upper bounds for the MALA and HMC Markov chains, and lower bounds for sampling and related problems. We defer to e.g. [Dal17, CCBJ18, DCWY18, DR18, DM19, DMM19, CDWY19, CV19, SL19, MMW+19, LST20a, LST20b, CLA+20] and references therein for a more complete account on progress on well-conditioned sampling.

**Theory of MALA and HMC.** MALA was originally proposed in [Bes94], and subsequently its practical and theoretical performance has received extensive treatment in the literature (cf. [PSC+15]). Several analyses related to the well-conditioned setting we study predate [DCWY18], e.g. [RT96, BRH12], but they consider more restricted settings or do not state explicit dependences on $\kappa$ and $d$.

Recently, a line of work has obtained a sequence of upper bounds on the mixing of MALA. First, [DCWY18] demonstrated a mixing time of $\widetilde{O}(\kappa d + \kappa^{1.5}\sqrt{d})$ from a polynomially warm start, and the same authors later proved the same bound under an explicit exponentially warm start [CDWY19]. Later, [LST20a] demonstrated that under an appropriate averaging scheme, the mixing time could be improved to $\widetilde{O}(\kappa d)$ with no low-order dependence. Finally, [CLA+20] recently demonstrated that from a polynomially warm start, MALA mixes in time $\widetilde{O}(\text{poly}(\kappa)\sqrt{d})$ for general $\kappa$-conditioned distributions and in time $\widetilde{O}(\text{poly}(\kappa)\sqrt[3]{d})$ for Gaussians, and posed the open question of attaining similar bounds from an explicit (exponentially) warm start, a primary motivation for our exploration.

HMC can be viewed as a multi-step generalization of MALA, with two parameters (a step size $\eta$ and a step count $K$); when $K = 1$, it matches MALA exactly. For larger $K$, the algorithm simulates the (continuous-time) Hamiltonian dynamics with respect to the potential $f(x) + \frac{1}{2}\|v\|_2^2$ where $f$ is the target's negative log-density and $v$ is an auxiliary "velocity;" intuitively, larger $K$ leads to more faithful discretizations. However, there are few explicit analyses of the (Metropolis-adjusted) HMC algorithm on well-conditioned distributions.[5] To our knowledge, the only upper bound for the mixing of HMC stronger than known analyses of its specialization MALA is by [CDWY19], which gave a suite of bounds balancing three parameters: the conditioning $\kappa$, the dimension $d$, and the *Hessian Lipschitz parameter $L_H$*, under an additional bounded third derivatives assumption. Supposing $L_H$ is polynomially bounded by the smoothness $L$, they demonstrate that HMC can sometimes achieve sublinear dependence on $d$ in number of gradient queries, where the improvement depends on $\kappa$ and $d$ (e.g. if $\kappa \in [d^{\frac{1}{3}}, d^{\frac{2}{3}}]$ and $L_H \leq L^{1.5}$, $\kappa d^{\frac{11}{12}}$ gradients suffice). This prompts the question: can HMC attain complexity *independent* of $d$, assuming higher derivative bounds, from an explicit warm start? Theorem 4 answers this negatively (at least in terms of relaxation time) using an exponentially-sized bad set; moreover, our hard distribution is a Gaussian, with vanishing derivatives of order $\geq 3$.

**Lower bounds for sampling.** The bounds most closely relevant to those in this paper are given by [LST20a], who showed that the MALA step size must scale inversely in $\kappa$ for the chain to have a constant chance of moving, and [CLA+20], who showed it must scale as $d^{-\frac{1}{2}}$. Theorem 2 matches or improves both bounds simultaneously while giving an explicit hard distribution and $\exp(-d)$-sized bad set. Moreover, both [LST20a, CLA+20] gave strictly spectral lower bounds, complemented by our Theorem 3, a mixing time lower bound. We briefly mention several additional lower bound results in sampling-adjacent literature, related to this work. Recently, [CLW20] exhibited an information-theoretic lower bound on unadjusted discretizations simulating the *underdamped Langevin dynamics*, whose dimension dependence matches the upper bound of [SL19] (leaving the precise $\kappa$ dependence open). Finally, [GLL20] and [CBL20] give information-theoretic lower bounds for estimating normalizing constants of well-conditioned distributions and the number of stochastic gradient queries required by first-order sampling under noisy gradient access respectively.

---

[5]There has been considerably more exploration of the unadjusted variant [MV18, MS19, BE21], which typically obtain mixing guarantees scaling polynomially in the inverse accuracy (as opposed to polylogarithmic).

## 2 Preliminaries

**General notation.** For $d \in \mathbb{N}$ we let $[d] := \{i \in \mathbb{N} \mid 1 \leq i \leq d\}$. For positive semidefinite $\mathbf{A}$, $\|\cdot\|_{\mathbf{A}}$ is its induced seminorm $\|x\|_{\mathbf{A}} = \sqrt{x^{\top}\mathbf{A}x}$. We use $\|\cdot\|_p$ to denote the $\ell_p$ norm for $p \geq 1$, and $\|\cdot\|_{\infty}$ is the maximum entry absolute value. We let $\mathcal{N}(\mu, \boldsymbol{\Sigma})$ denote the multivariate Gaussian with mean $\mu \in \mathbb{R}^d$ and covariance $\boldsymbol{\Sigma} \in \mathbb{R}^{d \times d}$. We let $\mathbf{I} \in \mathbb{R}^{d \times d}$ denote the identity matrix when dimensions are clear, and $\preceq$ is the Loewner order on the positive semidefinite cone. We say twice-differentiable $f : \mathbb{R}^d \to \mathbb{R}$ is $L$-smooth and $\mu$-strongly convex for $0 \leq \mu \leq L$ if $\mu\mathbf{I} \preceq \nabla^2 f(x) \preceq L\mathbf{I}$ for all $x \in \mathbb{R}^d$. For any $x, y \in \mathbb{R}^d$, this implies $\|\nabla f(x) - \nabla f(y)\|_2 \leq L\|x - y\|_2$, and that $f$ satisfies the quadratic bounds $f(x) + \langle \nabla f(x), y - x \rangle + \frac{\mu}{2}\|y - x\|_2^2 \leq f(y) \leq f(x) + \langle \nabla f(x), y - x \rangle + \frac{L}{2}\|y - x\|_2^2$. The condition number of such $f$ is $\kappa := \frac{L}{\mu}$. We assume $\kappa$ is at least a constant for convenience; a lower bound of 10 suffices for all our results. For $A \subseteq \mathbb{R}^d$, $A^c$ is its complement and $\pi(A) := \int_{x \in A} d\pi(x)$ is its measure under $\pi$. We say $\rho$ is $\beta$-warm with respect to $\pi$ if $\frac{d\rho}{d\pi}(x) \leq \beta$ everywhere, and define $\|\pi - \rho\|_{\mathrm{TV}} := \sup_{A \subseteq \mathbb{R}^d} \pi(A) - \rho(A)$. Finally, $\mathbb{E}_{\pi}[g] = \int g(x)d\pi(x)$ and $\mathrm{Var}_{\pi}[g] = \mathbb{E}_{\pi}[g^2] - (\mathbb{E}_{\pi}[g])^2$.

**Sampling.** Consider a Markov chain defined on $\mathbb{R}^d$ with transition kernel $\{\mathcal{T}_x\}_{x \in \mathbb{R}^d}$, so that $\int \mathcal{T}_x(y)dy = 1$ for all $x$. Further, denote the stationary distribution of the Markov chain by $\pi^*$. We define the Dirichlet form $g : \mathbb{R}^d \to \mathbb{R}$ with respect to the chain by $\mathcal{E}(g, g) = \frac{1}{2}\iint (g(x) - g(y))^2 \mathcal{T}_x(y)d\pi^*(x)dy$. The mixing of the chain is governed by its spectral gap:

$$\lambda\left(\{\mathcal{T}_x\}_{x \in \mathbb{R}^d}\right) := \inf_g \left\{ \frac{\mathcal{E}(g, g)}{\mathrm{Var}_{\pi^*}[g]} \right\}. \tag{1}$$

The relaxation time is the inverse spectral gap. Finally, we recall the definition of a Metropolis filter, which takes arbitrary proposal distributions $\{\mathcal{P}_x\}_{x \in \mathbb{R}^d}$ and defines a reversible chain with stationary distribution $\pi^*$. The Metropolis filtered chain has transition distributions $\{\mathcal{T}_x\}_{x \in \mathbb{R}^d}$ defined by

$$\mathcal{T}_x(y) := \mathcal{P}_x(y) \min\left(1, \frac{d\pi^*(y)\mathcal{P}_y(x)}{d\pi^*(x)\mathcal{P}_x(y)}\right) \text{ for all } y \neq x. \tag{2}$$

Whenever the proposal is rejected by the modified distributions above, the chain does not move.

**MALA.** We formally define the Metropolis-adjusted Langevin algorithm (MALA). Fix a distribution $\pi$ on $\mathbb{R}^d$, with density $\frac{d\pi}{dx}(x) = \exp(-f(x))$, and suppose $f$ is twice-differentiable. MALA is defined by performing a Euler discretization of the Langevin dynamics up to time $h > 0$, and then applying a Metropolis filter: define the proposal distribution at a point $x$ by $\mathcal{P}_x := \mathcal{N}(x - h\nabla f(x), 2h\mathbf{I})$. We obtain the MALA transition distribution by applying the definition (2), which yields

$$\mathcal{T}_x(y) \propto \exp\left(-\frac{\|y - (x - h\nabla f(x))\|_2^2}{4h}\right) \min\left(1, \frac{\exp\left(-f(y) - \frac{\|x - (y - h\nabla f(y))\|_2^2}{4h}\right)}{\exp\left(-f(x) - \frac{\|y - (x - h\nabla f(x))\|_2^2}{4h}\right)}\right). \tag{3}$$

**HMC.** The Metropolized HMC algorithm is governed by two parameters, a step size $\eta > 0$ and a step count $K \in \mathbb{N}$. From iterate $x$, HMC performs the following updates from $x_0 \leftarrow x$, $v_0 \sim \mathcal{N}(0, \mathbf{I})$.

1. For $0 \leq k < K$:

   (a) $v_{k+\frac{1}{2}} \leftarrow v_k - \frac{\eta}{2}\nabla f(x_k)$, $x_{k+1} \leftarrow x_k + \eta v_{k+\frac{1}{2}}$, $v_{k+1} \leftarrow v_k - \frac{\eta}{2}\nabla f(x_{k+1})$

Each loop is known in the literature as a "leapfrog" step, and discretizes Hamilton's equations for $\mathcal{H}(x, v) := f(x) + \frac{1}{2}\|v\|_2^2$; for additional background, we refer the reader to [CDWY19]. Metropolized HMC performs the above algorithm from a point $x$, and accepts $x_K$ with probability

$$\min\left\{1, \frac{\exp\left(-\mathcal{H}(x_K, v_K)\right)}{\exp\left(-\mathcal{H}(x_0, v_0)\right)}\right\}. \tag{4}$$

**Supplementary material.** Due to space constraints in this abridged version, we defer additional exposition and many proofs to the full version of this paper, contained in the supplement.

# 3 Lower bounds for MALA

In this section, we give the main constructions and technical tools we develop in proving Theorem 3. En route, we also summarize our proofs of Theorems 1 and 2. Our starting point is the observation (also made in [CLA$^+$20]) that for a MALA step size $h$, the spectral gap of the MALA Markov chain scales no better than $O(h + h^2)$, witnessed by a simple one-dimensional Gaussian.

**Lemma 1** (Corollary 1, supplement). *The spectral gap of MALA for sampling from the density* $\propto \exp(-f)$*, where* $f(x) = \frac{1}{2}x_1^2 + f_{-1}(x_{-1})$ *(* $x_{-1}$ *drops the first coordinate), is* $O(h + h^2)$*.*

Following Lemma 1, our strategy for proving Theorems 1 and 2 is to show a dichotomy on the choice of step size: either $h$ is so large such that we can construct an $\exp(d)$-warm start where the chain is extremely unlikely to move (e.g. the step almost always is filtered), or it is small enough to imply a poor spectral gap directly by Lemma 1. In the former case (large step size), it is enough to give an $\exp(-d)$-sized set according to the stationary measure, from which the chain rejects with probability $1 - \text{poly}(d^{-1})$: this implies a *conductance* lower bound with the given witness set, which yields a spectral gap bound by the well-known Cheeger's inequality [Che69]. We use this argument to rule out different step size ranges with explicit functions and witness sets.

**Ruling out** $h = \omega(\sqrt{\log d} \cdot \frac{1}{\kappa\sqrt{d}})$**: proof of Theorem 1.** In the Gaussian case, we achieve this by explicitly characterizing the rejection probability. Our hard function is the simple quadratic $f_{\text{hq}}(x) = \frac{1}{2}x_1^2 + \frac{\kappa}{2}\|x_{-1}\|_2^2$. Straightforward calculations then yield the following bound on the log-acceptance probability in (3) (ignoring the first coordinate for simplicity, as it does not dominate).

**Lemma 2** (Lemma 3, supplement). *For fixed* $x \in \mathbb{R}^d$ *and* $y \sim \mathcal{P}_x$ *according to MALA, with high probability,* $f_{\text{hq}}(x) - f_{\text{hq}}(y) + \frac{1}{4h}\left(\|y - (x - h\nabla f_{\text{hq}}(x))\|_2^2 - \|x - (y - h\nabla f_{\text{hq}}(y))\|_2^2\right)$ *is at most*

$$\Theta\left(h\kappa^2\left((2h\kappa - h^2\kappa^2)\|x\|_2^2 - 2hd\right)\right) + (2h)^{1.5}\kappa^2|1 - h\kappa|\|x\|_2\sqrt{\log d}. \tag{5}$$

From this point, we observe that as long as $\|x\|_2^2$ is at least a constant factor away from its expectation ($\approx \frac{d}{\kappa}$), the dominant behavior of (5) scales as $-h^2\kappa^2 d$. We pick the "small ball" warm start $\|x\|_2^2 \leq \frac{d}{2\kappa}$, which captures at least $\exp(-d)$ of the mass of $\pi^*$. These calculations imply that from this warm start, the rejection probability is $\exp(-\Omega(h^2\kappa^2 d))$, which implies any $h = \omega(\sqrt{\log d} \cdot \frac{1}{\kappa\sqrt{d}})$ cannot leave the set with high probability; combined with Lemma 1, this proves Theorem 1.

**Ruling out** $h = \omega(\log d \cdot \frac{1}{\kappa d})$**: proof of Theorem 2.** We now move onto the general well-conditioned setting. As a thought experiment, we observe the upper bound analysis of [DCWY18] for MALA has a $d$ dependence which is bottlenecked by the *noise term* only. Even the "Metropolized random walk" where the proposal is simply $x_g \leftarrow x + \sqrt{2h}g$ has upper bound analyses scaling linearly in $d$. Thus, it is natural to study the effect of the noise, and construct a hard distribution based around it.

We first formalize this intuition, demonstrate that for step sizes not ruled out by Theorem 1, all terms in the rejection probability other than those due to the effect of the noise are low-order.

**Lemma 3** (Lemma 10, supplement). *For fixed* $x \in \mathbb{R}^d$ *with* $\|\nabla f(x)\|_2 = O(\sqrt{\kappa d})$*, with high probability over* $g \sim \mathcal{N}(0, \mathbf{I})$*, and* $x_g \leftarrow x + \sqrt{2h}g$*,* $y \leftarrow x_g - h\nabla f(x)$ *according to MALA,*

$$f(x) - f(y) + \frac{1}{4h}\left(\|y - (x - h\nabla f(x))\|_2^2 - \|x - (y - h\nabla f(y))\|_2^2\right) \leq o(h\kappa d)$$
$$+ f(x) - f(x_g) + \frac{1}{4h}\left(\|x_g - (x - h\nabla f(x))\|_2^2 - \|x - (x_g - h\nabla f(x_g))\|_2^2\right).$$

In other words, to show a high-probability $-\Omega(h\kappa d)$ upper bound on the log-acceptance, it suffices to understand the second line of the above display (since other terms do not dominate). Moreover, the effect of the noise is coordinatewise separable (since $\mathcal{N}(0, \mathbf{I})$ is a product distribution). Thus, it suffices to show a hard one-dimensional density where the log-acceptance has expectation $-\Omega(h\kappa)$, and apply sub-Gaussian concentration to show a product distribution very likely obtains $-\Omega(h\kappa d)$.

At this point, we reduce to the following self-contained problem: let $x \in \mathbb{R}$, let $\pi^* \propto \exp(-f_{\text{1d}})$ be one-dimensional with second derivative $\leq \kappa$, and let $x_g = x + \sqrt{2h}g$ for $g \sim \mathcal{N}(0, 1)$. We wish to

construct $f_{1d}$ such that for $x$ in a constant probability region over $\exp(-f_{1d})$ (the "bad set"),

$$\mathbb{E}_{g\sim\mathcal{N}(0,1)}\left[-f_{1d}(x_g) + f_{1d}(x) - \frac{1}{2}\langle x - x_g, f'_{1d}(x) + f'_{1d}(x_g)\rangle\right] = -\Omega(h\kappa), \qquad (6)$$

where the contents of the expectation in (6) are the log-rejection probability along one coordinate by a straightforward calculation. By forming a product distribution using $f_{1d}$ as a building block, and combining with the remaining low-order terms due to the drift $\nabla f(x)$, we attain an $\exp(-d)$-sized region where the rejection probability is $\exp(-\Omega(h\kappa d))$, completing Theorem 2.

It remains to construct such a hard $f_{1d}$. The calculation

$$-f_{1d}(x_g) + f_{1d}(x) - \frac{1}{2}\langle x - x_g, f'_{1d}(x) + f'_{1d}(x_g)\rangle = -2h\int_0^1 \left(\frac{1}{2} - s\right)g^2 f''_{1d}(x + s(x_g - s))ds$$

suggests the following approach: since $f''_{1d} > 0$ always, and the integral puts negative mass on $s \in [0, \frac{1}{2})$ and positive on $s \in (\frac{1}{2}, 1]$, we want our bad set to have large $f''_{1d}$, but most moves $g$ to result in a smaller $f''_{1d}$. Our construction patterns this intuition: we choose[6]

$$f_{1d}(x) = \frac{\kappa}{3}x^2 - \frac{\kappa h}{3}\cos\frac{x}{\sqrt{h}} \implies f''_{1d}(x) = \frac{2\kappa}{3} + \frac{\kappa}{3}\cos\frac{x}{\sqrt{h}},$$

such that our bad set is when $\cos\frac{x}{\sqrt{h}}$ is relatively large (which occurs with probability $\to \frac{1}{2}$ for small $h$ in one dimension). The period of our construction scales with $\sqrt{h}$, so that most moves $\sqrt{2hg}$ of size $O(\sqrt{h})$ will "skip a period" and hence hit a region with small second derivative, satisfying (6).

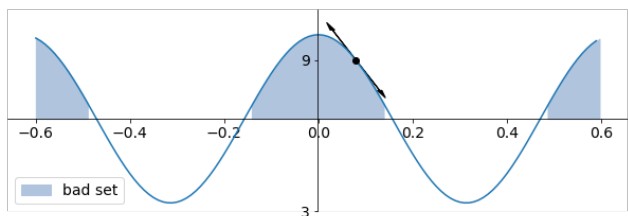

Figure 1: Second derivative of our hard function $f_{1d}$, $\kappa = 10$, $h = 0.01$. Starting from inside the hard region, on average over $g \sim \mathcal{N}(0, \mathbf{I})$, a move by $\sqrt{2hg}$ decreases the second derivative.

We demonstrate our hard function $f_{\text{hard}}$, a product of $d - 1$ copies of $f_{1d}$ and a one-dimensional Gaussian, satisfies the following bound. Combining it with Lemmas 3 and 1, and using the hard set of points whose gradient is $O(\sqrt{\kappa d})$ and whose coordinatewise cosine is $\Omega(1)$ (which we show has measure $\geq \exp(-d)$, since it coordinatewise captures roughly half the mass), yields Theorem 2.

**Lemma 4** (Lemmas 7-8, supplement). *For fixed $x \in \mathbb{R}^d$ with $\cos\frac{x_i}{\sqrt{h}} > 0.1$ coordinatewise for $2 \leq i \leq d$, with high probability over $g \sim \mathcal{N}(0, \mathbf{I})$ and $x_g \leftarrow x + \sqrt{2hg}$,*

$$f(x) - f(x_g) + \frac{1}{4h}\left(\|x_g - (x - h\nabla f(x))\|_2^2 - \|x - (x_g - h\nabla f(x_g))\|_2^2\right) = -\Omega(h\kappa d).$$

**From relaxation time to mixing time: proof of Theorem 3.** We now turn our attention to proving Theorem 3, our main mixing time lower bound. Implicitly in the above discussion, we already proved such a lower bound for $h = \omega(\log d \cdot \frac{1}{\kappa d})$: we use either the cosine construction $f_{\text{hard}}$ or the Gaussian construction $f_{\text{hq}}$ for relevant ranges of $h$, along with their $\exp(-d)$-sized hard sets, to show the chain cannot leave the starting set in polynomially many steps within these step size ranges.

We consider the remaining range, $h = O(\log d \cdot \frac{1}{\kappa d})$. Intuitively, our goal is to show that it takes roughly $\frac{1}{h}$ steps to explore the whole space, a simple goal in the absence of filtering which becomes problematic when accounting for the filter. Because it is correlated between coordinates, it is hard to reason about the effect on any particular coordinate, and moreover Gaussian concentration does not

---

[6]We note [CLA$^+$20] also used a (different, but similar) cosine-based construction for their lower bound.

directly apply when the acceptances are not independent of the choice of noise. The question here then becomes: how do we decouple the randomness of the process from the randomness of the filter?

We accomplish this decoupling in two steps. First, we analyze the effect of an "idealized" MALA with no filter, where with high probability over the randomly sampled Gaussian vectors, it takes roughly $\frac{1}{h \log d} \approx \frac{\kappa d}{\log^2 d}$ steps to move $\|x\|_2$ by a constant. It remains to account for the filter.

Our main observation is that for the only remaining range $h = O(\log d \cdot \frac{1}{\kappa d})$, we know such a small step size (scaling with $d$ rather than $\sqrt{d}$) is overkill for rapid mixing on Gaussians, which only require a step size $\sqrt{d}$ larger. Thus, for this range we choose our hard density to be the simple Gaussian $\mathcal{N}(0, \mathbf{I})$. We show that with constant probability, in $\frac{\kappa d}{\log^2 d}$ steps the true MALA chain *never rejects*. This allows us to couple MALA with our idealized chain, proving a mixing time lower bound.

Formally, our main technical lemma to this effect is stated in the following. By initializing at the $\exp(d)$-warm start $\|x\|_2^2 \leq \frac{1}{2}d$ (the "small ball" from Theorem 1's proof), and combining with our lower bounds on the other ranges of $h$, we conclude our proof of Theorem 3.

**Lemma 5** (Lemma 11, supplement). *Consider iterating the MALA Markov chain with $h = O(\log d \cdot \frac{1}{\kappa d})$ on $\mathcal{N}(0, \mathbf{I})$ for $T$ iterations, with starting distribution the marginal on $\|x\|_2^2 \leq \frac{1}{2}d$. With probability at least $\frac{99}{100}$, both of the following events occur for $T = o\left(\frac{\kappa d}{\log^2 d}\right)$:*

1. *Throughout the Markov chain with iterates $\{x_t\}_{0 \leq t < T}$, we always have $\|x_t\|_2 \leq 0.9\sqrt{d}$.*

2. *Throughout the Markov chain, the Metropolis filter never rejected.*

## 4 Lower bounds for HMC

In this section, we present our main HMC lower bound, Theorem 4. We recall that HMC has two parameters: a number of steps $K \geq 2$ (as the $K = 1$ case is handled by MALA), and a step size $\eta > 0$. Throughout, we focus our attention on the case of a quadratic $f(x) = \frac{1}{2}x^\top \mathbf{A}x$. Without loss of generality (by rotational invariance of HMC), we assume $\mathbf{A} = \mathbf{diag}(\lambda)$ for $1 \leq \lambda \leq \kappa$ entrywise.

**Structure of HMC: a detour to Chebyshev polynomials.** The starting point of our lower bound construction is a novel characterization of HMC's behavior on quadratics via Chebyshev polynomials. By directly analyzing the recurrences generating the HMC iterates (cf. Section 2), we show the following relationship between some iterate $x_k$ and the initial position and velocity $(x_0, v_0)$.

**Lemma 6** (Lemma 13, supplement). *For $f(x) = \frac{1}{2}x^\top \mathbf{diag}(\lambda) x$, iterates $\{x_k\}_{0 \leq k \leq K}$ satisfy*

$$[x_k]_i = p_k(\eta^2 \lambda_i)[x_0]_i + \eta q_k(\eta^2 \lambda_i)[v_0]_i,$$

*where* $p_k(z) := \sum_{0 \leq j \leq k} (-1)^j \cdot \frac{k}{k+j} \cdot \binom{k+j}{2j} z^j$, $q_k(z) := \sum_{0 \leq j \leq k-1} (-1)^j \cdot \binom{k+j}{2j+1} z^j$.

We further observe that the polynomials $p_k$ and $q_k$ are related to the classical Chebyshev polynomials. Let $T_k$ be the $k^{\text{th}}$ Chebyshev polynomial of the first kind, and let $U_k$ the $k^{\text{th}}$ Chebyshev polynomial of the second kind. These polynomial families are very well-studied due to their extremal properties, and indeed their relationship to the *acceleration* phenomenon in first-order optimization has been known for some time [Har13, Bac19]. Roughly, these polynomials oscillate between $\pm 1$ in the range $[-1, 1]$ (which contains all their zeroes), and outside this range they rapidly explode.

We observe that the Chebyshev polynomials are related to the HMC coefficient polynomials:

$$p_k(z) = T_k\left(1 - \frac{z}{2}\right), \ q_k(z) = U_{k-1}\left(1 - \frac{z}{2}\right).$$

Magically, the extremal points in $[-1, 1]$ where $T_k$ is $\pm 1$ are *exactly the same* as the zeroes of $U_{k-1}$. Combining with Lemma 6, this yields the somewhat surprising conclusion that there are choices of step size $\eta$ where the chain cannot move in a coordinate, oscillating between $\pm[x_0]_i$ *regardless* of the choice of $v_0$! We use this observation heavily in our lower bound construction to rule out $\eta$ ranges.

**Ruling out the Chebyshev extremal region $\eta^2 \geq \frac{\pi^2}{\kappa K^2}$.** We first restrict to the intermediate range $1 \geq \eta^2 \geq \frac{\pi^2}{\kappa K^2}$. We show (Proposition 4, supplement) $\exists j \in [K-1]$ such that $1 \leq \lambda_j \leq \kappa$, for

$$\lambda_j := \frac{2\left(1 - \cos\left(\frac{j\pi}{K}\right)\right)}{\eta^2}, \text{ for } 1 \geq \eta^2 \geq \frac{\pi^2}{\kappa K^2}.$$

The reason we choose this definition is that the $\{\lambda_j\}_{j \in [K-1]}$ are precisely the points where (for $z := \eta^2 \lambda_j$) $T_K\left(1 - \frac{z}{2}\right) = \pm 1$ and $U_{k-1}\left(1 - \frac{z}{2}\right) = 0$. Hence, by picking a hard quadratic function with at least one eigenvalue at the specified $\lambda_j$, we can force the chain to never leave any symmetric set in that coordinate, regardless of the random velocity chosen and number of steps $K$.

Finally, for $\eta^2 \geq 1$, we observe the hard function $f(x) = \frac{\kappa}{2}\|x\|_2^2$ is a simple counterexample. In particular, for $\lambda = \kappa$, $\eta^2 \geq 1$, and $z = \eta^2 \lambda$, we see that $p_k(z)$ blows up due to the behavior of Chebyshev polynomials outside $[-1, 1]$, as $1 - \frac{z}{2}$ falls outside this range. We use this to argue the acceptance probability from any starting point and any number of steps is $\exp(-\Omega(d))$.

**Stronger $\eta$ bounds under a constant gap.** The Chebyshev polynomial argument demonstrates that to avoid this extremal behavior, we should choose $\eta^2 = O(\frac{1}{\kappa K^2})$. We now give a lower bound argument under the stronger assumption that $\eta^2 \leq \frac{1}{\kappa K^2}$, with the understanding there is a constant gap between this assumption and the region ruled out by the Chebyshev argument.

The main observation (Lemma 14, supplement) is that under this assumption, the coefficients of $p_k$ and $q_k$ supplied in Lemma 6 decay rapidly. In particular, for polynomial $p_K$, the coefficient ratios are

$$\frac{(-\eta^2\kappa)^{j+1}\left(\frac{K}{K+j+1}\right)\binom{K+j}{2j}}{(-\eta^2\kappa)^j\left(\frac{K}{K+j}\right)\binom{K+j}{2j}} = (-\eta^2\kappa)\frac{(K+j)(K-j)}{(2j+2)(2j+1)} \in [-0.1, 0],$$

assuming $\eta^2\kappa \leq \frac{1}{K^2}$. This implies that both the polynomials $p_K$ and $q_K$ are governed by their first terms, up to constants, and thus we can apply the proof techniques of Theorem 1. Indeed, we demonstrate that up to a $K$ factor, in this step size range HMC behaves just like MALA, and thus its spectral gap only loses a $K$ (which balances the $K$ gradient queries per iteration).

**Removing the constant gap.** The final obstacle in our HMC construction is to close the gap between the $\eta^2 \leq \frac{1}{\kappa K^2}$ required by our geometric argument, and the $\eta^2 \leq \frac{\pi^2}{\kappa K^2}$ implied by behavior of Chebyshev polynomials. We bypass this by taking a hard quadratic $f_{\text{hqc}}$ which weights two coordinates by 1 and $\kappa$, and the remaining coordinates by $\frac{\kappa}{\pi^2}$. On $d - 2$ coordinates, we obtain

$$\left(-\eta^2\frac{\kappa}{\pi^2}\right)\frac{(K+j)(K-j)}{(2j+2)(2j+1)} \in [-0.1, 0] \text{ for all } \eta^2 \leq \frac{\pi^2}{\kappa K^2},$$

showing the relevant polynomials rapidly converge. This is the dominant behavior, as eventually the denominators of these ratios outweighs any constant gap, and thus the $\kappa$ coordinate can only hurt us by so much, concluding the proof of Theorem 4.

## 5 Conclusion

We presented relaxation time lower bounds for the MALA and HMC Markov chains at every step size and scale, as well as a mixing time bound for MALA from an exponentially warm start. We highlight in this section a number of unexplored directions left open by our work, beyond direct strengthenings of our results, which we find interesting and defer to a future exploration.

**Variable or random step sizes.** The lower bounds of this paper were for MALA and HMC Markov chains with a *fixed step size*. For variable step sizes which take e.g. values in a bounded multiplicative range, we believe our arguments can be modified to also give relaxation time lower bounds for the resulting Markov chains. However, the arguments of Section 4 (our HMC lower bound) are particularly brittle to large multiplicative ranges of candidate step sizes, because they rely on the locations of Chebyshev polynomial zeroes, which only occur in a bounded range. From an algorithm design perspective, this suggests that adaptively or randomly choosing step size ranges may be effective in improving the performance of HMC. Such a result would also give theoretical justification to the No-U-Turn sampler of [HG14], a common HMC alternative in practice. We state as an

explicit open problem: can one obtain improved upper bounds, such as a $\sqrt{\kappa}$ dependence or a dimension-independent rate, for example by using variations of these strategies (variable step sizes)?

**Necessity of $\kappa$ lower bound.** All of our witness sets throughout the paper are $\exp(-d)$ sized. It was observed in [DCWY18] that it is possible to construct a starting distribution with warmness arbitrarily close to $\sqrt{\kappa}^d$; the marginal restriction of our witness set obtains this bound for all $\kappa \geq e^2$. However, recently [LST20b] proposed a *proximal point reduction* for sampling, showing (for mixing bounds scaling at least linearly in $\kappa$) it suffices to sample a small number of regularized distributions, with conditioning arbitrarily close to 1. Adjusting constants, we can modify our Gaussian lower bounds (Theorems 1 and 4) to have witness sets with measure $c^d$ for $c$ arbitrarily close to 1. However, our witness set for the family of hard non-Gaussian distributions encounters a natural barrier at measure $2^d$, as it is sign-restricted by the cosine function. We find it interesting to see if a stronger construction rules out existing warm starts for all $\kappa \geq 1$, or if an upper bound can take advantage of the [LST20b] reduction to obtain improved dependences on dimension.

## Acknowledgments

We would like to thank Santosh Vempala for numerous helpful conversations, pointers to the literature, and writing suggestions throughout the course of this project.

YL and RS are supported by NSF awards CCF-1749609, DMS-1839116, and DMS-2023166, a Microsoft Research Faculty Fellowship, a Sloan Research Fellowship, and a Packard Fellowship. KT is supported by NSF Grant CCF-1955039 and the Alfred P. Sloan Foundation.

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
