# Lower Bounds on Metropolized Sampling Methods
# for Well-Conditioned Distributions

Yin Tat Lee[*]     Ruoqi Shen[†]     Kevin Tian[‡]

## Abstract

We give lower bounds on the performance of two of the most popular sampling methods in practice, the Metropolis-adjusted Langevin algorithm (MALA) and multi-step Hamiltonian Monte Carlo (HMC) with a leapfrog integrator, when applied to well-conditioned distributions. Our main result is a nearly-tight lower bound of $\widetilde{\Omega}(\kappa d)$ on the mixing time of MALA from an exponentially warm start, matching a line of algorithmic results [DCWY18, CDWY19, LST20a] up to logarithmic factors and answering an open question of [CLA⁺20]. We also show that a polynomial dependence on dimension is necessary for the relaxation time of HMC under any number of leapfrog steps, and bound the gains achievable by changing the step count. Our HMC analysis draws upon a novel connection between leapfrog integration and Chebyshev polynomials, which may be of independent interest.

---

[*]University of Washington and Microsoft Research, `yintat@uw.edu`
[†]University of Washington, `shenr3@cs.washington.edu`
[‡]Stanford University, `kjtian@stanford.edu`

# Contents

# 1 Introduction

Sampling from a continuous distribution in high dimensions is a fundamental problem in algorithm design. As sampling serves as a key subroutine in a variety of tasks in machine learning [AdFDJ03], statistical methods [RC99], and scientific computing [Liu01], it is an important undertaking to understand the complexity of sampling from families of distributions arising in applications.

The more restricted problem of sampling from a particular family of distributions, which we call "well-conditioned distributions," has garnered a substantial amount of recent research effort from the algorithmic learning and statistics communities. This specific family is interesting for a number of reasons. First of all, it is *practically relevant*: Bayesian methods have found increasing use in machine learning applications [Bar12], and many distributions arising from these methods are well-conditioned, such as multivariate Gaussians, mixture models with small separation, and densities arising from Bayesian logistic regression with a Gaussian prior [DCWY18]. Moreover, for several of the most widely-used sampler implementations in popular packages [Aba16, CGH+17], such as the Metropolis-adjusted Langevin algorithm (MALA) and Hamiltonian Monte Carlo (HMC), the target density having a small condition number is in some sense a *minimal assumption for known provable guarantees* (discussed more thoroughly in Section 1.3, when we survey prior work).

Finally, the highly-documented success of first-order (gradient-based) methods in optimization [Bec17], which are particularly favorable in the well-conditioned setting, has driven a recent interest in connections between optimization and sampling. Exploring this connection has been highly fruitful: since seminal work of [JKO98], which demonstrated that the continuous-time Langevin dynamics which MALA and HMC discretize has an interpretation as gradient descent on density space, a flurry of work including [Dal17, CCBJ18, DCWY18, DR18, DM19, DMM19, CDWY19, CV19, SL19, MMW+19, LST20a, LST20b, CLA+20] has obtained improved upper bounds for the mixing of various discretizations of the Langevin dynamics for sampling from well-conditioned densities. Many of these works have drawn inspiration from techniques from first-order optimization.

On the other hand, demonstrating *lower bounds* on the complexity of sampling tasks (in the well-conditioned regime or otherwise) has proven to be a remarkably challenging problem. To our knowledge, there are very few unconditional lower bounds for sampling tasks (i.e. the complexity of sampling from a family of distributions under some query model). This is in stark contrast to the theory of optimization, where there are matching upper and lower bounds for a variety of fundamental tasks and query models, such as optimization of a convex function under first-order oracle access [Nes03]. This gap in the development of the algorithmic theory of sampling is the primary motivation for our work, wherein we aim to answer the following more restricted question.

> *What is the complexity of the popular sampling methods, MALA and HMC,*
> *for sampling well-conditioned distributions?*

The problem we study is still less general than *unconditional query lower bounds* for sampling, in that our lower bounds are *algorithm-specific*; we characterize the performance of particular algorithms for sampling a distribution family. However, we believe asking this question, and developing an understanding of it, is an important first step towards a theory of complexity for sampling. On the one hand, lower bounds for specific algorithms highlight weaknesses in their performance, pinpointing their shortcomings in attaining faster rates. This is useful from an algorithm design perspective, as it clarifies what the key technical barriers are to overcome. On the other hand, the hard instances which arise in designing lower bounds may have important structural properties which pave the way to stronger and more general (i.e. *algorithm-agnostic*) lower bounds.

For these reasons, in this work we focus on characterizing the complexity of the MALA and HMC algorithms (see Sections 2.3 and 2.4 for algorithm definitions), which are often the samplers of choice in practice, by lower bounding their performance when they are used to sample from densities proportional to $\exp(-f(x))$, where $f : \mathbb{R}^d \to \mathbb{R}$ has a finite condition number. In particular, $f$ is said to have a condition number of $\kappa < \infty$ if it is $L$-smooth and $\mu$-strongly convex (has second derivatives in all directions in the range $[\mu, L]$), where $\kappa = \frac{L}{\mu}$. We will also overload this terminology and say the density itself has condition number $\kappa$. We call such a density (with finite $\kappa$) "well-conditioned." Finally, we explicitly assume throughout that $\kappa = O(d^4)$, as otherwise in light of our lower bounds the general-purpose logconcave samplers of [LV07, JLLV20, Che21] are preferable.

## 1.1 Our results

Our primary contribution is a nearly-tight characterization of the performance of MALA for sampling from two high-dimensional distribution families without a warm start assumption: well-conditioned Gaussians, and the more general family of well-conditioned densities. In Sections 3 and 4, we prove the following two lower bounds on MALA's complexity, which is a one-parameter algorithm (for a given target distribution) depending only on step size. We also note that we fix a scale $[1, \kappa]$ on the eigenvalues of the function Hessian up front, because otherwise the non-scale-invariance of the step size can be exploited to give much more trivial lower bounds (cf. Appendix A).

**Theorem 1.** *For every step size, there is a target Gaussian on $\mathbb{R}^d$ whose negative log-density always has Hessian eigenvalues in $[1, \kappa]$, such that the relaxation time of MALA is $\Omega(\frac{\kappa\sqrt{d}}{\sqrt{\log d}})$.*

**Theorem 2.** *For every step size, there is a target density on $\mathbb{R}^d$ whose negative log-density always has Hessian eigenvalues in $[1, \kappa]$, such that the relaxation time of MALA is $\Omega(\frac{\kappa d}{\log d})$.*

To give more context on Theorems 1 and 2, MALA is an example of a Metropolis-adjusted Markov chain, which in every step performs updates which preserve the stationary distribution. Indeed, it can be derived by applying a Metropolis filter on the standard forward Euler discretization of the Langevin dynamics, a stochastic differential equation with stationary density $\propto \exp(-f(x))$:

$$dx_t = -\nabla f(x_t)dt + \sqrt{2}dW_t,$$

where $W_t$ is Brownian motion. Such *Metropolis-adjusted* methods typically provide total variation distance guarantees, and attain logarithmic dependence on the target accuracy.[1] The mixing of such chains is governed by their relaxation time, also known as the inverse *spectral gap* (the difference between 1 and the second-largest eigenvalue of the Markov chain transition operator).

However, in the continuous-space setting, it is not always clear how to relate the relaxation time to the *mixing time*, which we define as the number of iterations it takes to reach total variation distance $\frac{1}{e}$ from the stationary distribution from a given warm start (we choose $\frac{1}{e}$ for consistency with the literature, but indeed any constant bounded away from 1 will do). There is an extensive line of research on when it is possible to relate these two quantities (see e.g. [BGL14]), but typically these arguments are tailored to properties of the specific Markov chain, causing relaxation time lower bounds to not be entirely satisfactory in some cases. We thus complement Theorems 1 and 2 with a *mixing time* lower bound from an exponentially warm start, as follows.

---

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

In Appendix B, we also give some lower bounds on how much increasing $K$ can help the performance of HMC in the in-between range $\kappa \sqrt{d}$ to $\kappa d$. In particular, we demonstrate that if $K \leq d^c$ for some constant $c \approx 0.1$, then the $K$-step HMC Markov chain can only improve the relaxation time of Theorem 4 by roughly a factor $K^2$, showing that to truly go beyond a $\kappa d$ relaxation time by more than a $d^{o(1)}$ factor, the step size must scale polynomially with the dimension (Proposition 5). We further demonstrate how to extend the mixing time lower bound of Theorem 3 in a similar manner, demonstrating formally for small $K$ that (up to logarithmic factors) the gradient query complexity of HMC cannot be improved beyond $\kappa d$ by more than roughly a $K$ factor (Proposition 6).

Our mixing lower bound technique in Theorem 3 does not directly extend to give a complementary mixing lower bound for Theorem 4 for all $K$, but we defer this to interesting future work.

## 1.2 Technical overview

In this section, we give an overview of the techniques we use to show our lower bounds. Throughout for the sake of fixing a scale, we assume the negative log-density has Hessian between $\mathbf{I}$ and $\kappa \mathbf{I}$.

**MALA.** Our starting point is the observation made in [CLA$^+$20] that for a MALA step size $h$, the spectral gap of the MALA Markov chain scales no better than $O(h + h^2)$, witnessed by a simple one-dimensional Gaussian. Thus, our strategy for proving Theorems 1 and 2 is to show a dichotomy on the choice of step size: either $h$ is so large such that we can construct an $\exp(d)$-warm start where the chain is extremely unlikely to move (e.g. the step almost always is filtered), or it is small enough to imply a poor spectral gap. In the Gaussian case, we achieve this by explicitly characterizing the rejection probability and demonstrating that choosing the "small ball" warm start where $\|x\|_2^2$ is smaller than its expectation by a constant ratio suffices to upper bound $h$.

Given the result of Theorem 1, we see that if MALA is to move at all with decent probability from an exponentially warm start, we must take $h \ll 1$, so the spectral gap in this regime is simply $O(h)$. We now move onto the more general well-conditioned setting. As a thought experiment, we note that the upper bound analyses of [DCWY18, CDWY19, LST20a] for MALA have a dimension dependence which is bottlenecked by the *noise term* only. In particular, the MALA iterates apply a filter to the move $x' \leftarrow x - h \nabla f(x) + \sqrt{2h} g$, where $g \sim \mathcal{N}(0, \mathbf{I})$ is a standard Gaussian vector. However, even for the more basic "Metropolized random walk" where the proposal is simply $x' \leftarrow x + \sqrt{2h} g$, the dimension dependence of upper bound analyses scales linearly in $d$. Thus, it is natural to study the effect of the noise, and construct a hard distribution based around it.

We first formalize this intuition, and demonstrate that for step sizes not ruled out by Theorem 1, all terms in the rejection probability calculation other than those due to the effect of the noise $g$ are low-order. Moreover, because the effect of the noise is coordinatewise separable (since $\mathcal{N}(0, \mathbf{I})$ is a product distribution), to demonstrate a $\widetilde{O}(\frac{1}{\kappa d})$ upper bound on $h$ it suffices to show a hard one-dimensional distribution where the log-rejection probability has expectation $-\Omega(h\kappa)$, and apply sub-Gaussian concentration to show a product distribution has expectation $-\Omega(h\kappa d)$.

At this point, we reduce to the following self-contained problem: let $x \in \mathbb{R}$, let $\pi^* \propto \exp(-f_{1d})$ be one-dimensional with second derivative $\leq \kappa$, and let $x_g = x + \sqrt{2h} g$ for $g \sim \mathcal{N}(0, 1)$. We wish to construct $f_{1d}$ such that for $x$ in a constant probability region over $\exp(-f_{1d})$ (the "bad set"),

$$\mathbb{E}_{g \sim \mathcal{N}(0,1)} \left[ -f_{1d}(x_g) + f_{1d}(x) - \frac{1}{2} \left\langle x - x_g, f'_{1d}(x) + f'_{1d}(x_g) \right\rangle \right] = -\Omega(h\kappa), \tag{1}$$

where the contents of the expectation in (1) are the log-rejection probability along one coordinate by a straightforward calculation. By forming a product distribution using $f_{1d}$ as a building block, and combining with the remaining low-order terms due to the drift $\nabla f(x)$, we attain an $\exp(-d)$-sized region where the rejection probability is $\exp(-\Omega(h\kappa d))$, completing Theorem 2.

It remains to construct such a hard $f_{1d}$. The calculation

$$-f_{1d}(x_g) + f_{1d}(x) - \frac{1}{2}\left\langle x - x_g, f'_{1d}(x) + f'_{1d}(x_g)\right\rangle = -2h\int_0^1\left(\frac{1}{2} - s\right)g^2 f''_{1d}(x + s(x_g - s))ds$$

suggests the following approach: because the above integral places more mass closer to the starting point, we wish to make sure our bad set has large second derivative, but most moves $g$ result in a much smaller second derivative. Our construction patterns this intuition: we choose[4]

$$f_{1d}(x) = \frac{\kappa}{3}x^2 - \frac{\kappa h}{3}\cos\frac{x}{\sqrt{h}} \implies f''_{1d}(x) = \frac{2\kappa}{3} + \frac{\kappa}{3}\cos\frac{x}{\sqrt{h}},$$

such that our bad set is when $\cos\frac{x}{\sqrt{h}}$ is relatively large (which occurs with probability $\to \frac{1}{2}$ for small $h$ in one dimension). The period of our construction scales with $\sqrt{h}$, so that most moves $\sqrt{2h}g$ of size $O(\sqrt{h})$ will "skip a period" and hence hit a region with small second derivative, satisfying (1).

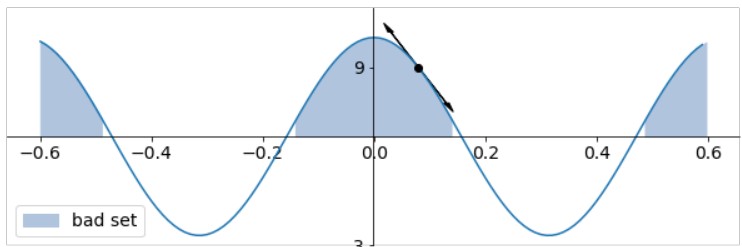

Figure 1: Second derivative of our hard function $f_{1d}$, $\kappa = 10$, $h = 0.01$. Starting from inside the hard region, on average over $g \sim \mathcal{N}(0, \mathbf{I})$, a move by $\sqrt{2h}g$ decreases the second derivative.

**HMC.** We further demonstrate that similar hard Gaussians as the one we use for MALA also place an upper bound on the step size of HMC for any number of steps $K$. Our starting point is a novel characterization of HMC iterates on Gaussians: namely, when the negative log-density is quadratic, we show that the HMC iterates implement a linear combination between the starting position and velocity, where the coefficients are given by *Chebyshev polynomials*. For step size $\eta$ of size $\Omega(\frac{1}{K\sqrt{\kappa}})$ for specific constants, we show the HMC chain begins to cycle because of the locations of the Chebyshev polynomials' zeroes, and cannot move. Moreover, for sufficiently small step size $\eta$ outside of this range, it is straightforward by examining the coefficients of Chebyshev polynomials to show that they are the same (up to constant factors) as in the MALA case, at which point our previous lower bound holds. It takes some care to modify our hard Gaussian construction to rule out all constant ranges in the $\eta \approx \frac{1}{K\sqrt{\kappa}}$ region, but by doing so we obtain Theorem 4.

We remark that the observation that HMC iterates are implicitly implementing a Chebyshev polynomial approximation appears to be unknown in the literature, and is a novel contribution of our work. We believe understanding this connection is a worthwhile endeavor, as a similar connection between polynomial approximation and first-order convex optimization has led to various

---

[4]We note [CLA+20] also used a (different, but similar) cosine-based construction for their lower bound.

interesting interpretations of Nesterov's accelerated gradient descent method [Har13, Bac19].

## 1.3 Prior work

Sampling from well-conditioned distributions (as well as distributions with more stringent bounds on higher derivatives) using discretizations of the Langevin dynamics is an extremely active and rich research area, so for brevity we focus on discussing two types of related work in this section: upper bounds for the MALA and HMC Markov chains, and lower bounds for sampling and related problems. We refer the reader to e.g. [Dal17, CCBJ18, DCWY18, DR18, DM19, DMM19, CDWY19, CV19, SL19, MMW+19, LST20a, LST20b, CLA+20] and the references therein for a more complete account on progress on the more general problem of well-conditioned sampling.

**Theoretical analyses of MALA and HMC.** MALA was originally proposed in [Bes94], and subsequently its practical and theoretical performance in different settings has received extensive treatment in the literature (cf. the survey [PSC+15]). A number of theoretical analyses related to the well-conditioned setting we study predate the work of [DCWY18], such as [RT96, BRH12], but they typically consider more restricted settings or do not state explicit dependences on $\kappa$ and $d$.

Recently, a line of work has obtained a sequence of stronger upper bounds on the mixing of MALA. First, [DCWY18] demonstrated that MALA achieves a mixing time of $\widetilde{O}(\kappa d + \kappa^{1.5}\sqrt{d})$ from a polynomially warm start, and the same set of authors later proved the same mixing time under an exponentially warm start (which can be explicitly constructed) in [CDWY19]. It was later demonstrated in [LST20a] that under an appropriate averaging scheme, the mixing time could be improved to $\widetilde{O}(\kappa d)$ from an exponentially warm start with no low-order dependence. Finally, a recent work [CLA+20] demonstrated that from a polynomially warm start, MALA mixes in time $\widetilde{O}(\text{poly}(\kappa)\sqrt{d})$ for general $\kappa$-conditioned distributions and in time $\widetilde{O}(\text{poly}(\kappa)\sqrt[3]{d})$ for $\kappa$-conditioned Gaussians, and posed the open question of attaining similar bounds from an explicit (exponentially) warm start. This latter work was a primary motivation for our exploration.

The HMC algorithm with a leapfrog integrator (which we refer to as HMC for simplicity) can be viewed as a multi-step generalization of MALA, as it has two parameters (a step size $\eta$ and a step count $K$), and when $K = 1$ the implementation matches MALA exactly. For larger $K$, the algorithm simulates the (continuous-time) Hamiltonian dynamics with respect to the potential $f(x) + \frac{1}{2}\|v\|_2^2$ where $f$ is the target's negative log-density and $v$ is an auxiliary "velocity" variable. The intuition is that larger $K$ leads to more faithful discretizations of the true dynamics.

However, there are few explicit analyses of the (Metropolis-adjusted) HMC algorithm, applied to well-conditioned distributions.[5] To our knowledge, the only theoretical upper bound for the mixing of (multi-step) HMC stronger than known analyses of its one-step specialization MALA is by [CDWY19], which gave a suite of bounds trading off three problem parameters: the conditioning $\kappa$, the dimension $d$, and the *Hessian Lipschitz parameter* $L_H$, under the additional assumption that the log-density has bounded third derivatives. Assuming that $L_H$ is polynomially bounded by the problem smoothness $L$, they demonstrate that HMC with an appropriate $K$ can sometimes achieve sublinear dependence on $d$ in number of gradient queries, where the quality of this improvement depends on $\kappa$ and $d$ (e.g. if $\kappa \in [d^{\frac{1}{3}}, d^{\frac{2}{3}}]$ and $L_H \le L^{1.5}$, $\kappa d^{\frac{11}{12}}$ gradients suffice). This prompts the question: can HMC attain query complexity *independent* of $d$, assuming higher derivative bounds, from an explicit warm start? Theorem 4 answers this negatively (at least in terms of relaxation time) using an exponentially-sized bad set; moreover, our hard distribution is a Gaussian, with all derivatives of order at least 3 vanishing.

---

[5]There has been considerably more exploration of the unadjusted variant [MV18, MS19, BE21], which typically obtain mixing guarantees scaling polynomially in the inverse accuracy (as opposed to polylogarithmic).

**Lower bounds for sampling.** The bounds most closely relevant to those in this paper are given by [LST20a], who showed that the step size of MALA must scale inversely in $\kappa$ for the chain to have a constant chance of moving, and [CLA$^+$20], who showed that the step size must scale as $d^{-\frac{1}{2}}$. Theorem 2 matches or improves both bounds simultaneously, proving that up to logarithmic factors the relaxation time of MALA scales *linearly* in both $\kappa$ and $d$, while giving an explicit hard distribution and $\exp(-d)$-sized bad set. Moreover, both [LST20a, CLA$^+$20] gave strictly spectral lower bounds, which are complemented by our Theorem 3, a mixing time lower bound.

We briefly mention several additional lower bound results in the sampling and sampling-adjacent literature, which are related to this work. Recently, [CLW20] exhibited an information-theoretic lower bound on unadjusted discretizations simulating the *underdamped Langevin dynamics*, whose dimension dependence matches the upper bound of [SL19] (while leaving the precise dependence on $\kappa$ open). Finally, [GLL20] and [CBL20] give information-theoretic lower bounds for estimating normalizing constants of well-conditioned distributions and the number of stochastic gradient queries required by first-order sampling methods under noisy gradient access respectively.

## 2 Preliminaries

In Section 2.1, we give an overview of notation and technical definitions used throughout the paper. We state standard helper concentration bounds we frequently use in Section 2.2. We then recall the definitions of the sampling methods which we study in this paper in Sections 2.3 and 2.4.

### 2.1 Notation

**General notation.** For $d \in \mathbb{N}$ we let $[d] := \{i \in \mathbb{N} \mid 1 \leq i \leq d\}$. We let $\|\cdot\|_2$ denote the Euclidean norm on $\mathbb{R}^d$ for any $d$; for any positive semidefinite matrix $\mathbf{A}$, we let $\|\cdot\|_{\mathbf{A}}$ be its induced seminorm $\|x\|_{\mathbf{A}} = \sqrt{x^\top \mathbf{A} x}$. We use $\|\cdot\|_p$ to denote the $\ell_p$ norm for $p \geq 1$, and $\|\cdot\|_\infty$ is the maximum absolute value of entries. We let $\mathcal{N}(\mu, \mathbf{\Sigma})$ denote the multivariate Gaussian with mean $\mu \in \mathbb{R}^d$ and covariance $\mathbf{\Sigma} \in \mathbb{R}^{d \times d}$. We let $\mathbf{I} \in \mathbb{R}^{d \times d}$ denote the identity matrix when dimensions are clear from context, and $\preceq$ is the Loewner order on the positive semidefinite cone. We let $\{W_t\}_{t \geq 0} \subset \mathbb{R}^d$ denote the standard Brownian motion when dimensions are clear from context.

**Functions.** We say twice-differentiable $f : \mathbb{R}^d \to \mathbb{R}$ is $L$-smooth and $\mu$-strongly convex for $0 \leq \mu \leq L$ if $\mu \mathbf{I} \preceq \nabla^2 f(x) \preceq L \mathbf{I}$ for all $x \in \mathbb{R}^d$. It is well-known that for any $x, y \in \mathbb{R}^d$, this implies $f$ has a Lipschitz gradient (i.e. $\|\nabla f(x) - \nabla f(y)\|_2 \leq L \|x - y\|_2$), and satisfies the quadratic bounds

$$f(x) + \langle \nabla f(x), y - x \rangle + \frac{\mu}{2} \|y - x\|_2^2 \leq f(y) \leq f(x) + \langle \nabla f(x), y - x \rangle + \frac{L}{2} \|y - x\|_2^2.$$

We define the condition number of such a function $f$ by $\kappa := \frac{L}{\mu}$. We will assume that $\kappa$ is at least a constant for convenience of stating bounds; a lower bound of 10 suffices for all our results.

**Distributions.** For distribution $\pi$ on $\mathbb{R}^d$, we say $\pi$ is logconcave if $\frac{d\pi}{dx}(x) = \exp(-f(x))$ for convex $f$; we say $\pi$ is $\mu$-strongly logconcave if $f$ is $\mu$-strongly convex. For $A \subseteq \mathbb{R}^d$ we let $A^c$ denote its complement and $\pi(A) := \int_{x \in A} d\pi(x)$ denote its measure under $\pi$. We say distribution $\rho$ is $\beta$-warm with respect to $\pi$ if $\frac{d\rho}{d\pi}(x) \leq \beta$ everywhere; we define their total variation $\|\pi - \rho\|_{\mathrm{TV}} := \sup_{A \subseteq \mathbb{R}^d} \pi(A) - \rho(A)$. Finally, we denote the expectation and variance of $g : \mathbb{R}^d \to \mathbb{R}$ under $\pi$ by

$$\mathbb{E}_\pi [g] = \int g(x) d\pi(x), \ \mathrm{Var}_\pi [g] = \mathbb{E}_\pi [g^2] - (\mathbb{E}_\pi [g])^2.$$

**Sampling.** Consider a Markov chain defined on $\mathbb{R}^d$ with transition kernel $\{\mathcal{T}_x\}_{x\in\mathbb{R}^d}$, so that $\int \mathcal{T}_x(y)dy = 1$ for all $x$. Further, denote the stationary distribution of the Markov chain by $\pi^*$. Define the Dirichlet form of functions $g, h : \mathbb{R}^d \to \mathbb{R}$ with respect to the Markov chain by

$$\mathcal{E}(g,h) := \int g(x)h(x)d\pi^*(x) - \iint g(y)h(x)\mathcal{T}_x(y)d\pi^*(x)dy.$$

A standard calculation demonstrates that

$$\mathcal{E}(g,g) = \frac{1}{2}\iint (g(x) - g(y))^2 \mathcal{T}_x(y)d\pi^*(x)dy.$$

The mixing of the chain is governed by its spectral gap, a classical quantity we now define:

$$\lambda\left(\{\mathcal{T}_x\}_{x\in\mathbb{R}^d}\right) := \inf_g \left\{ \frac{\mathcal{E}(g,g)}{\text{Var}_{\pi^*}[g]} \right\}. \tag{2}$$

The relaxation time is the inverse spectral gap. We also recall a result of Cheeger [Che69], showing the spectral gap is $O(\Phi)$, where $\Phi$ is the conductance of the chain:

$$\Phi\left(\{\mathcal{T}_x\}_{x\in\mathbb{R}^d}\right) := \inf_{A\subset\mathbb{R}^d|\pi^*(A)\leq\frac{1}{2}} \frac{\int_{x\in A} \mathcal{T}_x(A^c)d\pi^*(x)}{\pi^*(A)} \tag{3}$$

Finally, we recall the definition of a Metropolis filter. A Markov chain with transitions $\{\mathcal{T}_x\}_{x\in\mathbb{R}^d}$ and stationary distribution $\pi^*$ is said to be reversible if for all $x, y \in \mathbb{R}^d$,

$$d\pi^*(x)\mathcal{T}_x(y) = d\pi^*(y)\mathcal{T}_y(x).$$

The Metropolis filter is a way of taking an arbitrary set of proposal distributions $\{\mathcal{P}_x\}_{x\in\mathbb{R}^d}$ and defining a reversible Markov chain with stationary distribution $\pi^*$. In particular, the Markov chain induced by the Metropolis filter has transition distributions $\{\mathcal{T}_x\}_{x\in\mathbb{R}^d}$ defined by

$$\mathcal{T}_x(y) := \mathcal{P}_x(y)\min\left(1, \frac{d\pi^*(y)\mathcal{P}_y(x)}{d\pi^*(x)\mathcal{P}_x(y)}\right) \text{ for all } y \neq x. \tag{4}$$

Whenever the proposal is rejected by the modified distributions above, the chain does not move.

### 2.2 Concentration

Here we state several frequently used (standard) concentration facts.

**Fact 1** (Mill's inequality). *For one-dimensional Gaussian random variable $Z \sim \mathcal{N}(0, \sigma^2)$,*

$$\Pr[Z > t] \leq \sqrt{\frac{2}{\pi}}\frac{\sigma}{t}\exp\left(-\frac{t^2}{2\sigma^2}\right).$$

**Fact 2** ($\chi^2$ tail bounds, Lemma 1 [LM00]). *Let* $\{Z_i\}_{i \in [n]} \sim_{\text{i.i.d.}} \mathcal{N}(0, 1)$ *and* $a \in \mathbb{R}^n_{\geq 0}$. *Then*

$$\Pr\left[\sum_{i \in [n]} a_i Z_i^2 - \|a\|_2^2 \geq 2 \|a\|_2 \sqrt{t} + 2 \|a\|_\infty t\right] \leq \exp(-t),$$

$$\Pr\left[\sum_{i \in [n]} a_i Z_i^2 - \|a\|_2^2 \leq -2 \|a\|_2 \sqrt{t}\right] \leq \exp(-t).$$

**Fact 3** (Bernstein's inequality). *Let* $\{Z_i\}_{i \in [n]}$ *be independent mean-zero random variables with sub-exponential parameter* $\lambda$. *Then*

$$\Pr\left[\left|\sum_{i \in [n]} Z_i\right| > t\right] \leq \exp\left(-\frac{1}{2} \min\left(\frac{t^2}{n\lambda^2}, \frac{t}{\lambda}\right)\right).$$

## 2.3 Metropolis-adjusted Langevin algorithm

In this section, we formally define the Metropolis-adjusted Langevin algorithm (MALA) which we study in Sections 3 and 4. Throughout this discussion, fix a distribution $\pi$ on $\mathbb{R}^d$, with density $\frac{d\pi}{dx}(x) = \exp(-f(x))$, and suppose that $f$ is twice-differentiable for simplicity.

The MALA Markov chain is given by a discretization of the (continuous-time) Langevin dynamics

$$dx_t = -\nabla f(x_t)dt + \sqrt{2}dW_t,$$

which is well-known to have stationary density $\exp(-f(x))$. MALA is defined by performing a simple Euler discretization of the Langevin dynamics up to time $h > 0$, and then applying a Metropolis filter. In particular, define the proposal distribution at a point $x$ by

$$\mathcal{P}_x := \mathcal{N}\left(x - h\nabla f(x), 2h\mathbf{I}\right).$$

We obtain the MALA transition distribution by applying the definition (4), which yields

$$\mathcal{T}_x(y) \propto \exp\left(-\frac{\|y - (x - h\nabla f(x))\|_2^2}{4h}\right) \min\left(1, \frac{\exp\left(-f(y) - \frac{\|x-(y-h\nabla f(y))\|_2^2}{4h}\right)}{\exp\left(-f(x) - \frac{\|y-(x-h\nabla f(x))\|_2^2}{4h}\right)}\right). \tag{5}$$

The normalization constant above is that of the multivariate Gaussian with covariance $2h\mathbf{I}$.

## 2.4 Hamiltonian Monte Carlo

In this section, we formally define the (Metropolized) Hamiltonian Monte Carlo (HMC) method which we study in Section 6. We assume the same setting as Section 2.3.

The Metropolized HMC algorithm is governed by two parameters, a step size $\eta > 0$ and a step count $K \in \mathbb{N}$, and can be viewed as a multi-step generalization of MALA. In particular, when $K = 1$ it is straightforward to show that HMC is a reparameterization of MALA, see e.g. Appendix A of [LST20a]. More generally, from an iterate $x$, HMC performs the following updates.

1. $x_0 \leftarrow x$, $v_0 \sim \mathcal{N}(0, \mathbf{I})$
2. For $0 \leq k < K$:

(a) $v_{k+\frac{1}{2}} \leftarrow v_k - \frac{\eta}{2}\nabla f(x_k)$

(b) $x_{k+1} \leftarrow x_k + \eta v_{k+\frac{1}{2}}$

(c) $v_{k+1} \leftarrow v_k - \frac{\eta}{2}\nabla f(x_{k+1})$

3. Return $x_K$

Each loop of step 2 is known in the literature as a "leapfrog" step, and is a discretization of Hamilton's equations for the Hamiltonian function $\mathcal{H}(x,v) := f(x) + \frac{1}{2}\|v\|_2^2$; for additional background, we refer the reader to [CDWY19]. This discretization is well-known to have reversible transition probabilities (i.e. the transition density is the same if the endpoints are swapped) because it satisfies a property known as *symplecticity*. Moreover, the Markov chain has stationary density on the expanded space $(x,v) \in \mathbb{R}^d \times \mathbb{R}^d$ proportional to $\exp(-\mathcal{H}(x,v))$. Correspondingly, the Metropolized HMC Markov chain performs the above algorithm from a point $x$, and accepts with probability

$$\min\left\{1, \frac{\exp\left(-\mathcal{H}(x_K, v_K)\right)}{\exp\left(-\mathcal{H}(x_0, v_0)\right)}\right\}. \tag{6}$$

## 3 Lower bound for MALA on Gaussians

In this section, we derive a upper bound on the spectral gap of MALA when the target distribution is restricted to being a multivariate Gaussian (i.e. its negative log-density is a quadratic in some well-conditioned matrix $\mathbf{A}$). Throughout this section we will let $f(x) = \frac{1}{2}x^\top \mathbf{A}x$ for some $\mathbf{I} \preceq \mathbf{A} \preceq \kappa\mathbf{I}$. We remark here that without loss of generality, we have assumed that the minimizer of $f$ is the all-zeros vector and the strong convexity parameter is $\mu = 1$. These follow from invariance of condition number under linear translations and scalings of the variable.

Next, we define a specific hard quadratic function we will consider in this section, $f_{\mathrm{hq}} : \mathbb{R}^d \to \mathbb{R}$. Specifically, $f_{\mathrm{hq}}$ will be a quadratic in a diagonal matrix $\mathbf{A}$ which has $\mathbf{A}_{11} = 1$ and $\mathbf{A}_{ii} = \kappa$ for $2 \le i \le d$. We can rewrite this as

$$f_{\mathrm{hq}}(x) := \sum_{i \in [d]} f_i(x_i), \text{ where } f_i(c) = \begin{cases} \frac{1}{2}c^2 & i = 1 \\ \frac{\kappa}{2}c^2 & 2 \le i \le d \end{cases}. \tag{7}$$

Notice that $f_{\mathrm{hq}}$ is coordinate-wise separable, and behaves identically on coordinates $2 \le i \le d$ (and differently on coordinate 1). To this end for a vector $v \in \mathbb{R}^d$, we will denote its first coordinate by $v_1 \in \mathbb{R}$, and its remaining coordinates by $v_{-1} \in \mathbb{R}^{d-1}$. This will help us analyze the behavior of these components separately, and simplify notation.

We next show that for coordinate-separable functions with well-behaved first coordinate, such as our $f_{\mathrm{hq}}$, the spectral gap (defined in (2)) of the MALA Markov chain is governed by the step size $h$. The following is an extension of an analogous proof in [CLA$^+$20].

**Lemma 1.** *Consider the MALA Markov chain* (5), *with stationary distribution* $\pi^*$ *with negative log-density* $f$. *Suppose* $f$ *is coordinate-wise separable (i.e.* $f(x) = \sum_{i \in [d]} f_i(x_i)$). *If* $f(x) = f(-x)$ *for all* $x \in \mathbb{R}^d$, $f_1$ *is* $O(1)$-*smooth, and* $\mathbb{E}_{x_1 \sim \exp(-f_1)}[x_1^2] = \Theta(1)$, *the spectral gap* (2) *is* $O(h + h^2)$.

*Proof.* Recalling the definition (2), we choose $g(x) = x_1$; note that by symmetry of $f$ around the origin, we have $\mathbb{E}_{\pi^*}[g] = 0$, and thus by our assumption,

$$\mathrm{Var}_{\pi^*}[g] = \mathbb{E}_{x \sim \pi^*}[x_1^2] = \Theta(1).$$

Here we used that $\pi^*$ is a product distribution. Thus it suffices to upper bound $\mathcal{E}(g,g)$:

$$\begin{aligned}
\mathcal{E}(g,g) &= \frac{1}{2} \iint (x_1 - y_1)^2 \mathcal{T}_x(y) d\pi^*(x) dy \\
&\leq \frac{1}{2} \iint (x_1 - y_1)^2 \mathcal{P}_x(y) d\pi^*(x) dy \\
&= \frac{1}{2} \mathbb{E}_{x \sim \pi^*, \xi \sim \mathcal{N}(0,1)} \left[ \left( h f_1'(x_1) - \sqrt{2h} \xi \right)^2 \right] \\
&\leq \mathbb{E}_{x \sim \pi^*} \left[ h^2 \left( f_1'(x_1) \right)^2 \right] + 2 \mathbb{E}_{\xi \sim \mathcal{N}(0,1)} \left[ h \xi^2 \right] \\
&\leq O(h^2) \mathbb{E}_{x \sim \pi^*} \left[ x_1^2 \right] + 2h = O\left( h + h^2 \right).
\end{aligned}$$

In the second line, we used that whenever the Markov chain rejects the distribution both terms are zero; in the third, we used the definition of the MALA proposals; in the fourth, we used $(a+b)^2 \leq 2a^2 + 2b^2$ for $a, b \in \mathbb{R}$. Finally, the last line used that symmetry implies that the minimizer of $f$ is the origin, so applying Lipschitzness and $f_1'(0) = 0$ yields the desired bound. $\qquad\square$

This immediately implies a spectral gap bound on our hard function $f_{\mathrm{hq}}$.

**Corollary 1.** *The spectral gap of the MALA Markov chain for sampling from the density proportional to* $\exp(-f_{\mathrm{hq}})$*, where $f_{\mathrm{hq}}$ is defined in* (7)*, is $O(h + h^2)$.*

It remains to give a lower bound on the step size $h$, which we accomplish by upper bounding the acceptance probability of MALA. We will give a step size analysis for a fairly general characterization of Markov chains, where the proposal distribution from a point $x$ is

$$y = \begin{pmatrix} y_1 \\ y_{-1} \end{pmatrix}, \text{ where } y_1 = (1 - \alpha_1) x_1 + \beta_1 g_1 \tag{8}$$
$$\text{and } y_{-1} = (1 - \alpha_{-1}) x_{-1} + \beta_{-1} g_{-1}, \text{ for } g \sim \mathcal{N}(0, \mathbf{I}).$$

To be concrete, recall that the proposal distribution for MALA (5) is given by $y = x - h\mathbf{A}x + \sqrt{2h}g$. For the $\mathbf{A}$ used in defining $f_{\mathrm{hq}}$, this is of the form (8) with the specific parameters

$$\alpha_1 = h, \ \alpha_{-1} = h\kappa, \ \beta_1 = \beta_{-1} = \sqrt{2h}.$$

However, this more general characterization will save significant amounts of recalculation when analyzing updates of the HMC Markov chain in Section 6. Recalling the formula (5), we first give a closed form for the acceptance probability.

**Lemma 2.** *For $f(x) = \frac{1}{2} x^\top \mathbf{A} x$, we have*

$$f(x) - f(y) + \frac{1}{4h} \left( \|y - (x - h\nabla f(x))\|_2^2 - \|x - (y - h\nabla f(y))\|_2^2 \right) = \frac{h}{4} \|x\|_{\mathbf{A}^2}^2 - \frac{h}{4} \|y\|_{\mathbf{A}^2}^2.$$

*Supposing $y$ is of the form in* (8) *and $\mathbf{A}$ is as in* (7)*, we have*

$$\begin{aligned}
\frac{h}{4} \|x\|_{\mathbf{A}^2}^2 - \frac{h}{4} \|y\|_{\mathbf{A}^2}^2 &= \frac{h}{4} \left( \left( 2\alpha_1 - \alpha_1^2 \right) x_1^2 - \beta_1^2 g_1^2 - 2(1 - \alpha_1) \beta_1 x_1 g_1 \right) \\
&\quad + \frac{h\kappa^2}{4} \left( \left( 2\alpha_{-1} - \alpha_{-1}^2 \right) \|x_{-1}\|_2^2 - \beta_{-1}^2 \|g_{-1}\|_2^2 - 2(1 - \alpha_{-1}) \beta_{-1} \langle x_{-1}, g_{-1} \rangle \right).
\end{aligned}$$

*Proof.* This is a direct computation which we perform here for completeness: the given quantity is

$$\frac{1}{2}\|x\|_{\mathbf{A}}^2 - \frac{1}{2}\|y\|_{\mathbf{A}}^2 + \frac{1}{4h}\left(\|y - x + h\mathbf{A}x\|_2^2 - \|x - y + h\mathbf{A}y\|_2^2\right)$$

$$= \frac{1}{2}\|x\|_{\mathbf{A}}^2 - \frac{1}{2}\|y\|_{\mathbf{A}}^2 + \frac{1}{2}\langle y - x, \mathbf{A}x\rangle + \frac{h}{4}\|x\|_{\mathbf{A}^2}^2 - \frac{1}{2}\langle x - y, \mathbf{A}y\rangle - \frac{h}{4}\|y\|_{\mathbf{A}^2}^2 = \frac{h}{4}\|x\|_{\mathbf{A}^2}^2 - \frac{h}{4}\|y\|_{\mathbf{A}^2}^2.$$

The second equality follows from expanding the definition of $y$:

$$\|x\|_{\mathbf{A}^2}^2 - \|y\|_{\mathbf{A}^2}^2 = x_1^2 - \left((1 - \alpha_1)x_1 + \beta_1 g_1\right)^2 + \kappa^2\left(\|x_{-1}\|_2^2 - \|(1 - \alpha_{-1})x_{-1} + \beta_{-1}g_{-1}\|_2^2\right)$$

$$= \left(2\alpha_1 - \alpha_1^2\right)x_1^2 - \beta_1^2 g_1^2 - 2(1 - \alpha_1)\beta_1 x_1 g_1$$

$$+ \kappa^2\left(\left(2\alpha_{-1} - \alpha_{-1}^2\right)\|x_{-1}\|_2^2 - \beta_{-1}^2\|g_{-1}\|_2^2 - 2(1 - \alpha_{-1})\beta_{-1}\langle x_{-1}, g_{-1}\rangle\right).$$

$\square$

**Corollary 2.** *For any fixed $x \in \mathbb{R}^d$, and supposing $y$ is of the form in* (8) *and $\mathbf{A}$ is as in* (7),

$$\mathbb{E}_{g\sim\mathcal{N}(0,\mathbf{I})}\left[f(x) - f(y) + \frac{1}{4h}\left(\|y - (x - h\nabla f(x))\|_2^2 - \|x - (y - h\nabla f(y))\|_2^2\right)\right]$$

$$= \frac{h}{4}\left(\left(2\alpha_1 - \alpha_1^2\right)x_1^2 - \beta_1^2\right) + \frac{h\kappa^2}{4}\left(\left(2\alpha_{-1} - \alpha_{-1}^2\right)\|x_{-1}\|_2^2 - \beta_{-1}^2(d - 1)\right).$$

*Proof.* This follows from Lemma 2, independence of $g$ and $x$, and linearity of trace and expectation applied on squared coordinates of $g$, where we recognize $\mathbb{E}_{g\sim\mathcal{N}(0,\mathbf{I})}[gg^\top] = \mathbf{I}$. $\square$

Next, for a fixed $x$, consider the random variables $R_i^x$:

$$R_i^x = \begin{cases} \frac{h}{4}\left(\left(2\alpha_1 - \alpha_1^2\right)x_1^2 - \beta_1^2 g_1^2 - 2(1 - \alpha_1)\beta_1 x_1 g_1\right) & i = 1 \\ \frac{h\kappa^2}{4}\left(\left(2\alpha_{-1} - \alpha_{-1}^2\right)x_i^2 - \beta_{-1}^2 g_i^2 - 2(1 - \alpha_{-1})\beta_{-1}x_i g_i\right) & 2 \leq i \leq d \end{cases}$$

where $g \sim \mathcal{N}(0, \mathbf{I})$ is a standard Gaussian random vector. Notice that for a given realization of $g$, we have by Lemma 2 that

$$\sum_{i\in[d]} R_i^x = \frac{h}{4}\|x\|_{\mathbf{A}^2} - \frac{h}{4}\|y\|_{\mathbf{A}^2}. \tag{9}$$

We computed the expectation of $\sum_{i\in[d]} R_i^x$ in Corollary 2. We next give a high-probability bound on the deviation of $\sum_{i\in[d]} R_i^x$ from its expectation.

**Lemma 3.** *With probability at least $1 - \delta$ over the randomness of $g \sim \mathcal{N}(0, \mathbf{I})$,*

$$\sum_{i\in[d]} R_i^x - \mathbb{E}_{g\sim\mathcal{N}(0,\mathbf{I})}\left[\sum_{i\in[d]} R_i^x\right] \leq 2h|\alpha_1 - 1|\beta_1|x_1|\sqrt{\log\left(\frac{4}{\delta}\right)} + h\beta_1^2\sqrt{\log\left(\frac{4}{\delta}\right)}$$

$$+ 2h\kappa^2|\alpha_{-1} - 1|\beta_{-1}\|x_{-1}\|_2\sqrt{\log\left(\frac{4}{\delta}\right)} + h\kappa^2\beta_{-1}^2\sqrt{d\log\left(\frac{4}{\delta}\right)}.$$

*Proof.* In defining $\{R_i^x\}_{i\in[d]}$, the terms involving $\{x_i^2\}_{i\in[d]}$ are deterministic. Thus, we need to upper

bound the deviations of the remaining terms, namely

$$S_1^{(1)} := \frac{h}{2}(\alpha_1 - 1)\beta_1 x_1 g_1, \ S_1^{(2)} := \frac{h\beta_1^2}{4}\left(1 - g_1^2\right),$$

$$S_{-1}^{(1)} := \frac{h\kappa^2}{2}(\alpha_{-1} - 1)\beta_{-1}\sum_{2 \le i \le d} x_i g_i, \ S_{-1}^{(2)} := \frac{h\kappa^2\beta_{-1}^2}{4}\sum_{2 \le i \le d}\left(1 - g_i^2\right).$$

To motivate these definitions, $S_1^{(1)} + S_1^{(2)} + S_{-1}^{(1)} + S_{-1}^{(2)}$ is the left hand side of the display in the lemma statement. We begin with $S_{-1}^{(1)}$. Notice that this is a one-dimensional Gaussian random variable distributed as

$$\mathcal{N}\left(0, \sigma_1^2\right) \text{ where } \sigma_1 := \frac{h\kappa^2}{2}|\alpha_{-1} - 1|\beta_{-1}\left\|x_{-1}\right\|_2.$$

Thus, applying Mill's inequality yields

$$\Pr\left[S_{-1}^{(1)} > t\right] \le \sqrt{\frac{2}{\pi}}\frac{\sigma_1}{t}\exp\left(-\frac{t^2}{2\sigma_1^2}\right) \le \frac{\delta}{4}, \text{ for } t = 4\sigma_1\sqrt{\log\left(\frac{4}{\delta}\right)}.$$

Next, to bound the term $S_{-1}^{(2)}$, define

$$\sigma_2 := \frac{h\kappa^2\beta_{-1}^2}{4}\sqrt{d-1}.$$

Standard $\chi^2$ concentration results (Fact 2) then yield

$$\Pr\left[S_{-1}^{(2)} > t\right] \le \exp\left(-\frac{t^2}{4\sigma_2^2}\right) \le \frac{\delta}{4}, \text{ for } t = 2\sigma_2\sqrt{\log\left(\frac{4}{\delta}\right)}.$$

Similar bounds follow for $S_1^{(1)}$ and $S_1^{(2)}$, whose computations we omit for brevity. Taking a union bound over these four terms yields the desired claim. $\qquad\square$

Finally, we have a complete characterization of a bad set $\Omega \subset \mathbb{R}^d$ where, with high probability over the proposal distribution, the acceptance probability is extremely small.

**Proposition 1.** *Let $x \in \mathbb{R}^d$ satisfy $\left\|x_{-1}\right\|_2 \le \sqrt{\frac{2d}{3\kappa}}$ and $|x_1| \le 5\sqrt{\log d}$, and suppose $y$ is of the form in (8) and $\mathbf{A}$ is as in (7). Also suppose that*

$$|\alpha_{-1}| \le \frac{3}{5}\beta_{-1}^2\kappa, \ \beta_{-1} = \omega\left(\sqrt{\frac{\log d}{\kappa d}}\right), \ |\alpha_1| = O(|\alpha_{-1}|), \ \beta_1 = O(\beta_{-1}).$$

*Then with probability at least $1 - d^{-5}$ over the randomness of $g \sim \mathcal{N}(0, \mathbf{I})$, we have*

$$\frac{h}{4}\left\|x\right\|_{\mathbf{A}^2} - \frac{h}{4}\left\|y\right\|_{\mathbf{A}^2} = -\Omega\left(h\kappa^2\beta_{-1}^2 d\right).$$

*Proof.* We first handle terms involving $x_{-1}$ and $g_{-1}$. Combining (9), Corollary 2, and Lemma 3, we have with probability at least $1 - \frac{1}{2}d^{-5}$ over the randomness of $g \sim \mathcal{N}(0, \mathbf{I})$ that $\left\|x_{-1}\right\|_{\mathbf{A}_{-1}^2}^2 - \left\|y_{-1}\right\|_{\mathbf{A}_{-1}^2}^2$

(where $\mathbf{A}_{-1}$ is the Hessian of $f_{\mathrm{hq}}$ on the last $d-1$ coordinates) is upper bounded by

$$\frac{h\kappa^2}{4}\left(\left(2\alpha_{-1}-\alpha_{-1}^2\right)\|x_{-1}\|_2^2 - \beta_{-1}^2(d-1)\right) + 5h\kappa^2|\alpha_{-1}-1|\beta_{-1}\|x_{-1}\|_2\sqrt{\log d} + 3h\kappa^2\beta_{-1}^2\sqrt{d\log d}$$

$$\leq -\frac{h\kappa^2}{4.5}\beta_{-1}^2 d + \frac{h\kappa^2}{4}(2\alpha_{-1}-\alpha_{-1}^2)\|x_{-1}\|_2^2 + 5h\kappa^2|\alpha_{-1}-1|\beta_{-1}\|x_{-1}\|_2\sqrt{\log d}. \tag{10}$$

Here we dropped the last term in the first line by adjusting a constant since it is dominated for sufficiently large $d$. It remains to show that all the terms in the second line other than $-\frac{h\kappa^2}{4.5}\beta_{-1}^2 d$ are bounded by $O(h\kappa^2\beta_{-1}^2 d)$. We will perform casework on the size of $\alpha_{-1}$.

*Case 1: $|\alpha_{-1}| > 3$.* In this case, we have for sufficiently large $d$, by Young's inequality

$$5h\kappa^2|\alpha_{-1}-1|\beta_{-1}\|x_{-1}\|_2\sqrt{\log d} \leq \frac{1}{40}h\kappa^2|\alpha_{-1}|\beta_{-1}\|x_{-1}\|_2\sqrt{d}$$

$$\leq \frac{1}{80}h\kappa^2\beta_{-1}^2 d + \frac{1}{80}h\kappa^2\alpha_{-1}^2\|x_{-1}\|_2^2.$$

Plugging this bound into (10), we have the desired

$$\|x_{-1}\|_{\mathbf{A}_{-1}^2}^2 - \|y_{-1}\|_{\mathbf{A}_{-1}^2}^2 \leq -\frac{h\kappa^2}{5}\beta_{-1}^2 d + \frac{h\kappa^2}{4}\left(2\alpha_{-1}-0.9\alpha_{-1}^2\right)\|x_{-1}\|_2^2 \leq -\frac{h\kappa^2}{5}\beta_{-1}^2 d.$$

In the last inequality we used $2\alpha_{-1}-0.9\alpha_{-1}^2 \leq 0$ for $|\alpha_{-1}| > 3$.

*Case 2: $|\alpha_{-1}| \leq 3$.* In this case, we first observe by our assumed bounds on $\|x_{-1}\|_2$ and $\beta_{-1}$,

$$5h\kappa^2|\alpha_{-1}-1|\beta_{-1}\|x_{-1}\|_2\sqrt{\log d} \leq 20h\kappa^{1.5}\beta_{-1}\sqrt{d\log d} = o\left(h\kappa^2\beta_{-1}^2 d\right).$$

Thus, substituting into (10) and dropping the (nonpositive) term corresponding to $\alpha_{-1}^2$,

$$\|x_{-1}\|_{\mathbf{A}_{-1}^2}^2 - \|y_{-1}\|_{\mathbf{A}_{-1}^2}^2 \leq -\frac{h\kappa^2}{4.8}\beta_{-1}^2 d + \frac{h\kappa^2}{2}\alpha_{-1}\|x_{-1}\|_2^2$$

$$\leq -\frac{h\kappa^2}{4.8}\beta_{-1}^2 d + \frac{h\kappa\alpha_{-1}d}{3} = -\Omega\left(h\kappa^2\beta_{-1}^2 d\right).$$

In the second inequality, we used the assumed bound on $\|x_{-1}\|_2^2$, and in the last we used the bound $|\alpha_{-1}| \leq \frac{3}{5}\beta_{-1}^2\kappa$ to reach the conclusion.

To complete the proof we need to show terms involving $x_1$ and $g_1$ are small. In particular, combining (9), Corollary 2, and Lemma 3 and dropping nonnegative terms, it suffices to argue

$$\frac{h}{2}\alpha_1 x_1^2 + 5h|\alpha_1-1|\beta_1|x_1|\sqrt{\log d} + 3h\beta_1^2\sqrt{\log d} = o\left(h\kappa^2\beta_{-1}^2 d\right).$$

This bound clearly holds for the last term $h\beta_1^2\sqrt{\log d}$ using $\beta_1 = O(\beta_{-1})$. For the first term, it suffices to use our assumed bounds on $|\alpha_1|$ and $x_1$. Finally, the middle term $5h|\alpha_1-1|\beta_1|x_1|\sqrt{\log d}$ is low-order compared to the term $5h\kappa^2|\alpha_{-1}-1|\beta_{-1}\|x_{-1}\|_2\sqrt{\log d}$ which we argued about earlier, and hence does not affect any of our earlier bounds by more than a constant. The left-hand side of the above display is an upper bound of the first coordinate's contribution with probability at least $1 - \frac{1}{2}d^{-5}$, so a union bound shows the proof succeeds with probability $\geq 1 - d^{-5}$. $\qquad\square$

Finally, we are ready to give the main lower bound of this section.

**Theorem 1.** *For every step size, there is a target Gaussian on $\mathbb{R}^d$ whose negative log-density always has Hessian eigenvalues in $[1, \kappa]$, such that the relaxation time of MALA is $\Omega(\frac{\kappa\sqrt{d}}{\sqrt{\log d}})$.*

*Proof.* Let $\pi^*$ be the Gaussian with log-density $-f_{\mathrm{hq}}$ (7) throughout this proof. If $h = O\left(\frac{\sqrt{\log d}}{\kappa\sqrt{d}}\right)$, then Corollary 1 immediately implies the result, so for the remainder of the proof suppose

$$h = \omega\left(\frac{\sqrt{\log d}}{\kappa\sqrt{d}}\right). \tag{11}$$

We first recall that MALA Markov chains are an instance of (8) with

$$\alpha_1 = h, \ \alpha_{-1} = h\kappa, \ \beta_1 = \beta_{-1} = \sqrt{2h}.$$

It is easy to see that these parameters satisfy the assumptions in Proposition 1, for the given range of $h$. We define a "bad starting set" as follows:

$$\Omega := \left\{ x \ \middle| \ \|x_{-1}\|_2^2 \leq \frac{2d}{3\kappa}, \ x_1^2 \leq 25\log d \right\}. \tag{12}$$

For any $x \in \Omega$, and $h$ satisfying (11), Proposition 1 is applicable, and by our definition of $\Omega$, any $x \in \Omega$ has proposals which will be accepted with probability

$$\exp\left(-\Omega\left(h\kappa^2\beta_{-1}^2 d\right)\right) = \exp\left(-\Omega(h^2\kappa^2 d)\right) \leq \frac{1}{d^{10}}.$$

The conductance of the Markov chain (3) is then at most $\frac{2}{d^5}$ by the witness set $\Omega$ and the failure probability of Proposition 1, which concludes the proof by Cheeger's inequality [Che69], where we use the assumption that $\kappa = O(d^4)$. $\square$

Finally, as it clarifies the required warmness to escape the bad set in the proof of Theorem 1 (and is used in our mixing time bounds in Section 5), we lower bound the measure of $\Omega$ according to $\pi^*$. Applying Lemma 4 shows with probability at least $\exp(-\frac{1}{12}d)$, $\|x_{-1}\|_2^2 \leq \frac{d}{2\kappa}$, and Fact 1 shows that $x_1^2 \leq 25\log d$ with probability at least $\frac{1}{2}$; combining shows that the measure is at least $\exp(-d)$. We required one helper technical fact, a small-ball probability bound for Gaussians.

**Lemma 4.** *Let $v \sim \mathcal{N}(0, \mathbf{I})$ be a random Gaussian vector in $n$ dimensions. For large enough $n$,*

$$\Pr\left[\|v\|_2^2 \leq \frac{n}{2}\right] \geq \exp\left(-\frac{n}{12}\right).$$

*Proof.* Observe that $\|v\|_2^2$ follows a $\chi^2$ distribution with $n$ degrees of freedom. Thus this probability is governed by the $\chi^2$ cumulative density function, and is

$$\frac{1}{\Gamma(k)}\gamma(k, ck)$$

where we define $k := \frac{n}{2}$ and $c := \frac{1}{2}$; here $\Gamma$ is the standard gamma function, and $\gamma$ is the lower incomplete gamma function. Next, we have the bound from [ODL$^+$20]

$$\frac{1}{\Gamma(k)}\gamma(k, ck) \geq (1 - \exp(-\ell ck))^k, \ \ell := (\Gamma(k+1))^{-\frac{1}{k-1}}.$$

A direct calculation yields $\ell \geq \frac{2.5}{k} \implies 1 - \exp(-\ell ck) \geq \exp(-\frac{1}{6})$ for large enough $k$. Recalling we defined $k = \frac{n}{2}$ yields the conclusion. $\square$

## 4  Lower bound for MALA on well-conditioned distributions

In this section, we derive a lower bound on the relaxation time of MALA for sampling from a distribution with density proportional to $\exp(-f(x))$, where $f : \mathbb{R}^d \to \mathbb{R}$ is a (non-quadratic) target function with condition number $\kappa$. In particular, by exploiting the structure of non-cancellations which do not occur for quadratics, we will attain a stronger lower bound.

Our first step is to derive an upper bound on the acceptance probability for a general target function $f$ according to the MALA updates (5), analogously to Lemma 2 in the Gaussian case.

**Lemma 5.** *For any function $f : \mathbb{R}^d \to \mathbb{R}$, we have*

$$
f(x) - f(y) + \frac{1}{4h} \left( \|y - (x - h\nabla f(x))\|_2^2 - \|x - (y - h\nabla f(y))\|_2^2 \right)
$$
$$
= -f(y) + f(x) - \frac{1}{2} \langle x - y, \nabla f(x) + \nabla f(y) \rangle + \frac{h}{4} \|\nabla f(x)\|_2^2 - \frac{h}{4} \|\nabla f(y)\|_2^2 .
$$

*Proof.* This is a direct computation which we perform here for completeness:

$$
f(x) - f(y) + \frac{1}{4h} \left( \|y - (x - h\nabla f(x))\|_2^2 - \|x - (y - h\nabla f(y))\|_2^2 \right)
$$
$$
= f(x) - f(y) + \frac{1}{2h} \langle y - x, h\nabla f(x) \rangle - \frac{1}{2h} \langle x - y, h\nabla f(y) \rangle + \frac{h}{4} \|\nabla f(x)\|_2^2 - \frac{h}{4} \|\nabla f(y)\|_2^2
$$
$$
= -f(y) + f(x) - \frac{1}{2} \langle x - y, \nabla f(x) + \nabla f(y) \rangle + \frac{h}{4} \|\nabla f(x)\|_2^2 - \frac{h}{4} \|\nabla f(y)\|_2^2 .
$$

$\square$

Next, recall the proposal distribution of the MALA updates (5) sets $y = x - h\nabla f(x) + \sqrt{2h} g$ where $g \sim \mathcal{N}(0, \mathbf{I})$. We further split this update into a random step and a deterministic step, by defining

$$
x_g := x + \sqrt{2h} g, \text{ where } g \sim \mathcal{N}(0, \mathbf{I}) \text{ and } y = x_g - h\nabla f(x). \tag{13}
$$

This will allow us to reason about the effects of the stochastic and drift terms separately. We crucially will use the following decomposition of the equation in Lemma 5:

$$
-f(y) + f(x) - \frac{1}{2} \langle x - y, \nabla f(x) + \nabla f(y) \rangle + \frac{h}{4} \|\nabla f(x)\|_2^2 - \frac{h}{4} \|\nabla f(y)\|_2^2
$$
$$
= -f(x_g) + f(x) - \frac{1}{2} \langle x - x_g, \nabla f(x) + \nabla f(x_g) \rangle
$$
$$
+ f(x_g) - f(y) - \frac{1}{2} \langle x - x_g, \nabla f(y) - \nabla f(x_g) \rangle \tag{14}
$$
$$
- \frac{1}{2} \langle x_g - y, \nabla f(x) + \nabla f(y) \rangle + \frac{h}{4} \|\nabla f(x)\|_2^2 - \frac{h}{4} \|\nabla f(y)\|_2^2 .
$$

We will use the following observation, which gives an alternate characterization of the second line of (14), as well as a bound on the third and fourth lines for smooth functions.

**Lemma 6.** *For twice-differentiable $f : \mathbb{R}^d \to \mathbb{R}$, letting $x_s := x + s(x_g - x)$ for $s \in [0,1]$, we have*

$$-f(x_g) + f(x) - \frac{1}{2} \langle x - x_g, \nabla f(x) + \nabla f(x_g) \rangle = -2h \int_0^1 \left( \frac{1}{2} - s \right) g^\top \nabla^2 f(x_s) g \, ds.$$

*Moreover, assuming $f$ is $\kappa$-smooth,*

$$f(x_g) - f(y) - \frac{1}{2} \langle x - x_g, \nabla f(y) - \nabla f(x_g) \rangle$$

$$-\frac{1}{2} \langle x_g - y, \nabla f(x) + \nabla f(y) \rangle + \frac{h}{4} \|\nabla f(x)\|_2^2 + \frac{h}{4} \|\nabla f(y)\|_2^2$$

$$\leq 2 \left( h^2 \kappa + h^3 \kappa^2 \right) \|\nabla f(x)\|_2^2 + 3 \left( h^{1.5} \kappa + h^{2.5} \kappa^2 \right) \|g\|_2 \|\nabla f(x)\|_2 + h^2 \kappa^2 \|g\|_2^2 .$$

*Proof.* By integrating twice and using the definition $x_g = x + \sqrt{2h} g$,

$$f(x_g) = f(x) + \int_0^1 \langle \nabla f(x_s), x_g - x \rangle \, ds$$

$$= f(x) + \langle \nabla f(x), x_g - x \rangle + \int_0^1 \left\langle \int_0^s \nabla^2 f(x_t) (x_g - x) \, dt, x_g - x \right\rangle ds \qquad (15)$$

$$= f(x) + \langle \nabla f(x), x_g - x \rangle + 2h \int_0^1 (1 - s) g^\top \nabla^2 f(x_s) g \, ds.$$

Similarly,

$$f(x) = f(x_g) + \langle \nabla f(x_g), x - x_g \rangle + 2h \int_0^1 s g^\top \nabla^2 f(x_s) g \, ds. \qquad (16)$$

The first conclusion follows from combining (15) and (16). Next, assuming $f$ is $\kappa$-smooth,

$$f(x_g) - f(y) - \frac{1}{2} \langle x - x_g, \nabla f(y) - \nabla f(x_g) \rangle - \frac{1}{2} \langle x_g - y, \nabla f(x) + \nabla f(y) \rangle$$

$$= f(x_g) - f(y) + \frac{\sqrt{2h}}{2} \langle g, \nabla f(y) - \nabla f(x_g) \rangle - \langle x_g - y, \nabla f(y) \rangle - \frac{h}{2} \langle \nabla f(x), \nabla f(x) - \nabla f(y) \rangle$$

$$\leq f(x_g) - f(y) - \langle x_g - y, \nabla f(y) \rangle + \frac{\sqrt{2h}}{2} \|g\|_2 \|\nabla f(x_g) - \nabla f(y)\|_2 + \frac{h}{2} \|\nabla f(x)\|_2 \|\nabla f(x) - \nabla f(y)\|_2$$

$$\leq \frac{\kappa}{2} \|x_g - y\|_2^2 + \frac{\sqrt{2h}\kappa}{2} \|g\|_2 \|x_g - y\|_2 + \frac{h\kappa}{2} \|\nabla f(x)\|_2 \|x - y\|_2$$

$$\leq \frac{h^2 \kappa}{2} \|\nabla f(x)\|_2^2 + \frac{\sqrt{2}}{2} h^{1.5} \kappa \|g\|_2 \|\nabla f(x)\|_2 + \frac{h\kappa}{2} \|\nabla f(x)\|_2 \left( \sqrt{2h} \|g\|_2 + h \|\nabla f(x)\|_2 \right)$$

$$= h^2 \kappa \|\nabla f(x)\|_2^2 + \sqrt{2} h^{1.5} \kappa \|g\|_2 \|\nabla f(x)\|_2 .$$

$$(17)$$

The second line used the definitions of $x_g$ and $y$ in (13), and the third used Cauchy-Schwarz. The fourth used smoothness (which implies gradient Lipschitzness), and the fifth again used (13) and

the triangle inequality. Next, we bound the remaining terms $\frac{h}{4}\|\nabla f(x)\|_2^2 - \frac{h}{4}\|\nabla f(y)\|_2^2$:

$$
\begin{aligned}
\frac{h}{4}\|\nabla f(x)\|_2^2 - \frac{h}{4}\|\nabla f(y)\|_2^2 &= \frac{h}{4}\langle \nabla f(x) + \nabla f(y), \nabla f(x) - \nabla f(y)\rangle \\
&\leq \frac{h\kappa}{4}\left(2\|\nabla f(x)\|_2 + \kappa\|x - y\|_2\right)\|x - y\|_2 \\
\leq \frac{h\kappa}{4}\left(2\|\nabla f(x)\|_2 + h\kappa\|\nabla f(x)\|_2 + \sqrt{2h}\kappa\|g\|_2\right)&\left(h\|\nabla f(x)\|_2 + \sqrt{2h}\|g\|_2\right) \\
\leq \frac{1}{2}\left(h^2\kappa + h^3\kappa^2\right)\|\nabla f(x)\|_2^2 + \frac{\sqrt{2}}{2}&\left(h^{1.5}\kappa + h^{2.5}\kappa^2\right)\|g\|_2\|\nabla f(x)\|_2 + h^2\kappa^2\|g\|_2^2.
\end{aligned}
\tag{18}
$$

Combining (17) and (18) yields the conclusion. $\qquad\square$

We will ultimately use the second bound in Lemma 6 to argue that the third and fourth lines in (14) are low-order, so it remains to concentrate on the remaining term,

$$
-f(x_g) + f(x) - \frac{1}{2}\langle x - x_g, \nabla f(x) + \nabla f(x_g)\rangle = -2h\int_0^1\left(\frac{1}{2} - s\right)g^\top\nabla^2 f(x_s)g\,ds. \tag{19}
$$

Our goal is to demonstrate this term is $-\Omega(h\kappa d)$ over an inverse-exponentially sized region, for a particular hard distribution. As it is coordinate-wise separable, our proof strategy will be to construct a hard one-dimensional function, and replicate it to obtain a linear dependence on $d$.

We now define the specific hard function $f_{\mathrm{hard}} : \mathbb{R}^d \to \mathbb{R}$ we work with for the remainder of the section; it is straightforward to see $f_{\mathrm{hard}}$ is $\kappa$-smooth and 1-strongly convex.

$$
f_{\mathrm{hard}}(x) := \sum_{i\in[d]} f_i(x_i), \text{ where } f_i(c) = \begin{cases} \frac{1}{2}c^2 & i = 1 \\ \frac{\kappa}{3}c^2 - \frac{\kappa h}{3}\cos\frac{c}{\sqrt{h}} & 2 \leq i \leq d \end{cases}. \tag{20}
$$

We will now show that sampling from the distribution with density proportional to $\exp(-f_{\mathrm{hard}})$ is hard. First, notice that the function $f_{\mathrm{hard}}$ has condition number $\kappa$ and is coordinate-wise separable. It immediately follows from Lemma 1 that the spectral gap (defined in (2)) of the MALA Markov chain is governed by the step size $h$ as follows.

**Corollary 3.** *The spectral gap of the MALA Markov chain for sampling from the density proportional to $\exp(-f_{\mathrm{hard}})$, where $f_{\mathrm{hard}}$ is defined in (20), is $O(h + h^2)$.*

For the remainder of the section, we focus on upper bounding (19) over a large region according to the density proportional to $\exp(-f_{\mathrm{hard}})$. Recall $\{f_i\}_{i\in[d]}$ are the summands of $f_{\mathrm{hard}}$. For a fixed $x$, consider the random variables $S_i^x$:

$$
S_i^x = -f_i([x_g]_i) + f_i(x_i) - \frac{1}{2}(x_i - [x_g]_i)(f_i'(x_i) + f_i'([x_g]_i)).
$$

It is easy to check that for a given realization of $g$, we have

$$
\sum_{i\in[d]} S_i^x = -f(x_g) + f(x) - \frac{1}{2}\langle x - x_g, \nabla f(x) + \nabla f(x_g)\rangle,
$$

where the right-hand side of the above display is the left-hand side of (19). We bound the expectation of $\sum_{i\in[d]} S_i^x$, and its deviation from its expectation, in Lemma 7 and Lemma 8 respectively.

**Lemma 7.** *For any fixed* $x \in \left\{ x \mid -\frac{1}{2}\pi\sqrt{h} + 2\pi k_i\sqrt{h} \leq x_i \leq \frac{1}{2}\pi\sqrt{h} + 2\pi k_i\sqrt{h}, k_i \in \mathbb{N}, \forall 2 \leq i \leq d \right\}$ *and* $h \leq 1$, *the random variables* $S_i^x$, $1 \leq i \leq d$ *satisfy*

$$
\mathbb{E}_{g \sim \mathcal{N}(0,\mathbf{I})} \left[ S_i^x \right] \leq \begin{cases} 0 & i = 1 \\ -0.08h\kappa \cos \frac{x_i}{\sqrt{h}} & 2 \leq i \leq d \end{cases}.
$$

*Proof.* We remark that the condition on $x$ simply enforces coordinatewise in $2 \leq i \leq d$, $\cos \frac{x_i}{\sqrt{h}} > 0$. Consider some coordinate $2 \leq i \leq d$: since $[x_g]_i = x_i + \sqrt{2h}g_i$,

$$
\begin{aligned}
S_i^x &= -f_i\left(x_i + \sqrt{2h}g_i\right) + f_i(x_i) + \frac{\sqrt{2h}}{2}g_i\left(f_i'(x_i) + f_i'\left(x_i + \sqrt{2h}g_i\right)\right) \\
&= -\frac{\kappa}{3}\left(x_i + \sqrt{2h}g_i\right)^2 + \frac{\kappa h}{3}\cos\left(\frac{x_i}{\sqrt{h}} + \sqrt{2}g_i\right) + \frac{\kappa}{3}x_i^2 - \frac{\kappa h}{3}\cos\left(\frac{x_i}{\sqrt{h}}\right) \\
&\quad + \frac{\sqrt{2h}}{2}g_i\left(\frac{4\kappa}{3}x_i + \frac{2\sqrt{2h}\kappa}{3}g_i + \frac{\kappa\sqrt{h}}{3}\sin\left(\frac{x_i}{\sqrt{h}} + \sqrt{2}g_i\right) + \frac{\kappa\sqrt{h}}{3}\sin\left(\frac{x_i}{\sqrt{h}}\right)\right) \\
&= \frac{\kappa h}{3}\left(\cos\left(\frac{x_i}{\sqrt{h}} + \sqrt{2}g_i\right) - \cos\left(\frac{x_i}{\sqrt{h}}\right)\right) + \frac{\sqrt{2h}\kappa}{6}g_i\left(\sin\left(\frac{x_i}{\sqrt{h}} + \sqrt{2}g_i\right) + \sin\left(\frac{x_i}{\sqrt{h}}\right)\right)
\end{aligned}
$$

Here, we used that the quadratic terms in the second and third lines cancel (this also follows from examining the proof of Lemma [2](#)):

$$
-\frac{\kappa}{3}\left(x_i + \sqrt{2h}g_i\right)^2 + \frac{\kappa}{3}x_i^2 + \frac{\sqrt{2h}}{2}g_i\left(\frac{4\kappa}{3}x_i + \frac{2\sqrt{2h}\kappa}{3}g_i\right) = 0.
$$

By a direct computation, taking an expectation over $g_i \sim \mathcal{N}(0,1)$ yields

$$
\mathbb{E}_{g_i \sim \mathcal{N}(0,1)}\left[\cos\left(\frac{x_i}{\sqrt{h}} + \sqrt{2}g_i\right)\right] = \frac{\cos\left(\frac{x_i}{\sqrt{h}}\right)}{\exp(1)},
$$

$$
\mathbb{E}_{g_i \sim \mathcal{N}(0,1)}\left[\sin\left(\frac{x_i}{\sqrt{h}} + \sqrt{2}g_i\right)g_i\right] = \frac{\sqrt{2}\cos\left(\frac{x_i}{\sqrt{h}}\right)}{\exp(1)}.
$$

Putting these pieces together,

$$
\mathbb{E}_{g_i \sim \mathcal{N}(0,1)}\left[S_i^x\right] = \frac{\kappa h}{3}\left(\frac{2}{\exp(1)} - 1\right)\cos\left(\frac{x_i}{\sqrt{h}}\right) \leq -0.08\kappa h \cos\left(\frac{x_i}{\sqrt{h}}\right).
$$

Here, we used $\cos \frac{x_i}{\sqrt{h}} > 0$. For $i = 1$, Lemma [2](#) shows $\mathbb{E}_{g_1 \sim \mathcal{N}(0,1)}\left[S_1^x\right] = 0$. $\qquad\square$

**Lemma 8.** *With probability at least* $1 - \frac{1}{d^5}$ *over the randomness of* $g \sim \mathcal{N}(0,\mathbf{I})$,

$$
\sum_{i \in [d]} S_i^x - \mathbb{E}_{g \sim \mathcal{N}(0,\mathbf{I})}\left[\sum_{i \in [d]} S_i^x\right] \leq 10h\kappa\sqrt{d\log d}.
$$

*Proof.* By Lemma 6, for coordinate $1 \leq i \leq d$,

$$S_i^x = -2h \int_0^1 \left( \frac{1}{2} - s \right) f_i''([x_s]_i) ds g_i^2, \text{ where } \left| 2h \int_0^1 \left( \frac{1}{2} - s \right) f_i''([x_s]_i) ds \right| \leq \frac{h\kappa}{2}.$$

We attained the latter bound by smoothness. Now, each random variable $S_i^x - \mathbb{E}[S_i^x]$ is sub-exponential with parameter $\frac{h\kappa}{2}$ (for coordinates where the coefficient is negative, note the negation of a sub-exponential random variable is still sub-exponential). Hence, by Fact 3,

$$\Pr \left[ \sum_{i \in [d]} S_i^x - \mathbb{E}_{g \sim \mathcal{N}(0, \mathbf{I})} \left[ \sum_{i \in [d]} S_i^x \right] \geq 10 h\kappa \sqrt{d \log d} \right] \leq \frac{1}{d^5}.$$

$\square$

Now, we build a bad set $\Omega_{\text{hard}}$ with lower bounded measure that starting from a point $x \in \Omega_{\text{hard}}$, with high probability, $-\mathbb{E}_{g \sim \mathcal{N}(0, \mathbf{I})} \left[ \sum_{i \in [d]} S_i^x \right]$ is negative:

$$\Omega_{\text{hard}} = \left\{ x \mid |x_1| \leq 2, \forall 2 \leq i \leq d, \exists k_i \in \mathbb{Z}, |k_i| \leq \left\lfloor \frac{5}{\pi\sqrt{h\kappa}} \right\rfloor, \text{ such that} \right.$$

$$\left. -\frac{9}{20}\pi\sqrt{h} + 2\pi k_i \sqrt{h} \leq x_i \leq \frac{9}{20}\pi\sqrt{h} + 2\pi k_i \sqrt{h} \right\}. \quad (21)$$

In other words, $\Omega_{\text{hard}}$ is the set of points where $\cos x_i$ is large for $2 \leq i \leq d$, and coordinates are bounded. We first lower bound the measure of $\Omega_{\text{hard}}$, and show $\|\nabla f(x)\|_2$ is small within $\Omega_{\text{hard}}$. Our measure lower bound will not be used in this section, but will become relevant in Section 5.

**Lemma 9.** *Let $h \leq \frac{1}{10000\pi^2\kappa}$. Let $\pi^*$ have log-density $-f_{\text{hard}}$ (20). Then, $\pi^*(\Omega_{\text{hard}}) \geq \exp(-d)$. Moreover, for all $x \in \Omega_{\text{hard}}$, $\|\nabla f(x)\|_2 \leq 10\sqrt{\kappa d}$.*

*Proof.* We first consider a superset of $\Omega_{\text{hard}}$. We define the set, for $K := \left\lfloor \frac{5}{\pi\sqrt{h\kappa}} \right\rfloor$,

$$\Omega' = \left\{ x \mid |x_1| \leq 2, \forall 2 \leq i \leq d, -\frac{9}{20}\pi\sqrt{h} - 2\pi K\sqrt{h} \leq x_i \leq \frac{9}{20}\pi\sqrt{h} + 2\pi K\sqrt{h} \right\}.$$

It is easy to verify that $\Omega' \supseteq \Omega_{\text{hard}}$. We first show $\pi^*(\Omega')$ is lower bounded by $1.1^{-d}$. Since $f_{\text{hard}}$ is separable, the coordinates are independent, so it suffices to show each one-dimensional measure is lower bounded by $\frac{1}{1.1}$. This is a standard computation of Gaussian measure for the first coordinate, which we omit. For $2 \leq i \leq d$, since the marginal distribution is $\frac{\kappa}{3}$-strongly logconcave, it is sub-Gaussian with parameter $\frac{3}{\kappa}$ (see Lemma 1, [DCWY18]). It follows from a standard sub-Gaussian tail bound that the measure of the set $|x_i| \leq \frac{9}{\sqrt{\kappa}}$ is at least $\frac{1}{1.1}$. For our choice of $K$, by assumption on $h$, $2\pi\sqrt{h}K \geq \frac{10}{\sqrt{\kappa}} - 2\pi\sqrt{h} \geq \frac{9}{\sqrt{\kappa}}$. Combining across coordinates gives $\pi^*(\Omega') \geq 1.1^{-d}$.

Next, we lower bound $\frac{\pi^*(\Omega_{\text{hard}})}{\pi^*(\Omega')}$. We divide the support of the set $\Omega_{\text{hard}}$ and $\Omega'$ into small disjoint regions and bound $\frac{\pi^*(\Omega_{\text{hard}})}{\pi^*(\Omega')}$ for each small region and each coordinate separately. For $2 \leq i \leq d$,

$k \in \left[ -\left\lfloor \frac{5}{\pi\sqrt{h\kappa}} \right\rfloor - 1, \left\lfloor \frac{5}{\pi\sqrt{h\kappa}} \right\rfloor \right]$, $k \in \mathbb{Z}$, let

$$\Omega'^{(i,k)} = \left( 2\pi k\sqrt{h}, 2\pi(k+1)\sqrt{h} \right],$$

and

$$\Omega_{\text{hard}}^{(i,k)} = \left( 2\pi k\sqrt{h}, 2\pi k\sqrt{h} + \frac{9}{20}\pi\sqrt{h} \right] \cup \left[ 2\pi(k+1)\sqrt{h} - \frac{9}{20}\pi\sqrt{h}, 2\pi(k+1)\sqrt{h} \right].$$

Then, letting $\pi_i^*$ be the marginal of $\pi^*$ on coordinate $i$, we have

$$
\begin{aligned}
\frac{\pi_i^*\left(\Omega_{\text{hard}}^{(i,k)}\right)}{\pi_i^*\left(\Omega'^{(i,k)}\right)} &= \frac{\int_{2\pi k\sqrt{h}}^{2\pi k\sqrt{h}+\frac{9}{20}\pi\sqrt{h}} \exp\left(-\frac{\kappa}{3}x_i^2 + \frac{\kappa h}{3}\cos\frac{x_i}{\sqrt{h}}\right)dx_i + \int_{2\pi(k+1)\sqrt{h}-\frac{9}{20}\pi\sqrt{h}}^{2\pi(k+1)\sqrt{h}} \exp\left(-\frac{\kappa}{3}x_i^2 + \frac{\kappa h}{3}\cos\frac{x_i}{\sqrt{h}}\right)dx_i}{\int_{2\pi k\sqrt{h}}^{2\pi(k+1)\sqrt{h}} \exp\left(-\frac{\kappa}{3}x_i^2 + \frac{\kappa h}{3}\cos\frac{x_i}{\sqrt{h}}\right)dx_i} \\
&\geq \frac{\int_{2\pi k\sqrt{h}}^{2\pi k\sqrt{h}+\frac{9}{20}\pi\sqrt{h}} \exp\left(-\frac{\kappa}{3}x_i^2\right)dx_i + \int_{2\pi(k+1)\sqrt{h}-\frac{9}{20}\pi\sqrt{h}}^{2\pi(k+1)\sqrt{h}} \exp\left(-\frac{\kappa}{3}x_i^2\right)dx_i}{\int_{2\pi k\sqrt{h}}^{2\pi(k+1)\sqrt{h}} \exp\left(-\frac{\kappa}{3}x_i^2\right)dx_i \exp\left(\frac{\kappa h}{3}\right)} \\
&\geq \exp\left(-\frac{\kappa h}{3}\right) \cdot \frac{\frac{9}{10}\pi\sqrt{h}}{2\pi\sqrt{h}} \cdot \frac{\exp\left(-\frac{\kappa}{3}\left(2\pi(k+1)\sqrt{h}\right)^2\right)}{\exp\left(-\frac{\kappa}{3}\left(2\pi k\sqrt{h}\right)^2\right)} \geq 0.42.
\end{aligned}
$$

The second step used $\cos\frac{x_i}{\sqrt{h}} \geq 0$ for $x_i \in \Omega_{\text{hard}}^{(i,k)}$. The fourth step used the assumption $\kappa h \leq \frac{1}{10000\pi^2}$. Finally, letting $\Omega'^{(i)}$ and $\Omega_{\text{hard}}^{(i)}$ be the projections of $\Omega'$ and $\Omega_{\text{hard}}$ on the $i^{\text{th}}$ coordinate. For any $x_i \in \Omega_i'$ with $x \notin \Omega'^{(i,k)}$, and for all integers $k \in [-K-1, K]$, $x_i \in \Omega_{\text{hard}}^{(i)}$, so $\frac{\pi^*\left(\Omega_{\text{hard}}^{(i)}\right)}{\pi^*\left(\Omega'^{(i)}\right)} \geq 0.42$. Since the coordinates are independent under $\pi^*$, $\frac{\pi^*(\Omega_{\text{hard}})}{\pi^*(\Omega')} \geq 0.42^d$. Combining our lower bounds,

$$\pi^*\left(\Omega_{\text{hard}}\right) = \pi^*(\Omega')\frac{\pi^*(\Omega_{\text{hard}})}{\pi^*(\Omega')} \geq \left(\frac{1.1}{0.42}\right)^{-d} \geq \exp(-d).$$

Finally, we bound $\|\nabla f_{\text{hard}}(x)\|_2$ for $x \in \Omega'$, from the definition of $f_{\text{hard}}$ (20),

$$\|\nabla f_{\text{hard}}(x)\|_2 = \sqrt{f_1'(x)^2 + \sum_{i=2}^{d} f_i'(x)^2} = \sqrt{x_1^2 + \sum_{i=2}^{d}\left(\frac{2\kappa}{3}x_i + \frac{\kappa\sqrt{h}}{3}\sin\frac{x_i}{\sqrt{h}}\right)^2}.$$

Then directly plugging in the definition of $\Omega_{\text{hard}}$ and using $|\sin c| \leq |c|$ for all $c$,

$$\|\nabla f_{\text{hard}}(x)\|_2 \leq \sqrt{1.5^2 + (d-1)\left(9\sqrt{\kappa}\right)^2} \leq 10\sqrt{\kappa d}.$$

$\square$

Finally, we combine the bounds we derived to show the acceptance probability is small within $\Omega_{\text{hard}}$.

**Lemma 10.** *Let $h = o\left(\frac{1}{\kappa\log d}\right)$. For any $x \in \Omega_{\text{hard}}$, let $y = x - h\nabla f_{\text{hard}}(x) + \sqrt{2h}g$ for $g \sim \mathcal{N}(0, \mathbf{I})$.*

*With probability at least $1 - \frac{2}{d^5}$, we have*

$$f_{\text{hard}}(x) - f_{\text{hard}}(y) + \frac{1}{4h}\left(\|y - (x - h\nabla f_{\text{hard}}(x))\|_2^2 - \|x - (y - h\nabla f_{\text{hard}}(y))\|_2^2\right) = -\Omega\left(h\kappa d\right).$$

*Proof.* By combining Lemma 5 and the decomposition (14), the conclusion is equivalent to showing that the following quantity is $-\Omega(h\kappa d)$:

$$-f_{\text{hard}}(x_g) + f_{\text{hard}}(x) - \frac{1}{2}\langle x - x_g, \nabla f_{\text{hard}}(x) + \nabla f_{\text{hard}}(x_g)\rangle$$

$$+f_{\text{hard}}(x_g) - f_{\text{hard}}(y) - \frac{1}{2}\langle x - x_g, \nabla f_{\text{hard}}(y) - \nabla f_{\text{hard}}(x_g)\rangle$$

$$-\frac{1}{2}\langle x_g - y, \nabla f_{\text{hard}}(x) + \nabla f_{\text{hard}}(y)\rangle + \frac{h}{4}\|\nabla f_{\text{hard}}(x)\|_2^2 - \frac{h}{4}\|\nabla f_{\text{hard}}(y)\|_2^2.$$

For $x \in \Omega_{\text{hard}}$, every $x_i$ for $2 \leq i \leq d$ has $\cos\frac{x_i}{\sqrt{h}}$ bounded away from 0 by a constant and hence combining Lemmas 7 and 8 implies the first line is $-\Omega(h\kappa d)$ with probability at least $\frac{1}{d^5}$. Regarding the second and third lines, Lemma 6 shows that it suffices to bound (over the set $\Omega_{\text{hard}}$)

$$(h^2\kappa + h^3\kappa^2)\|\nabla f_{\text{hard}}(x)\|_2^2 + \left(h^{1.5}\kappa + h^{2.5}\kappa^2\right)\|g\|_2\|\nabla f_{\text{hard}}(x)\|_2 + h^2\kappa^2\|g\|_2^2 = o(h\kappa d).$$

Fact 2 implies $\|g\|_2 \leq \sqrt{2d}$ with probability at least $1 - \frac{1}{d^5}$. Combining this bound, the bound on $\|\nabla f_{\text{hard}}(x)\|_2$ from Lemma 9, and the upper bound on $h$ yields the conclusion. $\qquad\square$

We conclude by giving the main result of this section.

**Proposition 2.** *For $h = o(\frac{1}{\kappa \log d})$, there is a target density on $\mathbb{R}^d$ whose negative log-density always has Hessian eigenvalues in $[1, \kappa]$, such that the relaxation time of MALA is $\Omega(\frac{\kappa d}{\log d})$.*

*Proof.* The proof is identical to that of Theorem 1, where we use Lemma 10 in place of Proposition 1. $\qquad\square$

**Theorem 2.** *For every step size, there is a target density on $\mathbb{R}^d$ whose negative log-density always has Hessian eigenvalues in $[1, \kappa]$, such that the relaxation time of MALA is $\Omega(\frac{\kappa d}{\log d})$.*

*Proof.* This is immediate from combining Theorem 1 (with the hard function $f_{\text{hq}}$ in the range $h = \Omega(\frac{1}{\kappa \log d})$) and Proposition 2 (with the hard function $f_{\text{hard}}$ in the range $h = o(\frac{1}{\kappa \log d})$). $\qquad\square$

## 5 Mixing time lower bound for MALA

In this section, we derive a mixing time lower bound for MALA. Concretely, we show that for any step size $h$, there is a hard distribution $\pi^* \propto \exp(-f)$ such that $\nabla^2 f$ always has eigenvalues in $[1, \kappa]$, yet there is a $\exp(d)$-warm start $\pi_0$ such that the chain cannot mix in $o(\frac{\kappa d}{\log^2 d})$ iterations, starting from $\pi_0$. We begin by giving such a result for $h = O(\frac{\log d}{\kappa d})$ in Section 5.1, and combine it with our developments in Sections 3 and 4 to prove our main mixing time result.

## 5.1 Mixing time lower bound for small $h$

Throughout this section, let $h = O\left(\frac{\log d}{\kappa d}\right)$, and let $\pi^* = \mathcal{N}(0, \mathbf{I})$ be the standard $d$-dimensional multivariate Gaussian. We will let $\pi_0$ be the marginal distribution of $\pi^*$ on the set

$$\Omega := \left\{ x \mid \|x\|_2^2 \leq \frac{1}{2}d \right\}.$$

Recall from Lemma 4 that $\pi_0$ is a $\exp(d)$-warm start. Our main proof strategy will be to show that for such a small value of $h$, after $T = O(\frac{\kappa d}{\log^2 d})$ iterations, with constant probability both of the following events happen: no rejections occur throughout the Markov chain, and $\|x_t\|_2^2 \leq \frac{9}{10}d$ holds for all $t \in [T]$. Combining these two facts will demonstrate our total variation lower bound.

**Lemma 11.** *Let $\{x_t\}_{0 \leq t < T}$ be the iterates of the MALA Markov chain with step size $h = O\left(\frac{\log d}{\kappa d}\right)$, for $T = o(\frac{\kappa d}{\log^2 d})$ and $x_0 \sim \pi_0$. With probability at least $\frac{99}{100}$, both of the following events occur:*

1. *Throughout the Markov chain, $\|x_t\|_2 \leq 0.9\sqrt{d}$.*
2. *Throughout the Markov chain, the Metropolis filter never rejected.*

*Proof.* We inductively bound the failure probability of the above events in every iteration by $\frac{0.01}{T}$, which will yield the claim via a union bound. Take some iteration $t + 1$, and note that by triangle inequality, and assuming all prior iterations did not reject,

$$\|x_{t+1}\|_2 \leq \|x_0\|_2 + h\sum_{s=0}^{t} \|x_s\|_2 + \sqrt{2h}\left\|\sum_{s=0}^{t} g_s\right\|_2 \leq \|x_0\|_2 + 0.9hT\sqrt{d} + \sqrt{2h}\,\|G_t\|_2 \leq 0.8\sqrt{d} + \sqrt{2h}\,\|G_t\|_2\,.$$

Here, we applied the inductive hypothesis on all $\|x_s\|_2$, the initial bound $\|x_0\|_2 \leq \sqrt{\frac{1}{2}d}$, and that $hT = o(1)$ by assumption. We also defined $G_t = \sum_{s=0}^{t} g_s$, where $g_s$ is the random Gaussian used by MALA in iteration $s$; note that by independence, $G_t \sim \mathcal{N}(0, t+1)$. By Fact 2, with probability at least $\frac{1}{200T}$, $\|G_t\|_2 \leq 2\sqrt{Td}$, and hence $0.8\sqrt{d} + \sqrt{2h}\,\|G_t\|_2 \leq 0.9\sqrt{d}$, as desired.

Next, we prove that with probability $\geq 1 - \frac{1}{200T}$, step $t$ does not reject. This concludes the proof by union bounding over both events in iteration $t$, and then union bounding over all iterations. By the calculation in Lemma 2, the accept probability is

$$\min\left(1, \exp\left(\frac{h}{4}\left((2h - h^2)\|x_t\|_2^2 - 2h\|g\|_2^2 - 2\sqrt{2h}(1-h)\langle x_t, g\rangle\right)\right)\right).$$

We lower bound the argument of the exponential as follows. With probability at least $1 - d^{-5} \geq 1 - \frac{1}{400T}$, Facts 1 and 2 imply both of the events $\|g\|_2^2 \leq 2d$ and $\langle x_t, g\rangle \leq 10\sqrt{\log d}\,\|x\|_2$ occur. Conditional on these bounds, we compute (using $2h \geq h^2$ and the assumption $\|x_t\|_2 \leq 0.9\sqrt{d}$)

$$\left(2h - h^2\right)\|x_t\|_2^2 - 2h\|g\|_2^2 - 2\sqrt{2h}(1-h)\langle x_t, g\rangle \geq -4hd - 40\sqrt{hd\log d} \geq -44\log d.$$

Hence, the acceptance probability is at least

$$\exp\left(-11h\log d\right) \geq 1 - \frac{1}{400T},$$

by our choice of $T$ with $Th \log d = o(1)$, concluding the proof. $\qquad\square$

**Proposition 3.** *The MALA Markov chain with step size $h = O\left(\frac{\log d}{\kappa d}\right)$ requires $\Omega(\frac{\kappa d}{\log^2 d})$ iterations to reach total variation distance $\frac{1}{e}$ to $\pi^*$, starting from $\pi_0$.*

*Proof.* Let $\tilde{\pi}$ be the distribution of the MALA Markov chain after $T = o(\frac{\kappa d}{\log^2 d})$ steps without applying a Metropolis filter in any step, and let $\hat{\pi}$ be the distribution after applying the actual MALA chain (including rejections). To show $\|\hat{\pi} - \pi^*\|_{\mathrm{TV}} \geq \frac{1}{e}$, it suffices to show the bounds

$$\|\tilde{\pi} - \pi^*\|_{\mathrm{TV}} \geq \frac{2}{5}, \ \|\tilde{\pi} - \hat{\pi}\|_{\mathrm{TV}} \leq 0.01,$$

and then we apply the triangle inequality. By the coupling characterization of total variation, the second bound follows immediately from the second claim in Lemma 11, wherein we couple the two distributions whenever a rejection does not occur. To show the first bound, the measure of

$$\Omega_{\mathrm{large}} := \left\{x \mid \|x\|_2^2 \geq 0.81 d\right\}$$

according to $\pi^*$ is at least 0.99 by Fact 2, and according to $\tilde{\pi}$ it can be at most 0.01 by the first conclusion of Lemma 11. This yields the bound via the definition of total variation. $\qquad\square$

## 5.2 Proof of Theorem 3

Finally, we put together the techniques of Sections 3, 4, and 5.1 to prove Theorem 3.

**Theorem 3.** *For every step size, there is a target density on $\mathbb{R}^d$ whose negative log-density always has Hessian eigenvalues in $[1, \kappa]$, such that MALA initialized at an $\exp(d)$-warm start requires $\Omega(\frac{\kappa d}{\log^2 d})$ iterations to reach $e^{-1}$ total variation distance to the stationary distribution.*

*Proof.* We consider three ranges of $h$. First, if $h = \Omega\left(\frac{1}{\kappa \log d}\right)$, we use the hard function $f_{\mathrm{hq}}$ and the hard set in (12), which has measure at least $\exp(-d)$ according to the stationary distribution by Lemma 4. Then, applying Proposition 1 demonstrates that the chance the Markov chain can move over $d^5$ iterations is $o(\frac{1}{d})$, and hence it does not reach total variation $\frac{1}{e}$ in this time. Next, if $h = o\left(\frac{1}{\kappa \log d}\right) \cap \omega\left(\frac{\log d}{\kappa d}\right)$, we use the hard function $f_{\mathrm{hard}}$ and the hard set in (21), which has measure at least $\exp(-d)$ by Lemma 9. Applying Lemma 10 again implies the chain does not mix in $d^5$ iterations. Finally, if $h = O\left(\frac{\log d}{\kappa d}\right)$, applying Proposition 3 yields the claim. $\qquad\square$

# 6 Lower bounds for HMC

In this section, we derive a lower bound on the spectral gap of HMC. We first analyze some general structural properties of HMC in Section 6.1, as a prelude to later sections. We then provide a lower bound for HMC on quadratics in Section 6.2, with any number of leapfrog steps $K$.

### 6.1 Structure of HMC: a detour to Chebyshev polynomials

We begin our development with a bound on the acceptance probability for general HMC Markov chains. Recall from (6) that this probability is (for $\mathcal{H}(x,v) := f(x) + \frac{1}{2} \|v\|_2^2$)

$$\min\left\{1, \frac{\exp\left(-\mathcal{H}(x_K, v_K)\right)}{\exp\left(-\mathcal{H}(x_0, v_0)\right)}\right\}. \tag{6}$$

We first state a helper calculation straightforwardly derived from the exposition in Section 2.4.

**Fact 4.** *One step of the HMC Markov chain starting from $x_0$ generates iterates $\{(v_{k-\frac{1}{2}}, x_k, v_k)\}_{0 \le k \le K}$ defined recursively by the closed-form equations:*

$$v_{k-\frac{1}{2}} = v_0 - \frac{\eta}{2}\nabla f(x_0) - \eta \sum_{j \in [k-1]} \nabla f(x_j),$$

$$v_k = v_0 - \frac{\eta}{2}\nabla f(x_0) - \eta \sum_{j \in [k-1]} \nabla f(x_j) - \frac{\eta}{2}\nabla f(x_k),$$

$$x_k = x_0 + \eta k v_0 - \frac{\eta^2 k}{2}\nabla f(x_0) - \eta^2 \sum_{j \in [k-1]} (k-j)\nabla f(x_j).$$

When expanding the acceptance probability (6) using the equations in Fact 4, many terms conveniently cancel, which we capture in Lemma 12. This phenomenon underlies the improved performance of HMC on densities with highly-Lipschitz Hessians [CDWY19].

**Lemma 12.** *For the iterates given by Fact 4,*

$$\mathcal{H}(x_0, v_0) - \mathcal{H}(x_K, v_K) = \sum_{0 \le k \le K-1} \left( f(x_k) - f(x_{k+1}) + \frac{1}{2}\langle \nabla f(x_k) + \nabla f(x_{k+1}), x_{k+1} - x_k \rangle \right)$$

$$+ \frac{\eta^2}{8}\|\nabla f(x_0)\|_2^2 - \frac{\eta^2}{8}\|\nabla f(x_K)\|_2^2.$$

*Proof.* Recall $\mathcal{H}(x_0, v_0) - \mathcal{H}(x_K, v_K) = f(x_0) - f(x_K) + \frac{1}{2}\|v_0\|_2^2 - \frac{1}{2}\|v_K\|_2^2$. We begin by expanding

$$
\begin{aligned}
\frac{1}{2}\|v_0\|_2^2 - \frac{1}{2}\|v_K\|_2^2 &= \frac{1}{2}\|v_0\|_2^2 - \frac{1}{2}\left\|v_0 - \frac{\eta}{2}\nabla f(x_0) - \eta \sum_{j \in [K-1]} \nabla f(x_j) - \frac{\eta}{2}\nabla f(x_K)\right\|_2^2 \\
&= \eta\left\langle v_0, \frac{1}{2}\nabla f(x_0) + \sum_{j \in [K-1]} \nabla f(x_j) + \frac{1}{2}\nabla f(x_K)\right\rangle \\
&\quad - \frac{\eta^2}{2}\left\|\frac{1}{2}\nabla f(x_0) + \sum_{j \in [K-1]} \nabla f(x_j) + \frac{1}{2}\nabla f(x_K)\right\|_2^2 \\
&= \frac{\eta}{2}\sum_{0 \le k \le K-1} \langle v_0, \nabla f(x_k) + \nabla f(x_{k+1})\rangle \\
&\quad - \frac{\eta^2}{2}\sum_{0 \le k \le K-1} \left\langle \nabla f(x_k) + \nabla f(x_{k+1}), \frac{1}{2}\nabla f(x_0) + \sum_{j \in [k]} \nabla f(x_j)\right\rangle \\
&\quad + \frac{\eta^2}{8} \langle \nabla f(x_0) - \nabla f(x_K), \nabla f(x_0) + \nabla f(x_K)\rangle.
\end{aligned}
$$

Here the first equality used Fact 4. Moreover, for each $0 \le k \le K-1$, by Fact 4

$$
\begin{aligned}
\frac{1}{2}\langle \nabla f(x_k) + \nabla f(x_{k+1}), x_{k+1} - x_k\rangle &= \frac{\eta}{2}\langle \nabla f(x_k) + \nabla f(x_{k+1}), v_0\rangle \\
&\quad - \frac{\eta^2}{2}\left\langle \nabla f(x_k) + \nabla f(x_{k+1}), \frac{1}{2}\nabla f(x_0) + \sum_{j \in [k]} \nabla f(x_j)\right\rangle.
\end{aligned}
$$

Combining yields the result. $\qquad\square$

We state a simple corollary of Lemma 12 in the case of quadratics.

**Corollary 4.** *For $f(x) = \frac{1}{2}x^\top \mathbf{A} x$, the iterates given by Fact 4 satisfy*

$$
\mathcal{H}(x_0, v_0) - \mathcal{H}(x_K, v_K) = \frac{\eta^2}{8}\|\nabla f(x_0)\|_2^2 - \frac{\eta^2}{8}\|\nabla f(x_K)\|_2^2.
$$

*Proof.* It suffices to observe that for any two points $x, y \in \mathbb{R}^d$,

$$
f(x) - f(y) + \frac{1}{2}\langle \nabla f(x) + \nabla f(y), y - x\rangle = \frac{1}{2}x^\top \mathbf{A} x - \frac{1}{2}y^\top \mathbf{A} y + \frac{1}{2}\langle \mathbf{A}(x+y), y - x\rangle = 0.
$$

$\qquad\square$

Finally, it will be convenient to have a more explicit form of iterates in the case of quadratics, which follows directly from examining the recursion in Fact 4.

**Lemma 13.** *For $f(x) = \frac{1}{2}x^\top \mathbf{A}x$, the iterates $\{x_k\}_{0 \le k \le K}$ given by Fact 4 satisfy*

$$x_k = \left( \sum_{0 \le j \le k} D_{j,k}(\eta^2 \mathbf{A})^j \right) x_0 + \left( \eta \sum_{0 \le j \le k-1} E_{j,k}(\eta^2 \mathbf{A})^j \right) v_0, \tag{22}$$

*where $D_{j,k} := (-1)^j \cdot \dfrac{k}{k+j} \cdot \dbinom{k+j}{2j}$, $E_{j,k} := (-1)^j \cdot \dbinom{k+j}{2j+1}$.*

*Proof.* This formula can be verified to match the recursions of Fact 4 by checking the base cases $D_{0,k} = 1$, $D_{1,k} = -\frac{k^2}{2}$, $E_{0,k} = k$, and (where $D_{j,k} := 0$ for $j > k$ and $E_{j,k} := 0$ for $j \ge k$)

$$D_{j,k} = - \sum_{i \in [k-1]} (k-i)D_{j-1,i}, \; E_{j,k} = - \sum_{i \in [k-1]} (k-i)E_{j-1,i}.$$

In particular, by using the third displayed line of Fact 4, the coefficient of $(\eta^2 \mathbf{A})^j x_0$ in $x_k$ for $j \ge 2$ is the negated sum of the coefficients of $(\eta^2 \mathbf{A})^{j-1}$ in all $(k-i)x_i$. Similarly, the coefficient of $\eta(\eta^2 \mathbf{A})^j v_0$ in $x_k$ for $j \ge 1$ is the negated sum of the coefficients of $\eta(\eta^2 \mathbf{A})^{j-1}$ in all $(k-i)x_i$. The displayed coefficient identities follow from the binomial coefficient identities

$$\frac{k}{k+j}\binom{k+j}{2j} = \sum_{j-1 \le i \le k-1} \frac{(k-i)i}{i+j-1}\binom{i+j-1}{2j-2}, \; \binom{k+j}{2j+1} = \sum_{j \le i \le k-1}(k-i)\binom{i+j-1}{2j-1}.$$

$\square$

Lemma 13 motivates the definition of the polynomials

$$p_k(z) := \sum_{0 \le j \le k} D_{j,k} z^j, \; q_k(z) := \sum_{0 \le j \le k-1} E_{j,k} z^j. \tag{23}$$

In this way, at least in the case when $\mathbf{A} = \mathbf{diag}(\lambda)$ for a vector of eigenvalues $\lambda \in \mathbb{R}^d$, we can concisely express the coordinates of iterates in (22) by

$$[x_k]_i = p_k(\eta^2 \lambda_i)[x_0]_i + \eta q_k(\eta^2 \lambda_i)[v_0]_i. \tag{24}$$

Interestingly, the polynomial $p_k$ turns out to have a close relationship with the $k^{\text{th}}$ *Chebyshev polynomial* (of the first kind), which we denote by $T_k$. Similarly, the polynomial $q_k$ is closely related to the $(k-1)^{\text{th}}$ *Chebyshev polynomial* of the second kind, denoted $U_{k-1}$. The relationship between the Chebyshev polynomials and the phenomenon of *acceleration* for optimizing quadratics via first-order methods has been known for some time (see e.g. [Har13, Bac19] for discussions), and we find it interesting to further explore this relationship. Concretely, the following identities hold.

**Lemma 14.** *Following definitions* (22), (23),

$$p_k(z) = T_k\left(1 - \frac{z}{2}\right), \; q_k(z) = U_{k-1}\left(1 - \frac{z}{2}\right).$$

*Proof.* It is easy to check $p_0(z) = 1$ and $p_1(z) = 1 - \frac{z}{2}$, so the former conclusion would follow from

$$p_{k+1}(z) = (2-z)p_k(z) - p_{k-1}(z) \iff D_{j,k+1} = 2D_{j,k} - D_{j-1,k} - D_{j,k-1},$$

following well-known recursions defining the Chebyshev polynomials of the first kind. This identity can be verified by direct expansion. Moreover, for the latter conclusion, recalling the definition of Morgan-Voyce polynomials of the first kind $B_k(z)$, we can directly match $q_k(z) = B_{k-1}(-z)$. The conclusion follows from Section 4 of [AJ94], which shows $B_{k-1}(-z) = U_{k-1}(1 - \frac{z}{2})$ as desired (note that in the work [AJ94], the indexing of Chebyshev polynomials is off by one from ours). $\square$

Now for $z = \eta^2 \lambda_i$, we have from (24) and Lemma 14 that $[x_k]_i = \pm[x_0]_i$ precisely when

$$p_k(z) = T_k\left(1 - \frac{z}{2}\right) = \pm 1, \ q_k(z) = U_{k-1}\left(1 - \frac{z}{2}\right) = 0.$$

Hence, this occurs whenever $1 - \frac{z}{2}$ is both an *extremal point* of $T_k$ in the range $[-1, 1]$ and a root of $U_{k-1}$. Both of these occur exactly at the points $\cos(\frac{j}{k}\pi)$, for $0 \le j \le k$.

**Proposition 4.** *For $\kappa \ge \pi^2$ and $K \ge 2$, no $K$-step HMC Markov chain with step size $1 \ge \eta^2 \ge \frac{\pi^2}{\kappa K^2}$ can mix in finite time for all densities on $\mathbb{R}^d$ whose negative log-density's Hessian has eigenvalues between 1 and $\kappa$ for all points $x \in \mathbb{R}^d$, initialized at a constant-warm start.*

*Proof.* Fix a value of $1 \ge \eta \ge \sqrt{\frac{\pi^2}{\kappa K^2}}$. We claim there exists a $1 \le j \le K - 1$ such that for

$$\lambda := \frac{2\left(1 - \cos\left(\frac{j\pi}{K}\right)\right)}{\eta^2}, \ 1 \le \lambda \le \kappa.$$

Since $\lambda$ is a monotone function of $\eta$, it suffices to check the endpoints of the interval $[\frac{\pi^2}{\kappa K^2}, 1]$. For $\eta^2 = 1$, we choose $j = K - 1$, which using $\frac{2x^2}{\pi^2} \le 1 - \cos(x) \le \frac{x^2}{2}$ for all $-\pi \le x \le \pi$, yields

$$1 \le \frac{4(K-1)^2}{K^2} \le \lambda \le \frac{(K-1)^2\pi^2}{K^2} \le \pi^2 \le \kappa.$$

Similarly, for $\eta^2 = \frac{\pi^2}{\kappa K^2}$, we choose $j = 1$, which yields

$$1 \le \frac{4}{\eta^2 K^2} \le \lambda \le \frac{\pi^2}{\eta^2 K^2} \le \kappa.$$

Now, consider the quadratic $f(x) = \frac{1}{2}x^\top \mathbf{A}x$ where $\mathbf{A} \in \mathbb{R}^{d \times d}$ is a diagonal matrix, $\mathbf{A}_{11} = 1$, $\mathbf{A}_{ii} = \kappa$ for all $3 \le i \le d$, and $\mathbf{A}_{22} = \lambda := \frac{2\left(1 - \cos\left(\frac{j\pi}{K}\right)\right)}{\eta^2}$ for the choice of $j$ which makes $1 \le \lambda \le \kappa$. For any symmetric starting set capturing a constant amount of measure along the second coordinate, by Lemma 13 and the following exposition, $x_K = \pm x_0$ along the second coordinate regardless of the random choice of velocity and thus the chain cannot leave the starting set. $\square$

## 6.2 HMC lower bound for all $K$

We now give our HMC lower bound, via improving Proposition 4 by a dimension dependence. We begin in Section 6.2.1, where we give a stronger upper bound on $\eta$ in the range $\eta^2 \le \frac{1}{\kappa K^2}$. Noting that there is a constant-sized gap between this range and the bound in Proposition 4, we rule out this gap in Section 6.2.2. Finally, we handle the case of extremely large $\eta^2 \ge 1$ in Section 6.2.3. We put these pieces together to prove Theorem 4 in Section 6.2.4.

### 6.2.1 Upper bounding $\eta = O(K^{-1}\kappa^{-\frac{1}{2}})$ under a constant gap

For this section, we let $\mathbf{A}$ be the $d \times d$ diagonal matrix which induces the hard quadratic function $f_{\mathrm{hq}}$, defined in (7) and reproduced here for convenience:

$$f_{\mathrm{hq}}(x) := \sum_{i \in [d]} f_i(x_i), \text{ where } f_i(c) = \begin{cases} \frac{1}{2}c^2 & i = 1 \\ \frac{\kappa}{2}c^2 & 2 \leq i \leq d \end{cases}.$$

We also let $h := \frac{\eta^2}{2}$, $x := x_0$, $g := v_0$, and $y := x_K$ throughout for analogy to Section 3, so that we can apply Proposition 1. Next, note that by the closed-form expression given by Lemma 13, we can write the iterates of the HMC chain in the form (8), reproduced here:

$$y = \begin{pmatrix} y_1 \\ y_{-1} \end{pmatrix}, \text{ where } y_1 = (1 - \alpha_1)x_1 + \beta_1 g_1$$

$$\text{and } y_{-1} = (1 - \alpha_{-1})x_{-1} + \beta_{-1}g_{-1}, \text{ for } g \sim \mathcal{N}(0, \mathbf{I}).$$

Concretely, we have by Lemma 13 that

$$\begin{aligned} \alpha_1 &= -\sum_{1 \leq j \leq K} (-1)^j (2h)^j \left(\frac{K}{K+j}\right)\binom{K+j}{2j}, \\ \alpha_{-1} &= -\sum_{1 \leq j \leq K} (-1)^j (2h\kappa)^j \left(\frac{K}{K+j}\right)\binom{K+j}{2j}, \\ \beta_1 &= \sqrt{2h} \sum_{0 \leq j \leq K-1} (-1)^j (2h)^j \binom{K+j}{2j+1}, \\ \beta_{-1} &= \sqrt{2h} \sum_{0 \leq j \leq K-1} (-1)^j (2h\kappa)^j \binom{K+j}{2j+1}. \end{aligned} \tag{25}$$

By a straightforward computation, the parameters in (25) satisfy the conditions of Proposition 1.

**Lemma 15.** *Supposing $\eta^2 \leq \frac{1}{\kappa K^2}$, $\alpha_1$, $\alpha_{-1}$, $\beta_1$, $\beta_{-1}$ defined in (25) satisfy*

$$|\alpha_{-1}| \leq \frac{3}{5}\beta_{-1}^2\kappa, \ |\alpha_1| = O(|\alpha_{-1}|), \ \beta_1 = O(\beta_{-1}).$$

*Proof.* The proof follows since under $\eta^2 \leq \frac{1}{10\kappa K^2}$, all of the parameters in (25) are dominated by their first summand. We will argue this for $\alpha_{-1}$ and $\beta_{-1}$; the corresponding conclusions for $\alpha_1$ and $\beta_1$ follow analogously since $\kappa \geq 1$. Define the summands of $\alpha_{-1}$ and $\beta_{-1}$ by

$$c_j := (-1)^{j+1}(2h\kappa)^j \left(\frac{K}{K+j}\right)\binom{K+j}{2j}, \ 1 \leq j \leq K,$$

$$d_j := \sqrt{2h}(-1)^j(2h\kappa)^j \binom{K+j}{2j+1}, \ 0 \leq j \leq K-1.$$

Then, we compute that for all $1 \leq j \leq K-1$, assuming $2h\kappa K^2 \leq 1$,

$$0 \geq \frac{c_{j+1}}{c_j} = (-2h\kappa)\frac{(K+j)(K-j)}{(2j+2)(2j+1)} \geq -\frac{2h\kappa K^2}{12} \geq -0.1. \tag{26}$$

Similarly, for all $0 \le j \le K - 2$,

$$0 \ge \frac{d_{j+1}}{d_j} = (-2h\kappa) \frac{(K+j+1)(K-j-1)}{(2j+3)(2j+2)} \ge -\frac{2h\kappa K^2}{6} \ge -0.2. \tag{27}$$

By repeating these calculations for $\alpha_1$ and $\beta_1$, we see that all parameters are given by rapidly decaying geometric sequences, and thus the conclusion follows by examination from

$$\alpha_1 \in \left[0.8hK^2, hK^2\right], \ \alpha_{-1} \in \left[0.8h\kappa K^2, h\kappa K^2\right],$$

$$\beta_1 \in \left[0.8\sqrt{2h}K, \sqrt{2h}K\right], \ \beta_{-1} \in \left[0.8\sqrt{2h}K, \sqrt{2h}K\right].$$

$\square$

We obtain the following corollary by combining Lemma 15, Corollary 4, and Proposition 1.

**Corollary 5.** *Let $x \in \mathbb{R}^d$ satisfy $\|x_{-1}\|_2 \le \sqrt{\frac{2d}{3\kappa}}$ and $|x_1| \le 5\sqrt{\log d}$, let $(x_K, v_K)$ be the result of the $K$-step HMC Markov chain with step size $\eta = \sqrt{2h}$ with $\eta^2 \le \frac{1}{\kappa K^2}$ from $x_0 = x$, and let $\mathbf{A}$ be as in (7). Then with probability at least $1 - d^{-5}$ over the randomness of $v_0 \sim \mathcal{N}(0, \mathbf{I})$, we have*

$$\mathcal{H}(x_0, v_0) - \mathcal{H}(x_K, v_K) = -\Omega \left(h^2 \kappa^2 K^2 d\right).$$

*Proof.* It suffices to use the bounds on $\beta_{-1} = \Theta(\sqrt{h}K)$ shown in the proof of Lemma 15 and the conclusions of Corollary 4 and Proposition 1. $\square$

### 6.2.2 Removing the constant gap

We show how to improve the bound in Corollary 5 to only require $\eta^2 \le \frac{\pi^2}{\kappa K^2}$, which removes the constant gap between the requirement of Corollary 5 and the bound in Proposition 4. First, let $\mathbf{A}_c$ be the $D \times d$ diagonal matrix which induces the following hard quadratic function $f_{\mathrm{hqc}}$:

$$f_{\mathrm{hqc}}(x) := \sum_{i \in [d]} f_i(x_i), \text{ where } f_i(c) = \begin{cases} \frac{1}{2}c^2 & i = 1 \\ \frac{\kappa}{2\pi^2}c^2 & 2 \le i \le d - 1 \\ \frac{\kappa}{2}c^2 & i = d \end{cases}. \tag{28}$$

In other words, along the first $d - 1$ coordinates, $f_{\mathrm{hqc}}$ is the same as a $d - 1$-dimensional variant of $f_{\mathrm{hq}}$ with condition number $\frac{\kappa}{\pi^2}$. We define a coordinate partition of $x$ and $g$ into $x_1$, $x_{-1d}$, $x_d$, and $g_1$, $g_{-1d}$, $g_d$, and we define $\alpha_1$, $\alpha_{-1d}$, $\alpha_d$, $\beta_1$, $\beta_{-1d}$, $\beta_d$ in analogy with (8).

We first note that because of separability of $f_{\mathrm{hqc}}$, and since the assumption of Corollary 5 holds on the first $d - 1$ coordinates for $\eta^2 \le \frac{\pi^2}{\kappa K^2}$, we can immediately obtain a bound on the change in the Hamiltonian along these coordinates.

**Corollary 6.** *Let $x \in \mathbb{R}^d$ satisfy $\|x_{-1}\|_2 \le \sqrt{\frac{2\pi^2 d}{3\kappa}}$ and $|x_1| \le 5\sqrt{\log d}$, let $(x_K, v_K)$ be the result of the $K$-step HMC Markov chain with step size $\eta = \sqrt{2h}$ where $\eta^2 \le \frac{\pi^2}{\kappa K^2}$ from $x_0 = x$, and let $\mathbf{A}_c$ be as in (28). Then with probability at least $1 - 2d^{-5}$ over the randomness of $v_0 \sim \mathcal{N}(0, \mathbf{I})$, we have*

$$\mathcal{H}\left([x_0]_{[d-1]}, [v_0]_{[d-1]}\right) - \mathcal{H}\left([x_K]_{[d-1]}, [v_K]_{[d-1]}\right) = -\Omega \left(h^2 \kappa^2 K^2 d\right).$$

We now move to bounding the contribution of the last coordinate.

**Lemma 16.** *Let $(y, v_K)$ be the result of the $K$-step HMC Markov chain with step size $\eta = \sqrt{2h}$ where $\eta^2 \leq \frac{\pi^2}{\kappa K^2}$, and write $y_d = (1 - \alpha_d) x_d + \beta_d g_d$, for*

$$\alpha_d = -\sum_{1 \leq j \leq K} (-1)^j (2h\kappa)^j \left(\frac{K}{K+j}\right) \binom{K+j}{2j}, \quad \beta_d = \sqrt{2h} \sum_{0 \leq j \leq K-1} (-1)^j (2h\kappa)^j \binom{K+j}{2j+1}.$$

*Then, we have $|\alpha_d| = O(h\kappa K^2)$, $|\beta_d| = O(\sqrt{h}K)$.*

*Proof.* After the index $j$ is a sufficiently large constant, the geometric argument sequence of Lemma 15 applies (since the denominators of the ratios (26) and (27) grow with the index $j$); before then, each coefficient is within a constant factor of the first in absolute value. Thus, the coefficients can be at most a constant factor larger than the first in absolute value. $\square$

**Lemma 17.** *Let $|[x_0]_d| \leq \frac{\log d}{\sqrt{\kappa}}$, $|[v_0]_d| \leq \log d$, and let $(x_K, v_K)$ be the result of the $K$-step HMC Markov chain with step size $\eta = \sqrt{2h}$ where $\eta^2 \leq \frac{\pi^2}{\kappa K^2}$. Then with probability at least $1 - d^{-5}$ over the randomness of $v_0 \sim \mathcal{N}(0, \mathbf{I})$, we have*

$$\mathcal{H}\left([x_0]_d, [v_0]_d\right) - \mathcal{H}\left([x_K]_d, [v_K]_d\right) = o\left(h^2 \kappa^2 K^2 d\right).$$

*Proof.* We can assume $|[v_0]_d| = |g_d| \leq \log d$, which passes the high probability bound. By Corollary 4 and Lemma 2, we wish to bound

$$\frac{h\kappa^2}{4} \left(\left(2\alpha_d - \alpha_d^2\right) x_d^2 - \beta_d^2 g_d^2 - 2(1 - \alpha_d)\beta_d x_d g_d\right) = o\left(h^2 \kappa^2 K^2 d\right).$$

Dropping all clearly negative terms, and since $|\alpha_d| = O(1)$ by Lemma 16, it is enough to show

$$\left|h\kappa^2 \alpha_d x_d^2\right| = o\left(h^2 \kappa^2 K^2 d\right), \quad \left|h\kappa^2 \beta_d x_d g_d\right| = o\left(h^2 \kappa^2 K^2 d\right).$$

The first bound is immediate from assumptions. The second follows from assumptions as well since $\sqrt{h\kappa K^2}$ is at most a constant, so $\left|h\kappa^2 \beta_d x_d g_d\right| = O(h^{1.5} \kappa^{1.5} K \log^2 d) = O(h^2 \kappa^2 K^2 \log^2 d)$. $\square$

By combining Lemma 17 and Corollary 6, we obtain the following strengthening of Corollary 5.

**Corollary 7.** *Let $x \in \mathbb{R}^d$ satisfy $\|x_{-1d}\|_2 \leq \sqrt{\frac{2d}{3\kappa}}$, $|x_1| \leq 5\sqrt{\log d}$, and $|x_d| \leq \frac{\log d}{\sqrt{\kappa}}$, let $(x_K, v_K)$ be the result of the $K$-step HMC Markov chain with step size $\eta = \sqrt{2h}$ with $\eta^2 \leq \frac{\pi^2}{\kappa K^2}$ from $x_0 = x$, and let $\mathbf{A}_c$ be as in (28). Then with probability at least $1 - d^{-5}$ over the randomness of $v_0 \sim \mathcal{N}(0, \mathbf{I})$, we have*

$$\mathcal{H}(x_0, v_0) - \mathcal{H}(x_K, v_K) = -\Omega\left(h^2 \kappa^2 K^2 d\right).$$

### 6.2.3 Ruling out $\eta \geq 1$

Finally, we give a short argument ruling out the case $\eta \geq 1$ not covered by Proposition 4. In this section, let $\pi^* = \mathcal{N}(0, \kappa^{-1}\mathbf{I})$, with negative log-density $f(x) = \frac{\kappa}{2} \|x\|_2^2$. For $\eta \geq 1$ and $\kappa \geq 10$, (24) and straightforward lower bounds on Chebyshev polynomials outside the range $[-1, 1]$ demonstrate the proposal distribution is of the form (from starting point $x_0 \in \mathbb{R}^d$)

$$x_K \leftarrow \alpha x_0 + \beta v_0, \ v_0 \sim \mathcal{N}(0, 1), \ |\alpha| \geq 10, \ |\beta| \geq 1. \tag{29}$$

**Lemma 18.** *Letting $(x_K, v_K)$ be the result of $K$-step HMC from any $x_0$, and $f(x) = \frac{\kappa}{2} \|x\|_2^2$, for $\eta \geq 1$, with probability at least $1 - d^{-5}$ over the randomness of $v_0 \sim \mathcal{N}(0, \mathbf{I})$, we have*

$$\mathcal{H}(x_0, v_0) - \mathcal{H}(x_K, v_K) = -\Omega(d).$$

*Proof.* Following notation (29) and applying Corollary 4, it suffices to show

$$\|x_0\|_2^2 - \|\alpha x_0 + \beta v_0\|_2^2 = -\Omega(d).$$

Expanding, it suffices to upper bound

$$\left(1 - \alpha^2\right) \|x_0\|_2^2 - 2\alpha\beta \langle x_0, v_0 \rangle - \beta^2 \|v_0\|_2^2.$$

With probability at least $1 - d^{-5}$, Fact 2 shows $\|v_0\|_2^2 \geq \frac{1}{2}d$ and $\langle x_0, v_0 \rangle \geq -4\sqrt{\log d} \|x_0\|_2$. Hence,

$$\left(1 - \alpha^2\right) \|x_0\|_2^2 - 2\alpha\beta \langle x_0, v_0 \rangle - \beta^2 \|v_0\|_2^2 \leq -0.99\alpha^2 \|x_0\|_2^2 + 8\alpha\beta\sqrt{\log d} \|x_0\|_2 - \frac{\beta^2}{2}d$$

$$\leq 20\beta^2 \log d - \frac{\beta^2}{2}d = -\Omega(d).$$

Here, we used that $\alpha^2 \geq 100$ and took $d$ larger than a sufficiently large constant. $\qquad\square$

### 6.2.4 Proof of Theorem 4

A consequence of Corollary 5 is that if the step size $h = \omega(\frac{\sqrt{\log d}}{\kappa K \sqrt{d}})$, initializing the chain from any $x_0$ in the set $\Omega$ defined in (12) leads to a polynomially bad mixing time. We further relate the step size to the spectral gap of the HMC Markov chain in the following.

**Lemma 19.** *The spectral gap of the $K$-step HMC Markov chain for sampling from the density proportional to $\exp(-f_{\mathrm{hq}})$, where $f_{\mathrm{hq}}$ is defined in (7), is $O(hK^2 + h^2K^4)$.*

*Proof.* We follow the proof of Lemma 1; again let $g(x) = x_1$, and $\pi^*$ be the stationary distribution. For our function $f$, it is clear again that $\mathrm{Var}_{\pi^*}[g] = \Theta(1)$. Thus it suffices to upper bound $\mathcal{E}(g, g)$: letting $\mathcal{P}_x(y)$ be the proposal distribution of $K$-step HMC, and $\alpha_1$, $\beta_1$ be as in (25),

$$\mathcal{E}(g, g) \leq \frac{1}{2} \iint (x_1 - y_1)^2 \mathcal{P}_x(y) d\pi^*(x) dy$$

$$\leq \mathbb{E}_{x \sim \pi^*} \left[\alpha_1^2 x_1^2\right] + \mathbb{E}_{\xi \sim \mathcal{N}(0,1)} \left[\beta_1^2 \xi^2\right]$$

$$= \alpha_1^2 + \beta_1^2 = O\left(hK^2 + h^2K^4\right).$$

$\qquad\square$

Finally, by combining Lemma 19 and Corollary 7, we arrive at the main result of this section.

**Theorem 4.** *For every step size and count, there is a target Gaussian on $\mathbb{R}^d$ whose negative log-density always has Hessian eigenvalues in $[1, \kappa]$, such that the relaxation time of HMC is $\Omega(\frac{\kappa\sqrt{d}}{K\sqrt{\log d}})$.*

*Proof.* For $1 \geq \eta^2 \geq \frac{\pi^2}{\kappa K^2}$ it suffices to apply Proposition 4. For $\eta^2 \geq 1$, we apply Lemma 18. Otherwise, in the relevant range of $h = 2\eta^2$, the dominant term in Lemma 19 is $O(hK^2)$. Applying

Corollary 7 with the hard quadratic function $f_{\text{hqc}}$, the remainder of the proof follows analogously to that of Theorem 1. □

We remark that as in Theorem 1, it is straightforward to see that the measure of the bad region $\|x_{-1d}\|_2 \leq \sqrt{\frac{2d}{3\kappa}}$, $|x_1| \leq 5\sqrt{\log d}$, and $|x_d| \leq \frac{\log d}{\sqrt{\kappa}}$ used in the proof is at least $\exp(-d)$.

## 7 Conclusion

In this work, we presented relaxation time lower bounds for the MALA and HMC Markov chains at every step size and scale, as well as a mixing time bound for MALA from an exponentially warm start. We highlight in this section a number of unexplored directions left open by our work, beyond direct strengthenings of our results, which we find interesting and defer to a future exploration.

**Variable or random step sizes.** Our lower bounds were for MALA and HMC Markov chains with a *fixed step size*. For variable step sizes which take e.g. values in a bounded multiplicative range, we believe our arguments can be modified to give relaxation time lower bounds for the resulting Markov chains. However, the arguments of Section 6 (our HMC lower bound) are particularly brittle to large multiplicative ranges of candidate step sizes, because they rely on the locations of Chebyshev polynomial zeroes, which only occur in a bounded range. From an algorithm design perspective, this suggests that adaptively or randomly choosing step size ranges may be effective in improving the performance of HMC. Such a result would also give theoretical justification to the No-U-Turn sampler of [HG14], a common HMC alternative in practice. We state as an explicit open problem: can one obtain improved upper bounds, such as a $\sqrt{\kappa}$ dependence or a dimension-independent rate, for example by using variations of these strategies (variable step sizes)?

**Necessity of $\kappa$ lower bound.** All of our witness sets throughout the paper are $\exp(-d)$ sized. It was observed in [DCWY18] that it is possible to construct a starting distribution with warmness arbitrarily close to $\sqrt{\kappa}^d$; the marginal restriction of our witness set falls under this warmness bound for all $\kappa \geq e^2 \approx 8$. However, recently [LST20b] proposed a *proximal point reduction* approach to sampling, which (for mixing bounds scaling at least linearly in $\kappa$) shows that it suffices to sample a small number of regularized distributions, whose condition numbers are arbitrarily close to 1.

By adjusting constants, we can modify the proof of the Gaussian lower bounds (Theorems 1 and 4) to have witness sets with measure $c^d$ for a constant $c$ arbitrarily close to 1 (the bottleneck being Lemma 4). However, our witness set for the family of hard distributions in Section 4 encounters a natural barrier at measure $2^d$, since the set is sign-restricted by the cosine function (and hence can only contain roughly every other period). This bottleneck is encountered in the proof of Lemma 9. We find it interesting to see if a stronger construction rules out the existing warm starts for all $\kappa \geq 1$, or if an upper bound can take advantage of the reduction of [LST20b] to obtain improved dependences on dimension assuming $\kappa \approx 1$.

### Acknowledgments

We would like to thank Santosh Vempala for numerous helpful conversations, pointers to the literature, and writing suggestions throughout the course of this project.

YL and RS are supported by NSF awards CCF-1749609, DMS-1839116, and DMS-2023166, a Microsoft Research Faculty Fellowship, a Sloan Research Fellowship, and a Packard Fellowship. KT is supported by NSF Grant CCF-1955039 and the Alfred P. Sloan Foundation.

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

## A   Necessity of fixing a scale

We give a simple argument showing if the step size $\eta$ of the HMC algorithm does not depend on the "scale" of the problem, namely the eigenvalues of the function Hessian (as opposed to scale-invariant quantities, e.g. the condition number $\kappa$ and the dimension), then the task of proving lower bounds becomes much more trivial. In particular, we can adaptively pick a scale of the problem in response to the fixed $\eta$. This justifies the additional requirement in Theorems 1, 2, 3 and 4 of the fixed scale $[1, \kappa]$, which we remark is a *strengthening* of an analogous scale-free lower bound.

Concretely, suppose we wished to prove the statement of Theorem 4 but only on functions with condition number $\kappa$ (without specifying a range of eigenvalues). Then, for fixed $\eta$, $K$, consider

$$f(x) = \frac{\lambda}{2} x^2, \text{ where } \lambda := \frac{2\left(1 - \cos\left(\frac{\pi}{K}\right)\right)}{\eta^2}.$$

Clearly, $f : \mathbb{R} \to \mathbb{R}$ has condition number $1 \le \kappa$ for any $\kappa$. Then, the proof of Proposition 4 applies to show that the HMC Markov chain cannot leave any symmetric set, because the coefficients encounter extremal points or zeroes of the Chebyshev polynomials.

## B   HMC lower bounds beyond $\kappa\sqrt{d}$

Here, we analyze the behavior of HMC on the hard function (20). We will use this construction to demonstrate that when the number of steps $K$ is small, we cannot improve either the relaxation time (Section B.1) or the mixing time (Section B.2) of MALA by more than roughly a $O(K)$ factor.

### B.1   Relaxation time lower bound for small $K$

We first give a bound on the acceptance probability (6) for general HMC Markov chain. We expand the term $-\mathcal{H}(x_K, v_K) + \mathcal{H}(x_0, v_0)$ and extend the result given by Lemma 12.

**Lemma 20.** *For the iterates given by Fact 4, write $\tilde{x}_j := x_0 + \eta j v_0$ for $0 \le j \le K - 1$. Then, for a $\kappa$-smooth function $f$,*

$$-\mathcal{H}(x_K, v_K) + \mathcal{H}(x_0, v_0) \le \sum_{j=0}^{K-1} \left( -f(\tilde{x}_{j+1}) + f(\tilde{x}_j) + \frac{1}{2} \langle \eta v_0, \nabla f(\tilde{x}_{j+1}) + \nabla f(\tilde{x}_j) \rangle \right)$$

$$+ \eta K \|v_0\|_2 \max_{0 \le j \le K} \|\nabla f(x_j) - \nabla f(\tilde{x}_j)\|_2 + \frac{1}{2} \eta^2 K^2 \max_{0 \le j_1, j_2 \le K} \|\nabla f(\tilde{x}_K) - \nabla f(x_{j_2})\|_2 \|\nabla f(x_{j_1})\|_2$$

$$+ \frac{1}{2} \eta^2 K^2 \max_{0 \le j_1, j_2, j_3 \le K} \|\nabla f(x_{j_3})\|_2 \|\nabla f(x_{j_1}) - \nabla f(x_{j_2})\|_2.$$

*Proof.* Expanding $\mathcal{H}(x_0, v_0) - \mathcal{H}(x_K, v_K)$ according to the definition of $\mathcal{H}$, $x_K$ and $v_K$,

$$\mathcal{H}(x_0, v_0) - \mathcal{H}(x_K, v_K)$$

$$= -f(x_K) + f(x_0) - \frac{\left\| v_0 - \frac{\eta}{2}\nabla f(x_0) - \eta \sum_{j=1}^{K-1} \nabla f(x_j) - \frac{\eta}{2}\nabla f(x_K) \right\|_2^2}{2} + \frac{\|v_0\|_2^2}{2}$$

$$= -f(x_K) + f(\tilde{x}_K) - f(\tilde{x}_K) + f(x_0) + \left\langle v_0, \frac{\eta}{2}\nabla f(x_0) + \eta \sum_{j=1}^{K-1} \nabla f(x_j) + \frac{\eta}{2}\nabla f(x_K) \right\rangle$$

$$- \frac{1}{2}\left\| \frac{\eta}{2}\nabla f(x_0) + \eta \sum_{j=1}^{K-1} \nabla f(x_j) + \frac{\eta}{2}\nabla f(x_K) \right\|_2^2$$

$$= -f(x_K) + f(\tilde{x}_K) + \sum_{j=0}^{K-1} (-f(\tilde{x}_{j+1}) + f(\tilde{x}_j)) + \left\langle \eta v_0, \frac{1}{2}\nabla f(\tilde{x}_0) + \sum_{j=1}^{K-1} \nabla f(\tilde{x}_j) + \frac{1}{2}\nabla f(\tilde{x}_K) \right\rangle$$

$$- \frac{1}{2}\left\| \frac{\eta}{2}\nabla f(x_0) + \eta \sum_{j=1}^{K-1} \nabla f(x_j) + \frac{\eta}{2}\nabla f(x_K) \right\|_2^2$$

$$+ \left\langle \eta v_0, \left( \frac{1}{2}\nabla f(x_0) + \sum_{j=1}^{K-1} \nabla f(x_j) + \frac{1}{2}\nabla f(x_K) \right) - \left( \frac{1}{2}\nabla f(\tilde{x}_0) + \sum_{j=1}^{K-1} \nabla f(\tilde{x}_j) + \frac{1}{2}\nabla f(\tilde{x}_K) \right) \right\rangle$$

$$= \sum_{j=0}^{K-1} \left( -f(\tilde{x}_{j+1}) + f(\tilde{x}_j) + \frac{1}{2}\langle \eta v_0, \nabla f(\tilde{x}_{j+1}) + \nabla f(\tilde{x}_j)\rangle \right)$$

$$- f(x_K) + f(\tilde{x}_K) - \frac{1}{2}\left\| \frac{\eta}{2}\nabla f(x_0) + \eta \sum_{j=1}^{K-1} \nabla f(x_j) + \frac{\eta}{2}\nabla f(x_K) \right\|_2^2$$

$$+ \left\langle \eta v_0, \left( \frac{1}{2}\nabla f(x_0) + \sum_{j=1}^{K-1} \nabla f(x_j) + \frac{1}{2}\nabla f(x_K) \right) - \left( \frac{1}{2}\nabla f(\tilde{x}_0) + \sum_{j=1}^{K-1} \nabla f(\tilde{x}_j) + \frac{1}{2}\nabla f(\tilde{x}_K) \right) \right\rangle.$$

$$(30)$$

Now we bound the last two lines in the decomposition (30). For the second-to-last line of (30), by convexity of $f$ and the Cauchy-Schwarz inequality,

$$- f(x_K) + f(\tilde{x}_K) - \frac{1}{2}\left\| \frac{\eta}{2}\nabla f(x_0) + \eta \sum_{j=1}^{K-1} \nabla f(x_j) - \frac{\eta}{2}\nabla f(x_K) \right\|_2^2$$

$$\leq \left\langle \nabla f(\tilde{x}_K), \frac{1}{2}K\eta^2 \nabla f(x_0) + \eta^2 \sum_{j=1}^{K-1} (K-j)\nabla f(x_j) \right\rangle - \frac{1}{2}\left\| \frac{\eta}{2}\nabla f(x_0) + \eta \sum_{j=1}^{K-1} \nabla f(x_j) + \frac{\eta}{2}\nabla f(x_K) \right\|_2^2$$

$$\leq \frac{1}{2}\eta^2 K^2 \max_{0 \leq j_1, j_2, j_3 \leq K} \left( \nabla f(\tilde{x}_K)^\top \nabla f(x_{j_1}) - \nabla f(x_{j_2})^\top \nabla f(x_{j_3}) \right)$$

$$\leq \frac{1}{2}\eta^2 K^2 \left( \max_{0 \leq j_1, j_2 \leq K} \|\nabla f(\tilde{x}_K) - \nabla f(x_{j_2})\|_2 \|\nabla f(x_{j_1})\|_2 + \max_{0 \leq j_1, j_2, j_3 \leq K} \|\nabla f(x_{j_3})\|_2 \|\nabla f(x_{j_1}) - \nabla f(x_{j_2})\|_2 \right).$$

$$(31)$$

In the third line above, we used that the total "number of gradient inner products" for both terms

is $\frac{1}{2}\eta^2 K^2$, and took the largest such inner product difference.

Finally, for the last line of (30), by the Cauchy-Schwarz inequality,

$$\left\langle \eta v_0, \left( \frac{1}{2}\nabla f(x_0) + \sum_{j=1}^{K-1} \nabla f(x_j) + \frac{1}{2}\nabla f(x_K) \right) - \left( \frac{1}{2}\nabla f(\tilde{x}_0) + \sum_{j=1}^{K-1} \nabla f(\tilde{x}_j) + \frac{1}{2}\nabla f(\tilde{x}_K) \right) \right\rangle$$
$$\leq \eta K \left\| v_0 \right\|_2 \max_{0 \leq j \leq K} \left\| \nabla f(x_j) - \nabla f(\tilde{x}_j) \right\|_2 .$$

(32)

Combining (30), (31) and (32) proves the desired claim. $\qquad\square$

We define a hard function $f_{\text{hard}} : \mathbb{R}^d \to \mathbb{R}$ that is $\kappa$-smooth and 1-strongly convex (note it is the same hard function as in Section 6, under the change of variable $h = \frac{\eta^2}{2}$). We will show it is hard to sample from the density proportional to $\exp(-f_{\text{hard}})$ when $K$ is small.

$$f_{\text{hard}}(x) := \sum_{i \in [d]} f_i(x_i), \text{ where } f_i(c) = \begin{cases} \frac{1}{2}c^2 & i = 1 \\ \frac{\kappa}{3}c^2 - \frac{\kappa\eta^2}{6}\cos\left(\frac{\sqrt{2}c}{\eta}\right) & 2 \leq i \leq d \end{cases} . \tag{33}$$

**Lemma 21.** *For $\eta^2 \leq 1$, let $\tilde{x}_j := x_0 + \eta j v_0$ for $0 \leq j \leq K - 1$ and $v_0 \sim \mathcal{N}(0, \mathbf{I})$. Let $R^{(j)}$ be the random variable with given by $R^{(j)} = \sum_{i=1}^d R_i^{(j)}$ where*

$$R_i^{(j)} = -f_i([\tilde{x}_{j+1}]_i) + f_i([\tilde{x}_j]_i) + \frac{1}{2}\eta[v_0]_i \cdot (\nabla f_i([\tilde{x}_{j+1}]_i) + \nabla f_i([\tilde{x}_j]_i)).$$

*Then,*

$$\mathbb{E}_{v_0 \sim \mathcal{N}(0,1)}\left[ \sum_{j=0}^{K-1} R^{(j)} \right] \leq -0.02\kappa\eta^2 \sum_{i=2}^d \cos\frac{\sqrt{2}[x_0]_i}{\eta} . \tag{34}$$

*and*

$$\Pr\left[ \sum_{j=0}^{K-1} R^{(j)} - \mathbb{E}\left[ \sum_{j=0}^{K-1} R^{(j)} \right] \geq 10\eta^2 K \kappa \sqrt{d \log d} \right] \leq \frac{1}{d^5} . \tag{35}$$

*Proof.* In this proof, all expectations $\mathbb{E}$ are taken over $v_0 \sim \mathcal{N}(0, \mathbf{I})$, so we omit them. For $i = 1$,

$$\mathbb{E}\left[ \sum_{j=0}^{K-1} R_i^{(j)} \right] = \mathbb{E}\left[ -\frac{1}{2}([x_0]_1 + \eta K[v_0]_1)^2 + \frac{1}{2}[x_0]_1^2 + \frac{1}{2}\sum_{j=0}^{K-1} \eta[v_0]_1(2[x_0]_1 + \eta(2j+1)[v_0]_1) \right]$$
$$= \mathbb{E}\left[ -\frac{1}{2}[x_0]_1^2 - \frac{1}{2}\eta^2 K^2[v_0]_1^2 - \eta K[x_0]_1[v_0]_1 + \frac{1}{2}[x_0]_1^2 + \frac{1}{2}\eta^2 K^2[v_0]_1^2 + \eta K[x_0]_1[v_0]_1 \right] = 0.$$

We bound each coordinate $2 \le i \le d$ separately.

$$\mathbb{E}\left[\sum_{j=0}^{K-1} R_i^{(j)}\right]$$

$$=\mathbb{E}\left[\sum_{j=0}^{K-1} -f_i([\tilde{x}_{j+1}]_i) + f_i([\tilde{x}_j]_i) + \frac{1}{2}\eta[v_0]_i \cdot (\nabla f_i([\tilde{x}_{j+1}]_i) + \nabla f_i([\tilde{x}_j]_i))\right]$$

$$=-\frac{\kappa}{3}\mathbb{E}\left[([x_0]_i + \eta K[v_0]_i)^2 - [x_0]_i^2\right] + \frac{1}{3}\eta\kappa\mathbb{E}\left[[v_0]_i \cdot \left(2[x_0]_i + \eta\sum_{j=0}^{K-1}(2j+1)[v_0]_i\right)\right]$$

$$+\frac{\kappa\eta^2}{6}\mathbb{E}\left[\sum_{j=0}^{K-1} \cos\frac{\sqrt{2}\,([x_0]_i + \eta(j+1)[v_0]_i)}{\eta} - \cos\frac{\sqrt{2}\,([x_0]_i + \eta j[v_0]_i)}{\eta}\right]$$

$$+\frac{\sqrt{2}\eta^2\kappa}{12}\mathbb{E}\left[[v_0]_i \sum_{j=0}^{K-1}\left(\sin\frac{\sqrt{2}\,([x_0]_i + \eta j[v_0]_i)}{\eta} + \sin\frac{\sqrt{2}\,([x_0]_i + \eta(j+1)[v_0]_i)}{\eta}\right)\right]$$

$$=-\frac{\kappa\eta^2}{6}\sum_{j=0}^{K-1}\exp(-j^2) - \exp(-(j+1)^2) - j\exp(-j^2) - (j+1)\exp(-(j+1)^2)\cos\frac{\sqrt{2}[x_0]_i}{\eta}$$

The last line used the computation

$$\mathbb{E}\left[[v_0]_i \sin\frac{\sqrt{2}\,([x_0]_i + \eta j[v_0]_i)}{\eta}\right] = \sqrt{2}j\exp(-j^2)\cos\frac{\sqrt{2}[x_0]_i}{\eta},$$

$$\mathbb{E}\left[\cos\frac{\sqrt{2}\,([x_0]_i + \eta j[v_0]_i)}{\eta}\right] = \exp(-j^2)\cos\frac{\sqrt{2}[x_0]_i}{\eta}.$$

Next, we bound $\sum_{j=0}^{K-1}\left(\exp(-j^2) - \exp(-(j+1)^2) - j\exp(-j^2) - (j+1)\exp(-(j+1)^2)\right)$. For $j = 0$, $1 - \frac{2}{\exp(1)} \ge 0.264$. For $j = 1$, the negative terms have $-3\exp(-4) \ge -0.06$, and the positive terms can only help this inequality. For the remaining terms,

$$\sum_{j=2}^{K-1}\left(\exp(-j^2) - \exp(-(j+1)^2) - j\exp(-j^2) - (j+1)\exp(-(j+1)^2)\right)$$

$$\ge \sum_{j=2}^{K-1}\left(-j\exp(-j^2) - (j+1)\exp(-(j+1)^2)\right)$$

$$\ge -2\sum_{j=2}^{K}\left(j\exp(-j^2)\right) \ge -2\frac{2}{\exp(4)}\frac{1}{1 - 2\exp(-5)} \ge -0.075.$$

The last inequality used the ratio between two consecutive terms is bounded by $\frac{j+1}{j}\exp(j^2 - (j+1)^2) \le 2\exp(-5)$. Summing over $d$ coordinates proves (34).

Next, we prove the concentration property of $\sum_{j=0}^{K-1} R^{(j)}$. Let $\tilde{x}_{j,s} = \tilde{x}_j + s\eta v_0$, for $s \in [0,1]$ and

$j = 0, ..., K - 1$. By Lemma 6, we have

$$\sum_{j=0}^{K-1} R^{(j)} = \sum_{j=0}^{K-1} -\eta^2 \int_0^1 \left( \frac{1}{2} - s \right) v_0^\top \nabla^2 f(\tilde{x}_{j,s}) v_0 ds.$$

For coordinate $1 \le i \le d$, $\left| \eta^2 \int_0^1 \left( \frac{1}{2} - s \right) f_i''([x_{j,s}]_i) ds \right| \le \frac{\eta^2 \kappa}{2}$ by smoothness. Then, the random variables $\sum_{j=0}^{K-1} R_i^{(j)} - \mathbb{E}\left[ \sum_{j=0}^{K-1} R_i^{(j)} \right]$ for $1 \le i \le d$ are sub-exponential with parameter $\frac{\eta^2 \kappa K}{2}$ (for coordinates where the coefficient is negative, note the negation of a sub-exponential random variable is still sub-exponential). Hence, by Fact 3,

$$\Pr\left[ \sum_{i \in [d]} \left( \sum_{k=0}^{K-1} R_i^{(j)} - \mathbb{E}\left[ \sum_{k=0}^{K-1} R_i^{(j)} \right] \right) \ge 10\eta^2 K \kappa \sqrt{d \log d} \right] \le \frac{1}{d^5}.$$

$\square$

Now, we build a bad set $\Omega_{\text{hard}}$ with lower bounded measure that starting from a point $x_0 \in \Omega_{\text{hard}}$, such that with high probability, $-\mathbb{E}\left[ \sum_{j=0}^{K-1} R^{(j)} \right]$ is very negative. Let $h = \frac{1}{2}\eta^2$ so that we may use the results from Section 4. We use the bad set $\Omega_{\text{hard}}$ defined in (21).

$$\Omega_{\text{hard}} = \left\{ x \mid |x_1| \le 2, \forall 2 \le i \le d, \exists k_i \in \mathbb{Z}, |k_i| \le \left\lfloor \frac{5}{\pi\sqrt{h\kappa}} \right\rfloor, \text{ such that} \right.$$

$$\left. -\frac{9}{20}\pi\sqrt{h} + 2\pi k_i \sqrt{h} \le x_i \le \frac{9}{20}\pi\sqrt{h} + 2\pi k_i \sqrt{h} \right\}.$$

We restate Lemma 9 here, which lower bounds $\pi^*(\Omega_{\text{hard}})$ and bounds $\|\nabla f(x)\|_2$ for $x \in \Omega_{\text{hard}}$.

**Lemma 9.** *Let* $h \le \frac{1}{10000\pi^2\kappa}$. *Let* $\pi^*$ *have log-density* $-f_{\text{hard}}$ (20). *Then,* $\pi^*(\Omega_{\text{hard}}) \ge \exp(-d)$. *Moreover, for all* $x \in \Omega_{\text{hard}}$, $\|\nabla f(x)\|_2 \le 10\sqrt{\kappa d}$.

We can further show the following, which is used to bound the remaining terms in Lemma 20.

**Lemma 22.** *Let* $x_0 \in \Omega_{\text{hard}}$, $\eta K \le \frac{1}{100\sqrt{\kappa} \log d}$ *and* $d \ge 8$. *Let let* $x_j$ *for* $1 \le j \le K - 1$ *be given by the iterates in Fact 4 and* $\tilde{x}_K = x_0 + \eta K v_0$. *Then, with probability at least* $1 - \frac{1}{d^5}$ *over random* $v_0 \sim \mathcal{N}(0, \mathbf{I})$, $\|v_0\|_2 \le 4\sqrt{d} \log d$ *and for all* $0 \le j \le K$, $\|\nabla f(x_j)\|_2 \le 11\sqrt{\kappa d}$ *and* $\|\nabla f(\tilde{x}_K)\|_2 \le 11\sqrt{\kappa d}$.

*Proof.* We first derive a bound on $v_0 \sim \mathcal{N}(0, \mathbf{I})$. By a standard Gaussian tail bound, for $d \ge 8$, with probability at least $1 - \frac{1}{d^5}$, $|[v_0]_i| \le 4 \log d$ for all $1 \le i \le d$. Then, $\|v_0\|_2 \le \sqrt{16d(\log d)^2} = 4\sqrt{d} \log d$. Now, we prove the bound on $\|x_j - x_0\|_2$ and $\|\nabla f(x_j)\|_2$ using induction. First, $\|\nabla f(x_0)\| \le 11\sqrt{d\kappa}$ holds by Lemma 9. Assume for induction $\|\nabla f(x_k)\|_2 \le 11\sqrt{d\kappa}$ for $1 \le k < j$. Then,

$$\|x_j - x_0\|_2 \le \left\| \eta j v_0 - \frac{\eta^2 j}{2} \nabla f(x_0) - \eta^2 \sum_{k=1}^{j-1} (j-k) \nabla f(x_k) \right\|_2$$

$$\le 4\eta j \sqrt{d} \log d + \eta^2 j^2 \cdot 11\sqrt{\kappa d} \le \sqrt{\frac{d}{\kappa}}.$$

The last inequality used the assumption $\eta K \le \frac{1}{100\sqrt{\kappa}\log d}$. Since $f$ is $\kappa$-smooth, we have

$$\|\nabla f(x_j)\|_2 \le \|\nabla f(x_0)\|_2 + \kappa \|x_j - x_0\|_2 \le 10\sqrt{\kappa d} + \kappa\sqrt{\frac{d}{\kappa}} \le 11\sqrt{\kappa d}.$$

This completes the induction step. Finally, we have

$$\|\nabla f(\tilde{x}_K)\|_2 \le \|\nabla f(x_0)\|_2 + \kappa \|\eta K v_0\|_2 \le 10\sqrt{\kappa d} + 4\eta K \kappa \sqrt{d}\log d \le 11\sqrt{\kappa d},$$

where we used $\eta K \le \frac{1}{100\sqrt{\kappa}\log d}$. $\qquad\qquad\square$

**Lemma 23.** *Let $\eta$ and $K$ satisfy $K \le \frac{\sqrt{d}}{10000\sqrt{\log d}}$, and $\eta K^3 \le \frac{1}{100000\sqrt{\kappa}\log d}$. For any $x_0 \in \Omega_{\mathrm{hard}}$, let $(x_K, v_K)$ be given by the iterates in Fact 4 and $v_0 \sim \mathcal{N}(0, \mathbf{I})$. With probability at least $1 - \frac{2}{d^5}$,*

$$-\mathcal{H}(x_K, v_K) + \mathcal{H}(x_0, v_0) \le -\Omega\left(\eta^2 \kappa d\right).$$

*Proof.* We first remark that the bound on $\eta K^3$ implies we may apply Lemma 9 and Lemma 22. Next, for $x_0 \in \Omega_{\mathrm{hard}}$, $\cos\frac{\sqrt{2}[x_0]_i}{\eta}$ is bounded away from 0 for all $2 \le i \le d$. By Lemma 21, when $K \le \frac{\sqrt{d}}{10000\sqrt{\log d}}$, with probability at least $1 - \frac{1}{d^5}$, $\sum_{j=0}^{K-1} R^{(j)} \le -0.002\eta^2\kappa d$ (the expectation term dominates). By Lemma 22, with probability at least $1 - \frac{1}{d^5}$, the other terms in Lemma 20 have

$$\eta K \|v_0\|_2 \max_{0 \le j \le K} \|\nabla f(x_j) - \nabla f(\tilde{x}_j)\|_2 + \frac{1}{2}\eta^2 K^2 \max_{0 \le j_1, j_2 \le K} \|\nabla f(\tilde{x}_K) - \nabla f(x_{j_2})\|_2 \|\nabla f(x_{j_1})\|_2$$

$$+ \frac{1}{2}\eta^2 K^2 \max_{0 \le j_1, j_2, j_3 \le K} \|\nabla f(x_{j_3})\|_2 \|\nabla f(x_{j_1}) - \nabla f(x_{j_2})\|_2$$

$$\le 4\eta K \sqrt{d}\log d \cdot \kappa\eta^2 \left( K \|\nabla f(x_0)\|_2 + \sum_{j \in [K-1]} (K - j)\|\nabla f(x_j)\|_2 \right)$$

$$+ \eta^2 K^2 \cdot 11\sqrt{\kappa d} \cdot \kappa \left( \eta K \|v_0\|_2 + \eta^2 K \|\nabla f(x_0)\|_2 + \eta^2 \sum_{j \in [K-1]} (K - j)\|\nabla f(x_j)\|_2 \right)$$

$$\le 44\eta^3 K^3 \kappa^{1.5} d\log d + 44\eta^3 K^3 \kappa^{1.5} d\log d + 121\eta^4 K^4 \kappa^2 d \le 0.001\eta^2\kappa d.$$

The last inequality used the assumption $\eta \le \frac{1}{100000 K^3 \sqrt{\kappa}\log d}$. Combining the above bounds with Lemma 20 yields the claim. $\qquad\qquad\square$

**Proposition 5.** *For $\eta^2 K = O\left(\frac{\sqrt{\log d}}{\kappa\sqrt{d}}\right)$ and $K = O\left(d^{0.099}\right)$, there is a target density on $\mathbb{R}^d$ whose negative log-density is $\kappa$ smooth, such that relaxation time of HMC is $\Omega\left(\frac{\kappa d}{K^2}\right)$.*

*Proof.* It is straightforward to check that such a range of $\eta$ and $K$ satisfies the assumptions of Lemma 23. Applying Lemma 23 with the hard function $f_{\mathrm{hard}}$, the remainder of the proof follows analogously to that of Theorem 4. $\qquad\qquad\square$

We give a brief discussion of the implications of Proposition 5. For $\eta^2 K = \omega(\frac{\sqrt{\log d}}{\kappa\sqrt{d}})$, the proof of Theorem 4 rules out a polynomial relaxation time. In the remaining range, Proposition 5 implies that for small $K = O\left(d^{0.099}\right)$, the most we can improve the relaxation time of MALA (Theorem 2)

by taking multiple steps in HMC is by a $K^2$ factor. Since each iteration takes $K$ gradients, this is roughly an improvement of $K$ in the query complexity, and strengthens Theorem 4 for small $K$.

## B.2 Mixing time lower bound for small $K$

In this section, we first use prior results to narrow down the range of $\eta$ we consider (assuming $K$ is small). We then generalize the ideas of Section 5, our MALA mixing lower bound, to this setting.

**Mixing time lower bound for large $\eta$.** Suppose $K = O\left(d^{0.099}\right)$ throughout this section. The arguments of Section 6, specifically Proposition 4 and Lemma 18, imply mixing time lower bounds for all $\eta K = \Omega(\frac{1}{\sqrt{\kappa}})$ (using the "boosting constants" argument of Section 6.2.2 for sufficiently large $\kappa$ as necessary). For $\eta K = O(\frac{1}{\sqrt{\kappa}})$, the proof of Theorem 4 further implies mixing time lower bounds for all $\eta^2 K = \omega(\frac{\sqrt{\log d}}{\kappa \sqrt{d}})$. Hence, we can assume $\eta K = O(\frac{1}{\sqrt{\kappa}})$ and $\eta^2 K = O(\frac{\sqrt{\log d}}{\kappa \sqrt{d}})$.

Next, under the further assumption that $K = O\left(d^{0.099}\right)$, it is easy to check under the specified assumptions on $\eta$ and $K$, the preconditions of Lemma 23 are met. This implies that we can rule out $\eta^2 = \omega(\frac{\log d}{\kappa d})$ for polynomial-time mixing. Thus, in the following discussion we assume

$$K = O\left(d^{0.099}\right), \ \eta^2 = O\left(\frac{\log d}{\kappa d}\right). \tag{36}$$

**Mixing time lower bound for small $\eta$.** Let $\pi^* = \mathcal{N}(0, \mathbf{I})$ be the standard $d$-dimensional multivariate Gaussian. We will let $\pi_0$ be the marginal distribution of $\pi^*$ on the set

$$\Omega := \left\{ x \mid \|x\|_2^2 \leq \frac{1}{2} d \right\}.$$

Recall from Lemma 4 that $\pi_0$ is a $\exp(d)$-warm start. Our main proof strategy will be to show that for small $\eta$ and $K$ as in (36), after $T = O(\frac{\kappa d}{K^2 \log^3 d})$ iterations, with constant probability both of the following events happen: no rejections occur throughout the Markov chain, and $\|x_{t,K}\|_2^2 \leq \frac{9}{10} d$ holds for all $t \in [T]$. Combining these two facts will demonstrate our total variation lower bound.

**Lemma 24.** *Let $\{x_{t,k}, v_{t,k}\}_{0 \leq t < T, 0 \leq k \leq K}$ be the sub-iterates generated by the HMC Markov chain with step size $\eta^2 = O\left(\frac{\log d}{\kappa d}\right)$ and $\eta^2 K^2 \leq 1$, for $T = O(\frac{\kappa d}{K^2 \log^3 d})$ and $x_0 \sim \pi_0$; we denote the actual HMC iterates by $\{x_t\}_{0 \leq t < T}$. With probability at least $\frac{99}{100}$, both of the following events occur:*

1. *Throughout the Markov chain, $\|x_t\|_2 \leq 0.9\sqrt{d}$.*
2. *Throughout the Markov chain, the Metropolis filter never rejected.*

*Proof.* Let $h = \frac{1}{2}\eta^2$. We inductively bound the failure probability of the above events in every iteration by $\frac{0.01}{T}$, which will yield the claim via a union bound. Take some iteration $t + 1$, and note that by triangle inequality, and assuming all prior iterations did not reject,

$$\|x_{t+1,K}\|_2 \leq \|x_{0,0}\|_2 + \eta K \left\| \sum_{s=0}^{t} v_{s,0} \right\| + \eta^2 K \sum_{s=0}^{t} \sum_{k=1}^{K} \|x_{s,k}\|_2 \leq \|x_{0,0}\|_2 + 0.9\eta^2 K^2 T\sqrt{d} + \eta K \|G_t\|_2$$

$$\leq 0.8\sqrt{d} + \eta K \|G_t\|_2.$$

Here, we applied the inductive hypothesis on all $\|x_{s,k}\|_2$, the initial bound $\|x_{0,0}\|_2 \leq \sqrt{\frac{1}{2}d}$, and

that $\eta^2 K^2 T = o(1)$ by assumption. We also defined $G_t = \sum_{s=0}^{t} v_{t,0}$, where $v_{t,0}$ is the random Gaussian used by HMC in iteration $k$; note that by independence, $G_t \sim \mathcal{N}(0, t+1)$. By Fact 2, with probability at least $\frac{1}{200T}$, $\|G_t\|_2 \leq 2\sqrt{Td}$, and hence $0.8\sqrt{d} + \eta K \|G_t\|_2 \leq 0.9\sqrt{d}$, as desired.

Next, we prove that with probability $\geq 1 - \frac{1}{200T}$, step $t$ does not reject. This concludes the proof by union bounding over both events in iteration $t$, and then union bounding over all iterations. By Corollary 4 and the calculation in Lemma 15, when $\eta^2 K^2 \leq 1$, the accept probability is

$$\min\left(1, \exp\left(\frac{h}{4}\left((2\alpha - \alpha^2)\|x_{t,0}\|_2^2 - \beta^2 \|v_{t,0}\|_2^2 - 2(1-\alpha)\beta \langle x_{t,0}, v_{t,0}\rangle\right)\right)\right),$$

for some $\alpha \in \left[0.8hK^2, hK^2\right]$ and $\beta \in \left[0.8\sqrt{2h}K, \sqrt{2h}K\right]$. We lower bound the argument of the exponential as follows. With probability at least $1 - d^{-5} \geq 1 - \frac{1}{400T}$, Facts 1 and 2 imply both of the events $\|v_{t,0}\|_2^2 \leq 2d$ and $\langle x_{t,0}, v_{t,0}\rangle \leq 10\sqrt{\log d}\|x_{t,0}\|_2$ occur. Conditional on these bounds, we compute (using $2\alpha \geq \alpha^2$ and the assumption $\|x_t\|_2 \leq 0.9\sqrt{d}$)

$$\left(2\alpha - \alpha^2\right)\|x_{t,0}\|_2^2 - \beta^2 \|g\|_2^2 - 2(1-\alpha)\beta \langle x_{t,0}, g\rangle \geq -4hK^2d - 40\sqrt{h}K\sqrt{d\log d} \geq -O(K^2 \log d).$$

Hence, the acceptance probability is at least

$$\exp\left(-O\left(\eta^2 K^2 \log d\right)\right) \geq 1 - \frac{1}{400T},$$

by our choice of $T$ with $T\eta^2 K^2 \log d = o(1)$, concluding the proof. □

**Proposition 6.** *The HMC Markov chain with step size $\eta^2 = O\left(\frac{\log d}{\kappa d}\right)$ and $\eta^2 K^2 \leq 1$ requires $\Omega(\frac{\kappa d}{K^2 \log^3 d})$ iterations to reach total variation distance $\frac{1}{e}$ to $\pi^*$, starting from $\pi_0$.*

*Proof.* The proof is identical to Proposition 3, where we use Lemma 24 instead of Lemma 11. □