# OpenReview forum: "Lower Bounds on Metropolized Sampling Methods for Well-Conditioned Distributions"
_NeurIPS.cc/2021/Conference — NeurIPS 2021 Oral_

### Official Review · Reviewer_QuWv · 2021-06-24

**Rating:** 8
**Confidence:** 5

**Summary:**

This paper proves lower bounds on the performance of two important sampling algorithms, the Metropolis-adjusted Langevin algorithm (MALA) and Hamiltonian Monte Carlo (HMC), for strongly log-concave and log-smooth target distributions. The paper first proves upper bounds on the spectral gap for these algorithms, assuming that the target is either Gaussian or more generally log-concave. These results are then used to prove lower bounds on the mixing time of MALA for exponential warm starts, which are perhaps the clearest and most compelling results obtained. Therefore, I will summarize this result in more detail.

Suppose that the target distribution is strongly log-concave and log-smooth in dimension d and with condition number κ. Previous algorithmic works have shown that MALA mixes after dκ steps from an exponentially warm start (which can be obtained by initializing at a Gaussian with appropriate variance). On the other hand, the recent work by Chewi et al. shows that MALA mixes after poly(κ) d^{½} steps under a polynomially warm start (and it is unknown how to achieve such a warm start in practice). This current paper sheds light on this situation by constructing a hard target distribution such that when initialized at a particular exp(d) warm start, MALA requires κd steps to mix.

**Limitations And Societal Impact:**

Yes.

**Main Review:**

This is an excellent addition to the sampling literature: it studies two of the most important sampling algorithms using only natural assumptions and constitutes another major step towards understanding the complexity of these algorithms (and of the sampling problem more generally). Therefore, I strongly recommend acceptance.

However, there is an important detail which is glossed over in the introduction of the paper and deserves to be carefully discussed. Namely, one of the core results of this paper, the mixing time lower bound for MALA discussed above, is obtained by constructing a very specific initialization which is exponentially warm. Notably, this initialization is not the standard one used to initialize MALA (a Gaussian with appropriate variance), so the lower bound does not apply to the MALA algorithm as used in practice. Rather, this paper proves the weaker statement that the worst-case mixing time of MALA, over all target distributions in the class AND over all initializations with warm start parameter <= exp(d), is at least dκ. In other words, this paper shows that the previous analyses of MALA which achieve dκ cannot be improved, in the sense that the prior analyses assume only that the initialization is exp(d) warm and obtain their mixing time bounds (and so the analyses do not rule out that the initialization could be the adversarial one constructed in this paper), but it does not preclude the possibility that there are feasible ways of initializing MALA (indeed, perhaps even the standard one with a Gaussian) which mix faster than dκ.

Therefore, it seems that an important problem left open in this work is whether MALA initialized at an appropriate Gaussian has a mixing time closer to the κd lower bound in this work, or closer to the faster mixing time bound of Chewi et al.

To summarize, I believe this issue is important enough to merit careful discussion upfront. Yet this issue does not change the fact that the results obtained in this paper are quite interesting and important.

**Time Spent Reviewing:**

1

---

> ### Author Response · Authors · 2021-08-10
> **Response to Reviewer QuWv**
>
> Thank you very much for your thoughtful feedback and kind review. We greatly appreciate that you found our step towards understanding the algorithms we study to be important.
>
> As you pointed out, we chose the following notion of lower bounding an algorithm’s performance: $\sup_{\text{starts of warmness beta}} \inf_{\text{parameters of algorithm}} \text{runtime(parameters, warmness)}$. We chose this measure of evaluation for the following main reason. To the best of our knowledge all (Metropolized, with TV guarantees) sampling algorithms in the literature studying our problem setting and more general (non-well conditioned) logconcave sampling give guarantees depending on the starting distribution only through a warmness bound. In particular, no analysis we are aware of for well-conditioned distributions is able to use specific facts about the starting distribution (e.g. Gaussianity) to obtain improvements. Hence, our lower bound format ($\sup$ over distributions of a given warmness) is strong enough to rule out the types of upper bound analysis which are currently present in the literature.
>
> We agree that our lower bound analysis does not rule out more structured analyses, which use properties of the starting distribution beyond warmness. There are other notions of lower bound which are somewhat natural, such as $\inf_{\text{Gaussian starts}} \inf_{\text{parameters of algorithm}} \text{runtime(parameters, warmness)}$ (which seems more in line with what you are proposing may yield an improvement), which are interesting but would require novel types of upper bound analyses to be compatible. We remark that (in line with the warmness assumption our lower bound uses) the best-known warmness for a Gaussian initialization is $\kappa^{d/2}$, which is indeed exponential in $d$.
>
> We believe this discussion is worthwhile, and agree that we should clarify our choice of lower bound format more carefully in the exposition when first presented. We will make sure to do so in a revision, and thank you once more for your questions.

---

> > ### Comment · Reviewer_QuWv · 2021-08-12
> > **Acknowledged**
> >
> > Thank you for the response. I agree with your points.

---

### Official Review · Reviewer_D3MQ · 2021-07-09

**Rating:** 8
**Confidence:** 3

**Summary:**

This paper studies two well-known sampling algorithms for well-conditioned log concave distributions: the Metropolos adjusted Langevin algorithm (MALA) and Hamiltonian Montecarlo (HMC). The main contributions of the paper are:

1. A lower bound $\tilde\Omega(d\kappa)$ for the mixing time of MALA from an $\exp(d)$-warm start. This bound is sharp.
2. A lower bound on the relaxation time of HMC with $K$-steps per iteration of $\tilde\Omega(\kappa\sqrt{d}/K)$.
3. Lower bounds on the relaxation time for MALA under Gaussian and general distributions.

**Main Review:**

The paper addresses very relevant algorithms and the fundamental problem of sampling from log concave distributions. It introduces novel techniques, such as the use of Chebyshev polynomials. Some of their lower bounds are nearly tight, in an area where currently lower bounds (of any type) are scarce.

The main drawback of the paper is the fact that their analyses are algorithmic dependent. However, I am not aware of any general purpose lower bounds for sampling (except for very specific cases, such as in 1D).

**Time Spent Reviewing:**

2

---

> ### Author Response · Authors · 2021-08-10
> **Response to Reviewer D3MQ**
>
> Thank you for your helpful feedback. We are glad that you found the algorithms we address relevant, and the techniques we develop interesting.
>
> Although, as you mentioned, the types of lower bounds we prove are algorithm-dependent, we aimed to give a fairly comprehensive evaluation of the MALA and HMC families of algorithms, which are amongst the most-used in practice. We hope that taking this first step of lower bounds on sampling algorithms will serve as an important stepping stone for stronger and more general sampling lower bounds in the future. We hope this discussion of our motivations elevates your view of our paper’s contributions, and thanks once again for your review.

---

> > ### Comment · Reviewer_D3MQ · 2021-08-13
> > **Answer to Reviewer's response**
> >
> > Thank you, for the response. I am aware that this is an area where many fundamental questions (such as lower bounds) are not well-understood. This paper fills an important gap there, and my assessment of this contribution is quite positive already.
> >
> > Perhaps you could comment on the potential use of the techniques you introduce beyond the context of MALA and HMC?

---

> > > ### Author Response · Authors · 2021-08-18
> > > **Introduced techniques beyond the scope of our paper**
> > >
> > > Thank you for raising this question. We believe our work has implications for further explorations in this direction, both in terms of techniques and conceptual takeaways.
> > > 1. Our techniques of further interest include the (previously-unknown) characterization of HMC as implementing Chebyshev polynomials, which allows for easier direct computations in the Gaussian case. Moreover, we provide an argument which allows for directly lower bounding a mixing time, as opposed to the prior spectral gap arguments of previous works. This may be of technical interest since black-box reductions between the two typically are difficult in the case of continuously supported distributions.
> > > 2. Our paper also raises a number of interesting questions and suggested approaches from an upper bounds (i.e. algorithms) perspective, see discussion in Section 5. For example, our lower bounds are strictly for algorithms with fixed parameters, so it may be useful to explore random step sizes in a large range; similar ideas such as random step counts have found empirical successes (e.g. the No-U-Turn sampler). Moreover, our lower bound highlights a fundamental difficulty in designing algorithms starting from warmnesses above a certain (exponentially large) threshold, ruling out approaches such as extending the improved dimension dependence of [CLA+20] to use known starting distributions. This suggests that the next step in obtaining improved dimension dependences may be to focus on constructing improved warm starts, rather than sharpening the current style of analysis.

---

### Official Review · Reviewer_xRQt · 2021-07-14

**Rating:** 9
**Confidence:** 5

**Summary:**

The paper proves a mixing time lower bound for Metropolized Langevin (MALA) on distributions with smooth and strongly convex potentials, starting at an exponential warm start. The lower bound matches the best known upper bounds. It also proves a mixing time lower bound for HMC on the class of Gaussians with condition number kappa from an exponential warm start.

**Limitations And Societal Impact:**

yes

**Main Review:**

Essential points:
This is a very interesting paper that fills in a gap in the study of lower bounds of popular sampling algorithms. To my knowledge, it is the first paper that proves a non-asymptotic lower bound for sampling in high dimensions from a feasible start. It also raises many questions that are worth pursuing for a full understanding of the well known sampling algorithms, MALA and HMC. For instance, when focusing on the dimension dependence, the result of this paper shows that MALA converges at rate d under an exponential warm start, but past work shows that when the warmness is polynomial in the dimension, the convergence occurs at rate d^{½}, so is there a sharp threshold when the rate changes? Another interesting question is whether it is possible to construct a poly(d) warm start, using fewer than d queries to the potential. If this is not possible, then this paper would have nailed down the dimension dependence of MALA on distributions with smooth and strongly convex potentials.

Suggestions:
In Section 3 of supplementary: first paragraph says deriving a “lower bound on the spectral gap”, but instead it should be lower bound on relaxation time, or upper bound on spectral gap.


**Time Spent Reviewing:**

4

---

> ### Author Response · Authors · 2021-08-10
> **Response to Reviewer xRQT**
>
> Thank you very much for your kind review. We very much appreciate that you found our paper interesting, the problems we study useful to pursue, and the question our work raises worthy of further investigation.
>
> We agree with your comment regarding the spectral gap, and will fix this in a revision.

---

> > ### Comment · Reviewer_xRQt · 2021-08-13
> > **Acknowledgement**
> >
> > Thank you for your response.

---

### Decision · Program_Chairs · 2021-09-27

**Decision:**

Accept (Oral)

**Comment:**

This paper provides lower bounds on mixing and relaxation times of popular sampling algorithms for log-concave distributions. The reviewers all agree that this is a strong contribution that fills in a gap in the existing literature. This is a strong paper!